# Shared pathway-specific network mechanisms of dopamine and deep brain stimulation for the treatment of Parkinson's disease

Thomas S. Binns [1,2,3], Richard M. Köhler [1], Jojo Vanhoecke [1], Meera Chikermane [1], Moritz Gerster[3,4,5], Timon Merk [1], Franziska Pellegrini[3,6], Johannes L. Busch [1], Jeroen G. V. Habets[1], Alessia Cavallo[1,2,3], Jean-Christin Beyer[1], Bassam Al-Fatly [1], Ningfei Li [1], Andreas Horn[1,2,7,8], Patricia Krause[1], Katharina Faust[9], Gerd-Helge Schneider[9], Stefan Haufe [2,3,6,10,11], Andrea A. Kühn [1,2,3,12,13] & Wolf-Julian Neumann [1,2,3] ✉

Deep brain stimulation is a brain circuit intervention that can modulate distinct neural pathways for the alleviation of neurological symptoms in patients with brain disorders. In Parkinson's disease, subthalamic deep brain stimulation clinically mimics the effect of dopaminergic drug treatment, but the shared pathway mechanisms on cortex – basal ganglia networks are unknown. To address this critical knowledge gap, we combined fully invasive neural multisite recordings in patients undergoing deep brain stimulation surgery with normative MRI-based whole-brain connectomics. Our findings demonstrate that dopamine and stimulation exert distinct mesoscale effects through modulation of local neural population activity. In contrast, at the macroscale, stimulation mimics dopamine in its suppression of excessive interregional network synchrony associated with indirect and hyperdirect cortex – basal ganglia pathways. Our results provide a better understanding of the circuit mechanisms of dopamine and deep brain stimulation, laying the foundation for advanced closed-loop neurostimulation therapies.

Parkinson's disease (PD) is the fastest-growing neurodegenerative disorder[1,2], characterised by a loss of dopaminergic neurons in the substantia nigra[3]. Administration of the dopamine precursor levodopa and high-frequency deep brain stimulation (DBS) of the subthalamic nucleus (STN) in the basal ganglia are both established approaches for ameliorating the motor impairments of PD. However, the precise mechanisms of therapeutic action remain a matter of debate[4]. Local field potentials (LFP) recorded from PD patients undergoing surgery for the implantation of DBS electrodes have revealed excessive synchronisation of beta band (12–30 Hz) activity in the STN as a hallmark of the dopamine-depleted Parkinsonian state, which can be suppressed by dopaminergic medication and subthalamic DBS (STN-DBS)[5–9]. Cortex – basal ganglia interactions play a critical role in motor control, with the STN receiving input from cortical layer 5 neurons via two distinct pathways: the polysynaptic indirect pathway through the striatum; and the monosynaptic hyperdirect pathway from cortex to STN, thought to provide a rapid inhibition of movement[10–13]. Computational and animal models have highlighted monosynaptic hyperdirect input from the cortex as a key factor in the origin of pathological subthalamic synchrony, and the hyperdirect pathway has long been proposed as the primary target for clinical DBS effects[14–19]. However, in addition to the unclear effects of dopamine on the hyperdirect

pathway[19,20], methodological constraints have also hindered a rigorous investigation of the effects of dopamine and DBS on pathway-specific network activity in PD. Thus, a strategic knowledge gap remains in the question of how dopaminergic medication – the primary and most effective treatment for PD – affects the hyperdirect pathway, and to what degree DBS mimics neural circuit mechanisms of dopaminergic innervation. Such findings have critical relevance for the possibility of DBS to mimic temporally precise dopaminergic activity, with the lack of understanding posing a significant barrier to the development of novel brain circuit interventions for PD, and the extension of DBS to other brain disorders. In addition, understanding the effects of dopamine and DBS on cortex – STN network activity can highlight further Parkinsonian biomarkers, which in future adaptive stimulation paradigms could be incorporated to offer a richer picture of the pathological state for improved clinical outcomes[21,22]. To overcome these hurdles, we developed a multimodal approach to compare the neural circuit effects of levodopa and DBS with fully invasive cortex – STN multisite intracranial EEG recordings in patients undergoing DBS electrode implantation for PD, combined with normative MRI-based whole-brain connectivity mapping. We characterised

neurophysiological cortex – STN interactions using: imaginary coherency-based measures for undirected spectral coupling[23,24]; Granger causality for directed spectral coupling, characterising the direction of information flow[25,26]; and bispectral time delay analysis for estimating latencies of information transfer[27]. Using this approach, we provide direct human evidence for differential and shared pathway-specific effects of dopamine and DBS, from local neural population activity to whole-brain interregional communication.

## Results

To investigate the effects of dopamine and neuromodulation on pathological cortex – basal ganglia communication, we performed invasive electrophysiological recordings in 21 PD patients (see Supplementary Table 1) undergoing bilateral implantation of DBS electrodes to the STN. Resting-state recordings of unilateral electrocorticography (ECoG) targeted at the sensorimotor cortex and subthalamic LFP (STN-LFP) were performed through externalised leads in the days following surgical implantation of the electrodes (see Supplementary Table 2). The ECoG electrodes were later retracted at the time of neurostimulator implantation. This enabled the first

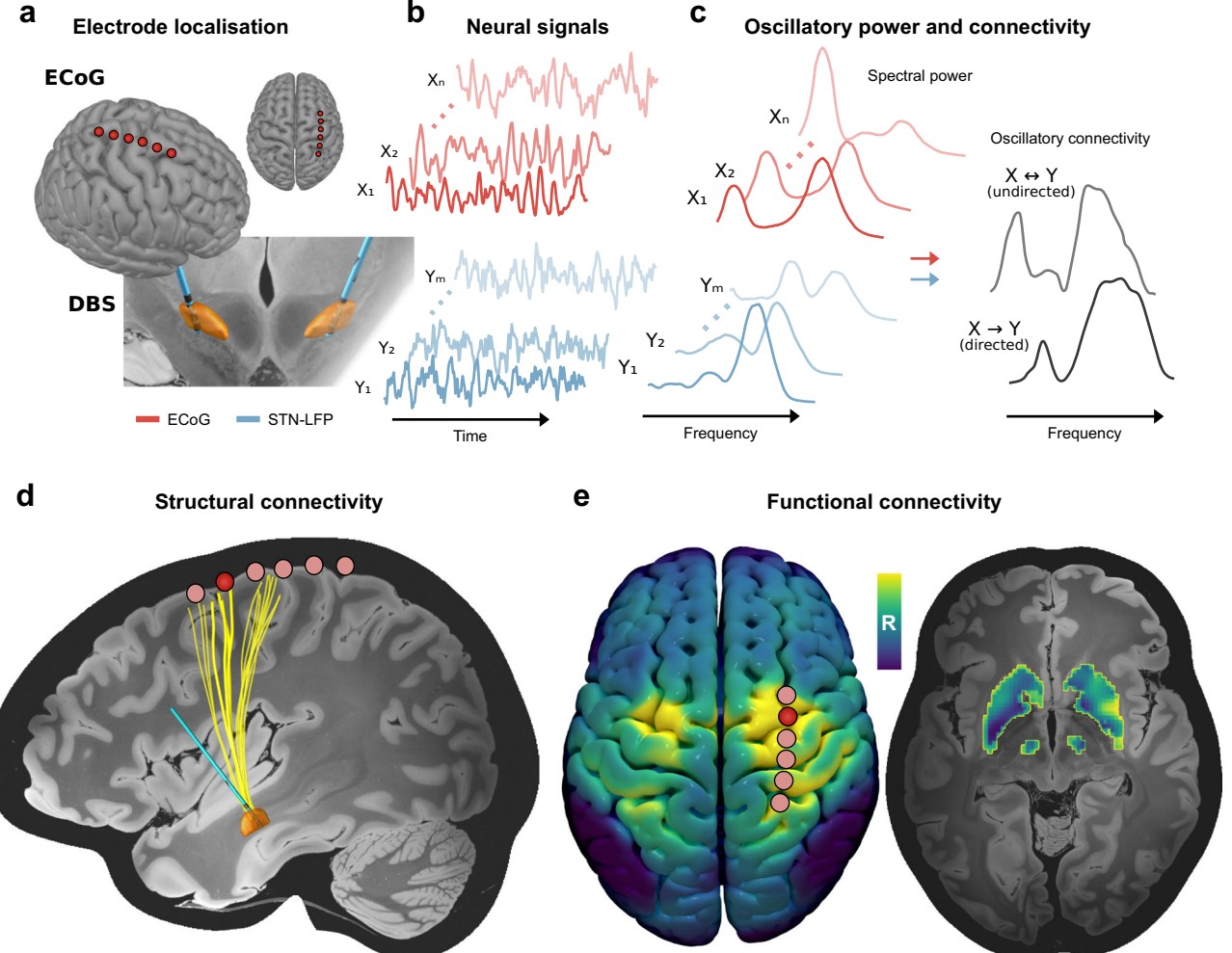

**Fig. 1 | Overview of data collection and analysis.** ECoG strips and DBS leads were implanted in Parkinson's disease patients. **a** Electrodes were localised using preoperative and postoperative neuroimaging. **b** Neurophysiological recordings were performed across multichannel electrodes for cortex and STN (coloured in orange). **c** Recordings of ECoG and STN-LFP signals were analysed to characterise local oscillatory power, in addition to the (un)directed oscillatory communication between the sites. **d** Atlas-based structural connectomes were used to map the hyperdirect pathway fibres connecting ECoG and STN-LFP

recording sites for subsequent comparison with oscillatory connectivity (single ECoG contact and the corresponding projections to STN highlighted).
**e** Normative functional connectomes were used to create whole-brain functional connectivity maps seeded from a given ECoG contact (highlighted; left) to capture connectivity between cortex and subcortical nuclei of the indirect pathway (right). DBS deep brain stimulation, ECoG electrocorticography, LFP local field potential, STN subthalamic nucleus.

systematic characterisation of the effects of dopamine and STN-DBS on invasively recorded human cortex – basal ganglia communication through repetitions of recordings after withdrawal and administration of dopaminergic medication (OFF therapy and ON levodopa, respectively; $n = 18$) and STN-DBS (ON STN-DBS; $n = 12$). After verification of anatomical localisation (Fig. 1a) and signal fidelity, all multisite recordings (Fig. 1b) were subjected to an analysis of local oscillatory power and connectivity, employing data-driven multivariate analytic approaches based on spatial filters and state-space models for the estimation of power as well as directed and undirected coupling[23,25,26,28] (Fig. 1c). These multivariate approaches maximise signal-to-noise ratio in the frequency domain, whilst providing interpretable spatial weights for each signal. Following this, bispectral analysis was performed to determine the time delay of information flow from cortex to STN[27]. Finally, coupling metrics were correlated with whole-brain MRI connectivity derived from precisely curated fibre tracts for DBS research[29] (Fig. 1d) and large-scale normative fMRI connectomes from PD patients[30,31] (Fig. 1e) to determine the association between cortex – STN signalling pathways and spectral coupling bands. Unless stated otherwise, statistical results were obtained from non-parametric permutation tests with 100,000 permutations at an alpha level of 0.05, with cluster correction to control for multiple comparisons where appropriate[32].

## Dopamine and DBS exert distinct effects on local population activity of cortex and STN

To investigate the effects of dopamine and high-frequency stimulation on local neural population activity, we decomposed cortical and subthalamic oscillatory activity in the frequency domain with Fourier analyses based on multitapers. Dopamine, in parallel with its effective symptom alleviation (UPDRS-III reduction $21.2 \pm 1.4$, mean ± SEM; $p < 0.05$), had frequency-specific modulatory effects on neural population synchrony measured as spectral power across brain areas. At the cortical level, dopamine suppressed canonical high beta (20–30 Hz) power (Fig. 2a; $p < 0.05$). A more conservative frequency-wise comparison further revealed distinct clusters from 8–10 Hz for increased mu rhythm/alpha (8–12 Hz) power alongside a high beta suppression between 22.5–27.5 Hz (both $p < 0.05$, cluster-corrected). At the subcortical level, we reproduced the well-described modulation of low beta (12–20 Hz) power in the STN[5,7,9] (Fig. 2a), with a significant reduction in grand average canonical low beta ($p < 0.05$) and a corresponding cluster from 11.5–19.5 Hz in the frequency-wise comparison ($p < 0.05$, cluster-corrected). DBS did not modulate cortical power spectra, even when selecting only contacts over the motor cortex, confirming some previous reports[33,34], but suppressed canonical high beta in the STN ($p < 0.05$) with a significant cluster extending from 24.5–27.5 Hz ($p < 0.05$, cluster-corrected). Significant suppression of STN low beta power with DBS was not observed ($p > 0.05$), potentially reflecting an absence of peaks in the low beta band. However, when isolating those STN-LFP channels with peaks in the low beta band, there remained no significant suppression in low beta power with DBS ($p > 0.05$; Supplementary Fig. 1). In contrast, channels with predominantly low beta and predominantly high beta peaks showed significant suppression of high beta power with DBS ($p < 0.05$). Finally, for the possibility that the distinct therapeutic mesoscale effects were influenced by differences between the subjects in the medication and stimulation groups, we re-analysed the data from only those 9 subjects with both ON levodopa and ON STN-DBS recordings. Permutation tests again showed significant reductions in cortex high beta power and STN low beta power with dopamine and a significant reduction in STN high beta power with DBS, as described for the full cohort above (Supplementary Fig. 2).

We next sought to examine the spatial contributions to spectral power using spatio-spectral decomposition – a multivariate approach that captures the strongest component of band power and its spatial

contributions[28,35] (Fig. 2b). We demonstrate the utility of spatio-spectral decomposition in a subject with a novel 16-channel DBS electrode (Fig. 2c; Boston Scientific Cartesia X), showing patient-specific precision mapping of low beta oscillations in proximity to the optimal stimulation target and the clinically most effective contacts. On the group level, this approach revealed the spatio-spectral specificity of the local population activity, with contributions localised to the motor cortex for high beta and dorsolateral STN for low beta (Fig. 2b; see Table 1 for MNI coordinates of peak contributions across frequencies and targets). Furthermore, motor cortex high beta localisations showed a striking similarity across medication and stimulation states, suggesting that the lack of modulation of grand average cortical high beta with DBS was not due to more spatially specific suppression of activity compared to dopamine (Supplementary Fig. 3). In addition, mu rhythm/alpha mapped most strongly to sensory and parietal cortices (Supplementary Fig. 4). Our results highlight the complex spatio-spectral patterns at which dopamine and DBS differentially modulate local neural population activity recorded with intracranial EEG. Most prominently, dopamine but not DBS suppressed high beta motor cortex activity, while increasing sensory and parietal mu rhythm/alpha. In the STN, dopamine suppressed low beta but not high beta power, while STN-DBS predominantly suppressed high beta power. These spectrally specific effects will be important to consider for the development of closed-loop DBS algorithms for adaptive DBS.

## Modulation of cortex – basal ganglia coupling is a shared mechanism of dopamine and DBS

Following the extraction of oscillatory power in cortex and STN, we aimed to characterise macroscale interregional cortex – STN oscillatory communication and the associated changes with dopamine and stimulation. For this, we utilised three distinct analytic approaches: (1) spatio-spectral patterns of undirected communication with imaginary coherency-based metrics[23,24]; (2) directional communication with Granger causality-based methods[25,26]; and (3) time delays of information flow with bispectral analysis[27]. First, oscillatory connectivity between cortical and subthalamic recording locations was determined across all available channel pairs using the grand average bivariate imaginary part of coherence – a measure of correlation in the frequency domain immune to spurious connectivity estimates from zero time-lag interactions such as volume conduction[24]. This revealed connectivity in the mu rhythm/alpha, low beta, and high beta ranges (Fig. 3b). Both dopamine and DBS induced a strikingly similar modulatory effect on high beta cortex – STN connectivity associated with the hyperdirect pathway[19,34], suppressing canonical high beta coupling (both $p < 0.05$) but not low beta coupling (both $p > 0.05$). Similarly, bin-wise comparisons revealed significant clusters of suppressed connectivity between 25–27 Hz (levodopa) and 30.5–32.5 Hz (STN-DBS; both $p < 0.05$, cluster-corrected). In contrast, a unique effect of DBS was identified, with an elevation of canonical mu rhythm/alpha connectivity ($p < 0.05$) and a corresponding significant cluster from 9.5–13.0 Hz ($p < 0.05$, cluster-corrected). However, the grand average bivariate approach described above does not provide information on spatial contributions to coupling and possesses a limited signal-to-noise ratio, given the inclusion of electrodes in regions outside the sources of oscillatory coupling. To overcome these limitations, we next analysed the maximised imaginary part of coherence – a multivariate extension that extracts the strongest connectivity component between two sets of channels and their spatial contributions[23,35]. Multivariate analysis revealed contributions to high beta coupling peaked in motor cortex and dorsolateral STN (Fig. 3a; see Table 2 for MNI coordinates of peak contributions across frequencies and targets), and recapitulated the shared therapeutic suppression of high beta coupling with dopamine and DBS (both $p < 0.05$). These findings present the modulation of oscillatory communication between the motor cortex and dorsolateral STN as a shared therapeutic network

**a** Differential modulation of power in cortex and STN by dopamine and DBS

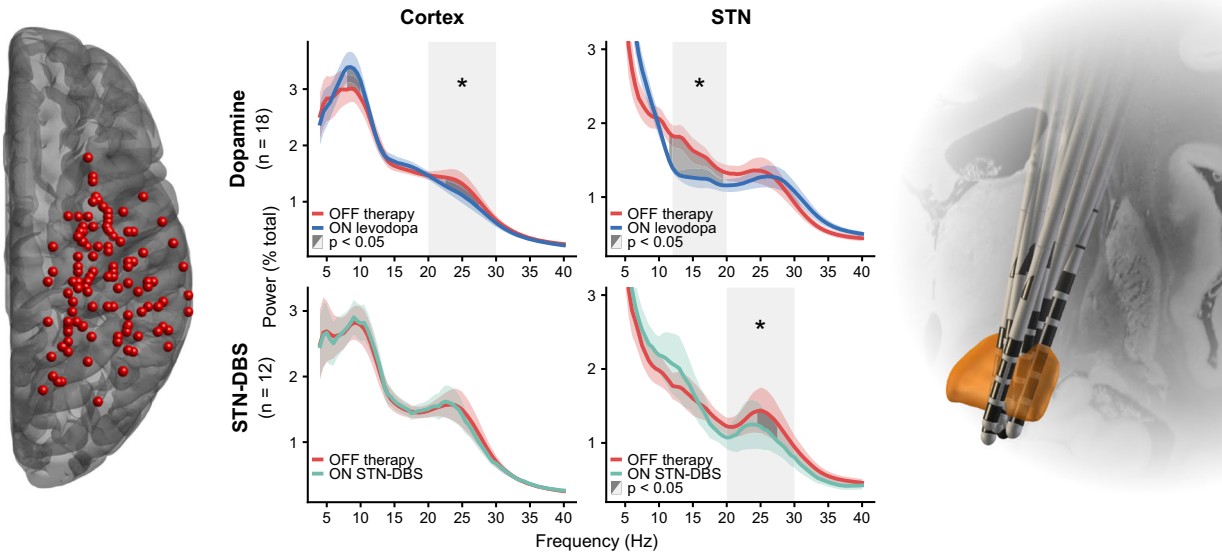

**b** Spatio-spectral decomposition reveals motor areas as sources of beta oscillations

**c** Precision mapping of local power with spatio-spectral decomposition

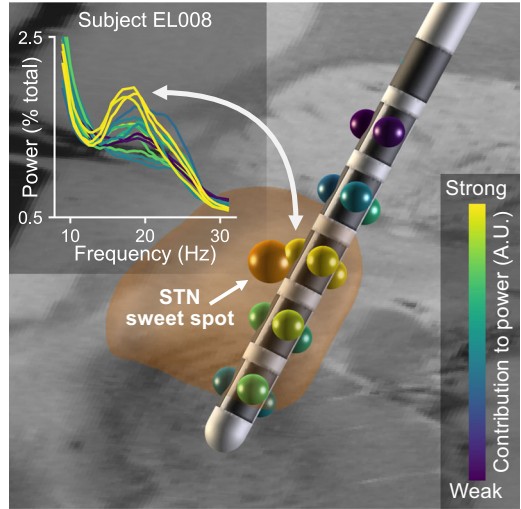

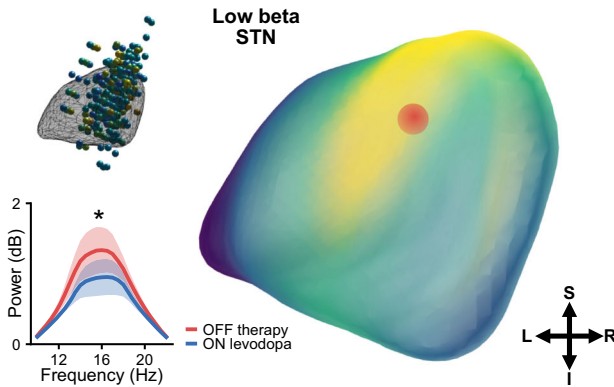

mechanism of dopamine and DBS, contrasting with the complex spectrally specific local effects on power.

### Cortex drives pathological subthalamic beta activity in Parkinson's disease

To quantify the direction of information flow between cortex and STN, we applied a multivariate form of time-reversed Granger causality[25,26].

Multivariate Granger causality quantifies the degree to which one set of signals, $X$, predicts another set of signals, $Y$, from which directionality is determined. Contrasting the Granger scores of $X \rightarrow Y$ and $Y \rightarrow X$ provides a measure of net directionality, revealing the driver-recipient communication relationship. Net Granger scores can be positive and negative, with positive values representing a dominance of information flow from cortex to STN. Finally, contrasting the net

**Fig. 2 | Cortical and subthalamic spectral power. a** There are distinct modulations in the grand average power spectra across all bipolar contacts in the cortex and STN OFF therapy, ON levodopa, and ON STN-DBS (centre; levodopa-cortex high beta, $t = 0.585$, $p = 0.01130$; levodopa-STN low beta, $t = 1.172$, $p = 0.00003$; DBS-STN high beta, $t = 0.858$, $p = 0.01842$), shown alongside locations of ECoG strips (left) and DBS leads (right) for the cohort. Shaded light grey areas indicate a significant difference in the average values of canonical frequency bands between conditions. Shaded dark grey areas indicate clusters of significant differences between conditions for the respective frequency bins. **b** Imaging spatial contributions to power with spatio-spectral decomposition reveals the motor cortex as the strongest contributor to high beta power in the cortex (red dot), and the dorsolateral aspect as the strongest contributor to low beta power in STN (red dot; within 0.5 mm of the previously established STN sweet spot[30]). Localisations are shown for individual electrodes and interpolated to anatomical surfaces. Spatio-spectral decomposition reproduces the medication-induced suppression of high beta power in the cortex ($t = 0.569$, $p = 0.00060$) and low beta power in STN ($t = 0.545$, $p = 0.01591$), shown as insets. **c** Precision mapping using a novel 16-contact DBS lead for low beta power in a single subject highlights the immediate relationship of spatio-spectral decomposition spatial contributions, univariate power spectra, and the optimal therapeutic DBS target (STN sweet spot)[30]. All panels: Shaded coloured areas show standard error of the mean; * $p < 0.05$; statistical results obtained from two-sided permutation tests with cluster correction to control for multiple comparisons where appropriate. Source data are provided as a Source Data file. A anterior, DBS deep brain stimulation, ECoG electrocorticography, I inferior, L left, P posterior, R right, S superior, STN subthalamic nucleus.

## Table 1 | Therapeutic effects on local power

| Region | Frequency band | Dopamine effect | DBS effect | Localisation (MNI) |
|---|---|---|---|---|
| Cortex | Mu rhythm/alpha | Elevation | No effect | 46.0, −50.0, 60.0 |
| Cortex | High beta | Suppression | No effect | 44.0, −6.0, 60.0 |
| STN | Low beta | Suppression | No effect | 12.2, −12.6, −6.4 |
| STN | High beta | No effect | Suppression | 12.0, 12.6, −6.6 |

Localisations taken as the point of strongest contribution to power in the spatio-spectral decomposition maps. *DBS* deep brain stimulation, *MNI* Montreal Neurological Institute coordinate space, *STN* subthalamic nucleus.

Granger scores with those obtained on the time-reversed signals eliminates spurious connectivity estimates arising from non-causal signal interactions. This provides one of the most conservative statistical validation methods for the presence of true physiological oscillatory coupling[26,36]. Between all recording locations, cortex drove connectivity with STN in the 8–50 Hz range across medication states ($p < 0.05$), indicative of true physiological oscillatory connectivity in the mu/alpha, low beta, and high beta frequency ranges. Similar profiles were observed when selecting ECoG channels according to anatomical location for motor and sensory cortices, which we subsequently analysed as the multivariate approach employed does not provide spatial information on the sources of activity. Here, directed communication from the motor cortex was not significantly altered by medication (Fig. 3c; $p > 0.05$), nor was communication from the sensory cortex by medication and stimulation (Supplementary Fig. 5). In contrast, stimulation suppressed the dominance of directed motor cortex to STN information flow in the 26.5–30.0 Hz range ($p < 0.05$; cluster-corrected), overlapping with the canonical high beta band. Further analysis suggested that this change in driver-recipient relationship with DBS was due to a selective reduction of information flow from cortex to STN, and not to increased information flow from STN to cortex (Supplementary Fig. 6). Accordingly, unique effects of stimulation were identified in the directionality of information flow between cortex and STN, indicative of a selective suppression of cortical drive with DBS in addition to the shared therapeutic suppression of high beta hyperdirect coupling.

### Cortico-subthalamic time delays suggest parallel coupling through mono- and poly-synaptic pathways

To provide additional insights into the neurophysiological underpinnings of cortico-subthalamic communication, we estimated the time delay of information flow from cortex to STN using the bispectrum – a frequency-resolved measure of non-linear signal interactions[27]. For the following analyses, we treated parietal cortex – STN interactions as a conservative physiological control where less communication is expected, securing the neuroanatomical specificity of our results. Pooled across the OFF therapy and ON levodopa conditions, the time of the strongest delay estimate, tau, occurred at $25.7 \pm 1.8$ ms for motor cortex – STN communication, timings

congruent with polysynaptic indirect pathway communication[37–39]. Significant differences in tau were not observed between medication states ($p > 0.05$; $25.6 \pm 2.5$ ms OFF therapy, $25.9 \pm 1.9$ ms ON levodopa) or stimulation states ($p > 0.05$; $26.0 \pm 3.0$ ms OFF therapy, $20.6 \pm 2.3$ ms ON STN-DBS). However, analysing only these global maxima of the strongest delay estimates neglects the reality of information flow via multiple pathways in discrete time windows, such as that which exists for the indirect and hyperdirect pathways. For this, we can instead consider the entire span of time delay estimates, identifying local maxima in the results. We first determined periods of significant communication, using parietal cortex – STN interactions as a statistical baseline (Fig. 3d). Estimates of motor cortex – STN communication were significant below 10 ms OFF therapy ($p < 0.05$, cluster-corrected), with a significant number of local peaks occurring in the 1–9 ms window OFF therapy ($p < 0.05$, Bonferroni-corrected; time of peaks $5.2 \pm 0.3$ ms), timings in line with monosynaptic hyperdirect pathway transmission[13,37,39,40]. In contrast, significant periods of communication were not observed below 10 ms with dopamine or DBS (both $p > 0.05$, cluster-corrected), nor were there a significant number of local peaks in the 1–9 ms window (both $p > 0.05$, Bonferroni-corrected), indicating a suppression of motor cortex – STN communication. In those subjects with data from both medication and stimulation states ($n = 9$), no significant differences in the degree of modulation of time delay estimate strength between dopamine and DBS were found ($p > 0.05$, cluster-corrected). Therefore, in addition to showing cortico-subthalamic communication in time windows congruent with indirect and hyperdirect pathway activity, these results further support the hypothesis that suppression of hyperdirect pathway communication is a shared therapeutic mechanism of dopamine and DBS.

### High beta oscillatory connectivity reflects structural connectivity of the hyperdirect pathway

Our results highlight the shared modulatory effects of dopamine and DBS on cortico-subthalamic oscillatory communication that may arise from distinct local changes in neural dynamics at the level of the cortex and basal ganglia. Because invasive neurophysiology alone cannot directly provide further information on indirect vs. hyperdirect pathway affiliation, and given the time delay analysis revealed evidence for the co-activation of both pathways, we have extended our analysis to

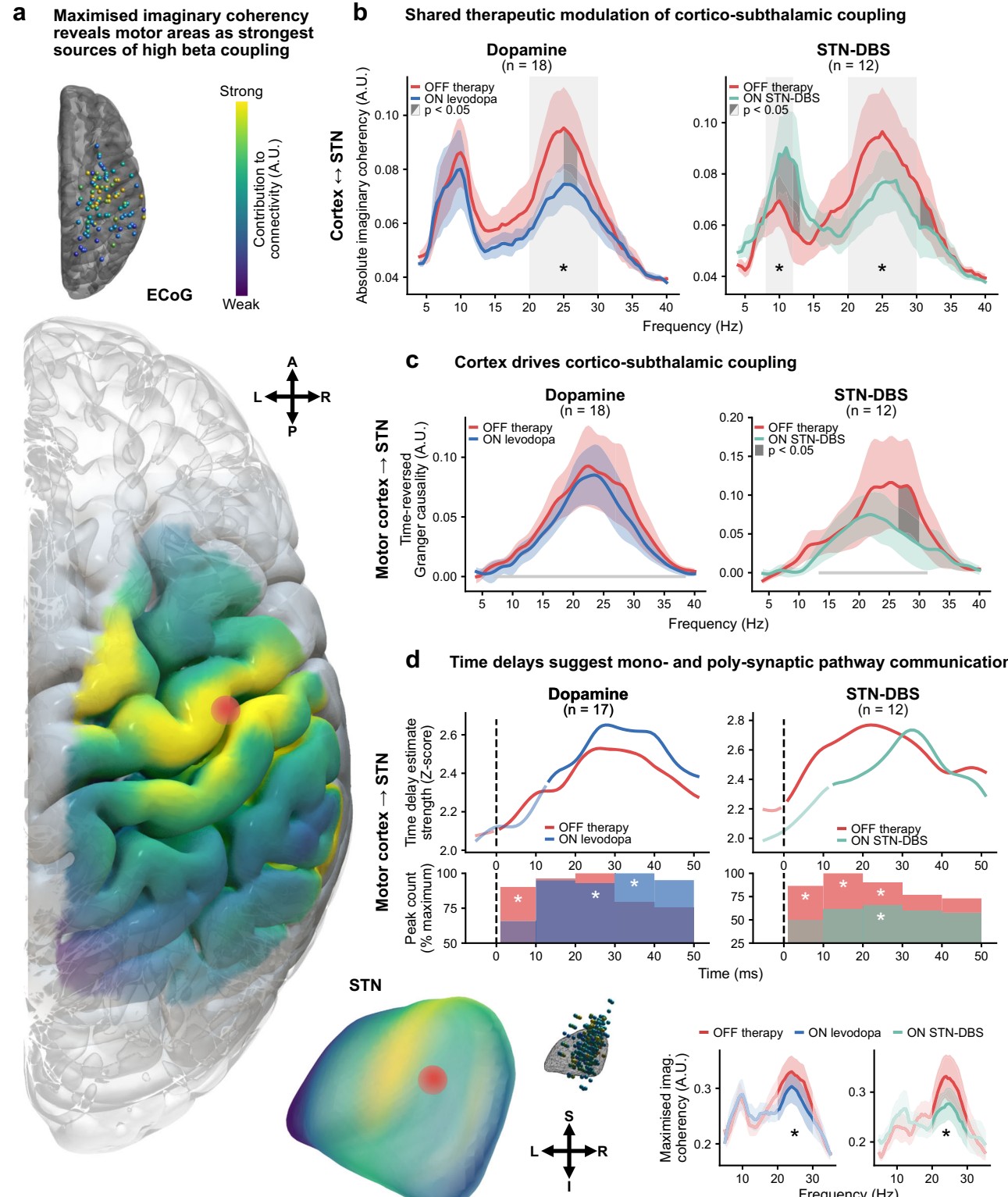

**a** Maximised imaginary coherency reveals motor areas as strongest sources of high beta coupling

**b** Shared therapeutic modulation of cortico-subthalamic coupling

**c** Cortex drives cortico-subthalamic coupling

**d** Time delays suggest mono- and poly-synaptic pathway communication

neuroimaging-based connectivity mapping. We identified the hyperdirect axonal fibres connecting the cortical and subthalamic recording locations as those fibres in the vicinity of each ECoG and STN-LFP contact, extracted from the Petersen DBS pathway atlas[29] (Fig. 4a). To analyse the relationship between hyperdirect pathway structural connectivity and oscillatory connectivity in specific frequency bands, we used a linear mixed effects model with medication state as a fixed effect and subjects as a random variable. With this model, the number of hyperdirect fibres connecting a given ECoG and STN-LFP contact

were compared to the maximised imaginary coherency spatial contribution maps. These comparisons were made for the low and high beta bands OFF therapy and ON levodopa. Thus, the model provides a statistical account for the association of oscillatory connectivity with structural hyperdirect pathway connectivity. This confirmed a significant, positive association of hyperdirect fibre counts and the contribution of contacts to high beta oscillatory connectivity ($\beta = 4.064$, $p < 0.05$), but not low beta oscillatory connectivity ($\beta = -1.142$, $p > 0.05$). In both cases, medication state did not have a significant

**Fig. 3 | Oscillatory cortico-subthalamic connectivity. a** Spatial contributions of maximised imaginary coherency reveal motor cortex and dorsolateral STN as the strongest contributors to high beta connectivity (red dots). Localisations are shown for individual electrodes and interpolated to surfaces alongside multivariate connectivity spectra. Maximised imaginary coherency reproduces the therapeutic suppression of high beta cortico-subthalamic connectivity, shown in the inset (levodopa, $t = 0.556$, $p = 0.03277$; DBS, $t = 0.615$, $p = 0.01883$). **b** Shared therapeutic modulation can be observed in the grand average imaginary coherency spectra, demonstrating suppression of cortico-subthalamic high beta connectivity with medication ($t = 0.523$, $p = 0.04312$) and DBS ($t = 0.653$, $p = 0.01752$). **c** Granger causality shows that the motor cortex drives communication with STN across medication and stimulation states, with a stimulation-specific suppression of high beta activity. Frequencies of significant connectivity OFF therapy marked as grey lines on the plots. **d** Bispectral time delay analysis highlights the contributions of mono- and poly-synaptic pathways to cortico-subthalamic communication, and the shared therapeutic suppression of monosynaptic pathway communication. Upper plots show the strength of grand average time delay estimates for each time bin, with opaque lines representing estimates which are significantly greater than the physiological control of parietal cortex – STN communication. The bottom plots show the number of peaks in each connection of the time delay estimates aggregated over 10 ms windows, with significance again determined against the physiological control of parietal cortex – STN communication. All panels: shaded coloured areas show standard error of the mean; shaded light grey areas indicate a significant difference in the average values of canonical frequency bands between conditions; shaded dark grey areas indicate clusters of significant differences between conditions for the respective frequency bins; * $p < 0.05$; statistical results obtained from two-sided (one-sided for panel **d**) permutation tests with cluster correction to control for multiple comparisons where appropriate. Source data are provided as a Source Data file. A anterior, DBS deep brain stimulation, ECoG electrocorticography, I inferior, L left, P posterior, R right, S superior, STN subthalamic nucleus.

**Table 2 | Therapeutic effects on oscillatory connectivity**

| Frequency band | Dopamine effect | DBS effect | Cortex localisation (MNI) | STN localisation (MNI) |
|---|---|---|---|---|
| Mu rhythm/alpha | No effect | Elevation | 40.0, − 60.0, 56.0 | 12.2, − 12.4, − 6.2 |
| High beta | Suppression | Suppression | 48.0, − 14.0, 58.0 | 12.0, − 12.8, − 7.0 |

Localisations taken as the point of strongest contribution to connectivity in the maximised imaginary coherency maps. *DBS* deep brain stimulation, *MNI* Montreal Neurological Institute coordinate space, *STN* subthalamic nucleus.

effect on this relationship (both $p > 0.05$). This is not unexpected, as although the degree of high beta coupling changes with dopamine, the multivariate spatial patterns capture the amount a given channel contributes to the connectivity that is present. Therefore, medication state will not have a significant effect on the relationship between hyperdirect pathway connectivity and high beta coupling in our model if the spatial patterns of coupling are stable across medication states, which is indeed the case (Supplementary Fig. 7). High beta cortico-subthalamic coupling was therefore selectively associated with hyperdirect pathway fibre tract connectivity.

## Low beta oscillatory connectivity reflects fMRI connectivity of the indirect pathway

Whilst atlas-based structural connectivity provides a straightforward estimate of connection probability for monosynaptic pathways, it has limited utility for the identification of polysynaptic connections such as the indirect pathway. To investigate the relationship of oscillatory connectivity in specific frequency bands with indirect pathway connectivity, we repeated the multimodal analysis using functional MRI connectivity derived from an openly available Parkinson's disease fMRI group connectome (previously used in Horn et al. [30,31]) based on data from the Parkinson's progression markers initiative database[41]. Taking the MNI coordinates for each ECoG contact as a seed, whole-brain connectivity maps were generated, from which the functional connectivity values to the indirect pathway nuclei of the basal ganglia (caudate, putamen, external segment of the globus pallidus (GPe), and STN) were parcellated and extracted[42]. Using a linear mixed effects model with the same architecture as for structural connectivity, the functional connectivity values were compared to maximised imaginary coherency spatial contribution maps for the ECoG contacts. Again, this comparison was made for the low and high beta bands OFF therapy and ON levodopa (Fig. 4b). Accordingly, the model provides a statistical account for the association of oscillatory coupling with fMRI connectivity. The relationship between fMRI connectivity from cortex to STN and the contribution of the cortex to oscillatory coupling was significant for both low and high beta bands (low beta, $\beta = 0.006$, $p < 0.05$; high beta, $\beta = 0.004$, $p < 0.05$). As this could reflect both indirect and hyperdirect pathway communication, we recreated the model to account for structures specific to the indirect pathway in addition to the STN, namely the putamen and GPe. This revealed a

significant relationship for low beta connectivity ($\beta = 0.005$, $p < 0.05$, Bayesian information criterion (BIC) $= -967.6$), and a similar but less robust effect for high beta ($\beta = 0.004$, $p < 0.05$, BIC $= -965.8$), the latter of which we reasoned was largely driven by the cortex – STN connection. Subsequent inclusion of putamen and GPe alone in the model confirmed this, with a significant effect observed only for the low beta band (low beta, $\beta = 0.005$, $p < 0.05$, BIC $= -946.9$; high beta, $\beta = 0.004$, $p > 0.05$; BIC $= -945.7$). There were no instances in which medication state had a significant effect on these relationships (all $p > 0.05$), again reflecting the similarity of spatial contributions to coupling across medication states (Supplementary Fig. 7). Thus, low beta cortico-subthalamic coupling was selectively associated with indirect pathway communication.

## Discussion

Our study systematically compares the effects of dopamine and STN-DBS on local mesoscale and interregional macroscale circuit communication with fully invasive neurophysiology and normative MRI connectomics in Parkinson's disease. We derive three major advances from our findings (Fig. 5). First, we demonstrate that dopamine and DBS exert distinct mesoscale spatio-spectral effects on local power. In addition to modulations of STN power described previously, we demonstrate a suppression of cortical high beta activity with dopamine mapped to motor areas of the cortex, as well as an elevation of sensory-parietal mu rhythm/alpha activity. Second, in contrast to the varying effects on local power, we identify the suppression of cortico-subthalamic high beta coupling as a shared therapeutic macroscale mechanism of dopamine and DBS. In line with prior associations of high beta coupling with hyperdirect pathway activity, we additionally identify the selective suppression of communication from cortex to STN in the sub-10 millisecond time period, associated with the transmission of information through this monosynaptic pathway. While the hyperdirect pathway has long been hypothesised to underlie the therapeutic effects of DBS, our study extends this pathway to be a target of dopaminergic effects as well. Finally, using normative connectomes of structural and functional connectivity, we demonstrate a selective association of low and high beta coupling with the activity of the indirect and hyperdirect pathways, respectively. Altogether, we argue that DBS circuit effects on cortex – basal ganglia communication can mimic the neural circuit mechanisms of dopaminergic innervation.

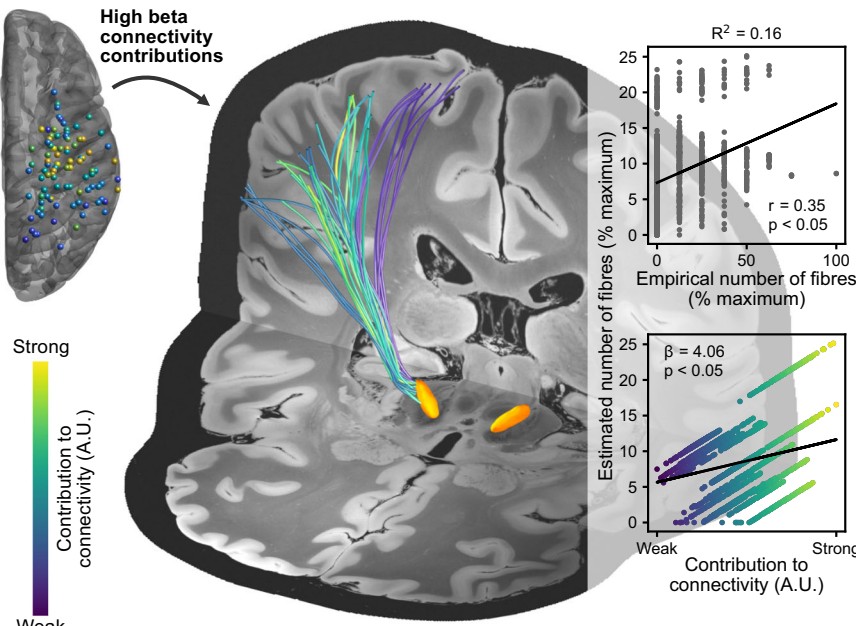

**a** **High beta coupling reflects hyperdirect pathway connectivity**

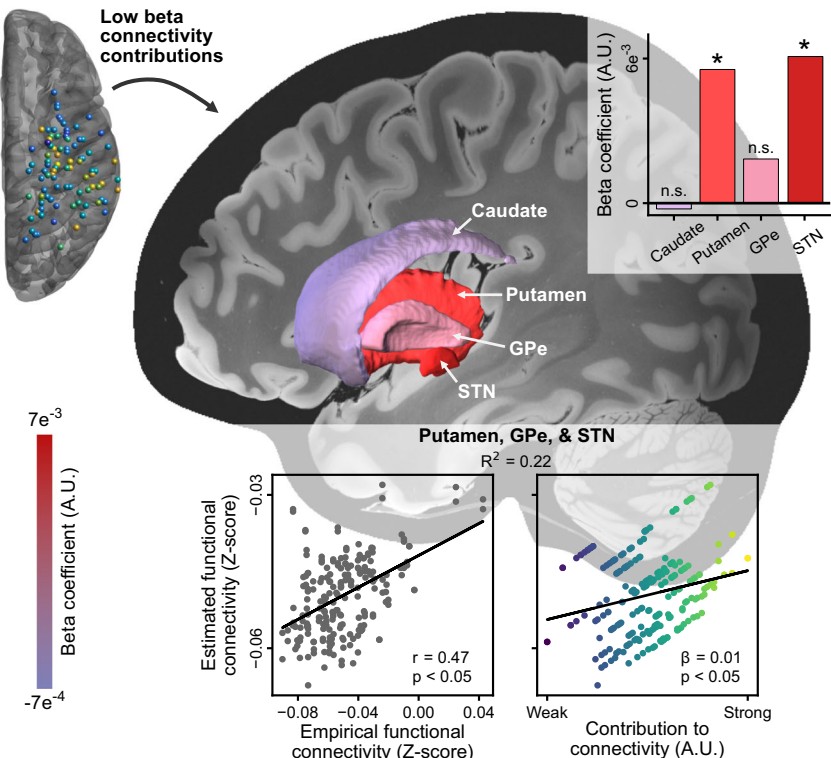

**b** **Low beta coupling reflects indirect pathway connectivity**

This has important implications for the development of next-generation neurotechnology, which may replace pathological cortex – basal ganglia communication with brain circuit neuroprosthetics to treat dopaminergic disorders.

The field of neuromodulation stands at the brink of a transformative time, as advancements in our understanding of brain networks, their implication in brain disorders, and their modulation with medication and neurostimulation are described with ever-increasing detail. While the neurosurgical planning for DBS still mostly revolves around the optimal local anatomical sweet spot, it is now clear that the underlying brain network – not the individual target structure – represents the cohesive entity that must be modulated for therapeutic success[4,15,43–45]. Advances in MRI connectomics have repeatedly shown that the effects from variance in DBS electrode localisation can be attributed to the differential modulation of structural and functional brain networks, most recently across therapeutic targets, symptoms, and diseases[46,47]. Furthermore, sensing-enabled brain implants for adaptive DBS now allow for the additional spatio-spectral dimension of brain circuit interrogation for real-time adjustments of therapy[48–54]. Given the critical role of the cortex as a source of input to the basal

**Fig. 4 | Cortico-subthalamic structural and functional connectivity.**
**a** Contribution of ECoG and STN-LFP contacts to high beta cortico-subthalamic connectivity correlates significantly with the number of hyperdirect pathway fibres connecting these regions. Hyperdirect pathway fibres are coloured according to the grand average contributions of ECoG and STN-LFP contacts to high beta cortico-subthalamic connectivity (obtained from maximised imaginary coherency). The inset shows the relationship between the estimated number of hyperdirect pathway fibres from the linear mixed effects model versus the empirical number of fibres (top; correlation coefficient, r, and p-value represent the quality of the linear mixed effects model as determined using the Pearson correlation; p < 0.00001) and the contributions to high beta oscillatory connectivity (bottom; beta coefficient and p-value reflect that reported in text and Supplementary Table 3; p < 0.00001). The conditional R² value representing the quality of the model is also shown.
**b** Contribution of ECoG contacts to low beta cortico-subthalamic connectivity (obtained from maximised imaginary coherency) correlates significantly with the

functional MRI connectivity of ECoG contact locations to indirect pathway nuclei. The upper inset shows the beta coefficients of the models for the caudate (p = 0.96616), putamen (p = 0.02526), GPe (p = 0.43925), and STN (p = 0.00699). Lower inset shows the relationship between the estimated functional connectivity from cortex to putamen, GPe, and STN versus the empirical functional connectivity to these regions (left; correlation coefficient, r, and p-value represent the quality of the linear mixed effects model as determined using the Pearson correlation; p < 0.00001), as well as the contributions of ECoG contacts to low beta connectivity (right; beta coefficient and p-value reflect that reported in text and Supplementary Table 7; p = 0.02313). The conditional R² value representing the quality of the model is also shown. * p < 0.05. All panels: statistical results obtained from linear mixed effects models and two-sided Pearson correlation tests. Source data are provided as a Source Data file. GPe external segment of the globus pallidus, n.s. not significant, STN subthalamic nucleus.

ganglia, understanding the role of cortex – basal ganglia communication in PD is essential. In particular, the monosynaptic cortico-subthalamic hyperdirect pathway has been posited as fundamental to the therapeutic effect of DBS[14,55]. However, significant knowledge gaps remain, as the direction of stimulation effect (i.e., activation vs. suppression) remains a matter of continuous debate, and the relationship to therapeutic symptom alleviation with dopaminergic agents – the first-line therapy for PD – has not been investigated with robust methods. The present study sheds light on multiple aspects of the shared therapeutic effects of dopamine and STN-DBS, most prominently demonstrating that both treatments can suppress oscillatory hyperdirect pathway communication in high beta frequencies. This corroborates the most recent human and non-human primate studies, which suggest that monosynaptic hyperdirect pathway input is not only relevant for therapeutic symptom alleviation but is also mechanistically implicated in the development of pathological circuit synchrony in PD[17,19,34,56,57]. Indeed, our study confirms multiple necessary considerations for this theory with the demonstration that: a) cortex drives subthalamic activity[19,34]; b) cortex – STN oscillatory connectivity is spatio-spectrally linked to hyperdirect and indirect pathway communication[19,34], as revealed through whole-brain MRI connectomics; and c) both dopamine and DBS suppress 1–9 ms monosynaptic input to the STN[13,37,39,40]. The mechanisms by which dopaminergic agents act on the hyperdirect pathway remain speculative. One plausible explanation is that dopamine reduces cortex – STN beta coupling by modulating striatal medium spiny neuron activity, leading a modulation of pallido-subthalamic communication. In the absence of dopamine, reciprocal GPe – STN network activity drives pathological beta activity[17,58]. While dopamine, unlike STN-DBS, does not block glutamatergic cortical input to the STN, it may lower the susceptibility to beta resonance from the hyperdirect pathway. Alternatively, or in addition, local dopamine receptor activation in the STN or cortex may also influence hyperdirect pathway communication directly. In addition, our findings further corroborate previous studies suggesting that cortico-subthalamic interactions via the indirect pathway are mediated predominantly by low beta coupling, while high beta coupling reflects hyperdirect communication[19,34]. This also aligns with a previous ECoG-LFP study reporting predominantly low beta coupling related to PD pathophysiology between cortex and internal pallidum, where there is no hyperdirect pathway connectivity[59]. Notably, while pallidal DBS is similarly effective to STN-DBS, pallidal stimulation does not directly affect hyperdirect afferents to the STN. We speculate that the more immediate modulation of the hyperdirect pathway with STN-DBS may explain the clinical observation that STN-DBS allows for a stronger reduction in dopaminergic medication when compared to pallidal DBS in PD patients. Nevertheless, even pallidal DBS likely induces modulation of hyperdirect synchrony through orthodromic changes in thalamo-cortical activation. Finally, our results provide an authoritative account on the dopaminergic

modulation of cortical activity and cortico-subthalamic hyperdirect pathway communication based on a fully invasive electrophysiological approach, one that can serve as a reference for non-invasive neurophysiology where previous findings have been difficult to relate to invasive animal literature[16,19,20,60–67]. Accordingly, our study highlights the shared neural network mechanisms of dopamine and DBS and may inspire neural circuit interventions that specifically target dynamic hyperdirect pathway communication with sensing-enabled neurotechnology in the spatio-spectral domain.

Indeed, further support for a dopaminergic modulation of cortico-subthalamic communication comes from our observation that dopamine selectively suppresses the degree of information flow from cortex to STN in the 1–9 ms window associated with hyperdirect pathway activity[13,37,40]. In line with previous reports, we find no dopamine-specific effect on the net dominance of the cortex in communication with STN, which may suggest that dopaminergic medication simultaneously reduces hyperdirect pathway signal transmission and the reciprocal communication with the cortex. This deviates from the specific DBS-induced reduction of cortical drive to the STN from ECoG contacts over the primary motor cortex, which suggests a more selective effect of stimulation on suppressing hyperdirect pathway activity whilst minimally altering the reciprocal feedback of information to the cortex. We speculate that dopaminergic medication allows cortex – basal ganglia circuits to entertain physiologically healthy communication, which is more rigorously jammed with high-frequency DBS, as discussed in the context of the informational lesion theory of DBS mechanisms[68]. Thus, the distinct effects of medication and stimulation on cortico-subthalamic communication may reflect the localised effects of DBS on subthalamic activity compared to the more widespread changes in basal ganglia circuitry resulting from dopaminergic action. Similar to dopaminergic medication, we also find that stimulation suppresses information flow in the 1–9 ms time window, indicative of reduced monosynaptic cortex – STN communication. Together, these findings clearly demonstrate the suppression of hyperdirect pathway communication as a shared therapeutic effect of dopaminergic medication and DBS. Although the hyperdirect pathway and high beta activity has been the focus of much work into cortico-subthalamic communication in PD, it is important to remember that these are only select aspects of the cortex – basal ganglia network. In the broadband 3–100 Hz range, although the bispectrum revealed a high degree of communication from cortex to STN in the time window congruent with monosynaptic signal transmission, the strongest time delay estimates occurred in a temporal range consistent with polysynaptic communication (~ 30 ms)[37–39], even for motor cortex. One obvious source of polysynaptic information flow is indirect pathway signalling mediated by low beta activity, but other oscillations are likely also involved, such as theta (4–8 Hz) activity – the cortico-subthalamic coupling of which was recently highlighted for its relevance in the Parkinsonian state[69] – and gamma (60–90 Hz) activity.

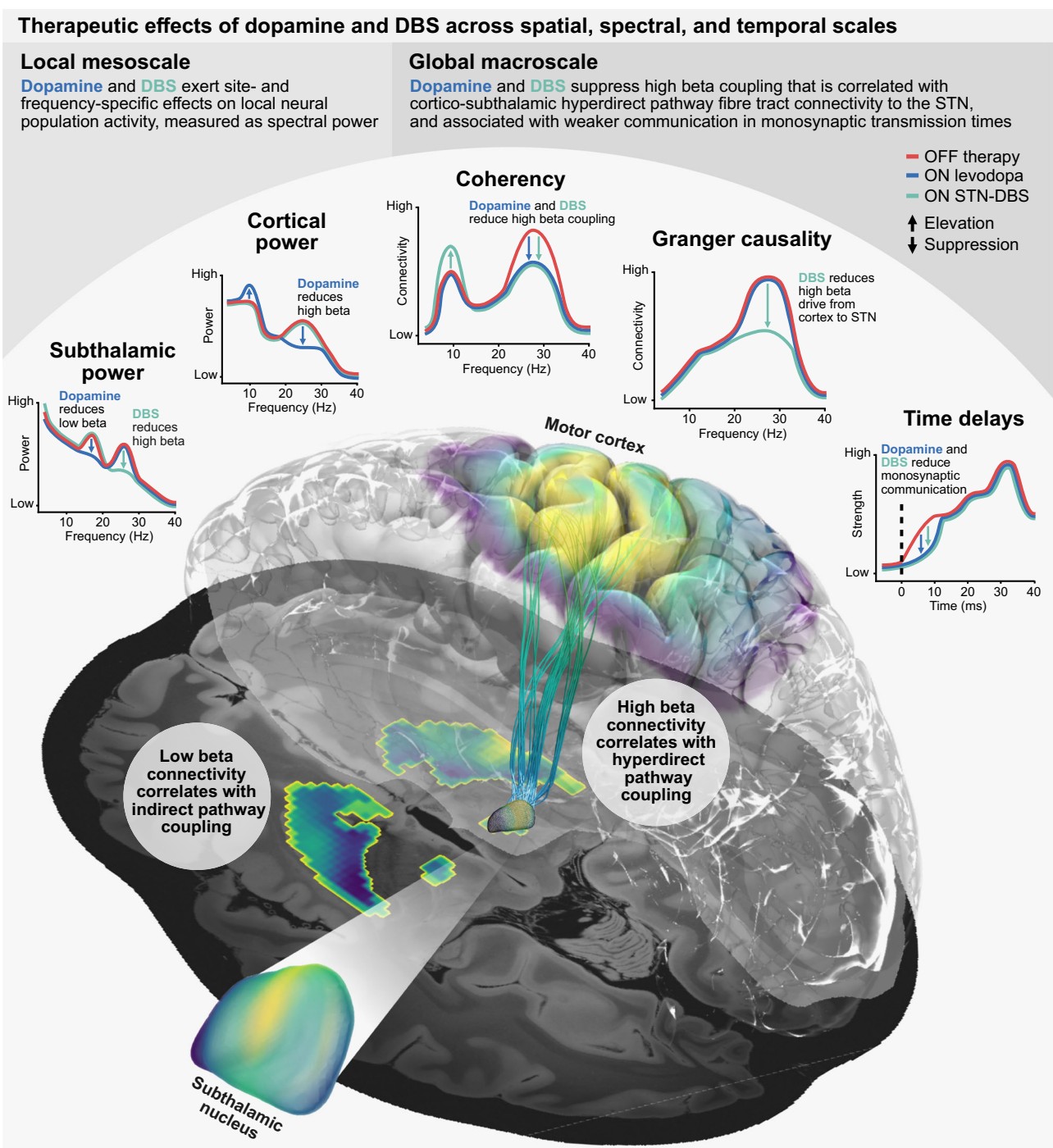

**Fig. 5 | Schematic summary of therapeutic effects.** At the mesoscale, dopamine and DBS exert site- and frequency-specific effects on cortical and subthalamic power, with dopamine increasing mu rhythm/alpha power and reducing high beta power in the cortex, whilst in the STN dopamine reduces low beta power and DBS predominantly reduces high beta power. See Table 1 for the localisations of power. At the macroscale, dopamine and DBS both reduce cortex – STN high beta coupling, activity associated with the monosynaptic hyperdirect pathway, in contrast to low beta coupling associated with the polysynaptic indirect pathway. In addition, there is a weakening of information flow from the cortex to STN with both dopamine and DBS in time ranges congruent with hyperdirect pathway communication. However, unique effects are also present, with DBS increasing cortex – STN mu rhythm/alpha coupling, and a directed suppression of cortex to STN high beta communication. See Table 2 for the localisations of coupling. DBS deep brain stimulation, STN subthalamic nucleus.

Ultimately, these results highlight the relevance of both indirect and hyperdirect pathway communication for cortico-subthalamic information flow.

In addition to considering the network effects of therapy, it is useful to assess the effects on local synchronous activity. Modulatory effects of medication and stimulation on STN-LFP activity have been frequently described in previous literature. In keeping with these prior studies, we find a reduction in subthalamic beta power with dopaminergic medication and STN-DBS[5,8,9,70], which other work has linked to clinical improvements in motor performance[7,9,53,70–72]. However, studies have also sought to characterise the effects of therapy on the level of the cortex. Here, our findings deviate from previous results, identifying a reduction in cortical high beta power localised to the primary motor cortex. Although previously undescribed in human PD patients,

our finding corresponds with observed effects in both rodent and non-human primate studies[16,18,63,67,73]. Furthermore, we find that dopamine increases mu rhythm/alpha activity localised to the sensory and parietal cortices, which may relate to increased low-frequency activity associated with prokinetic states[69,74,75]. This suggests that, in addition to the modulatory effects on STN power, the therapeutic effects of the dopaminergic medication may involve a reduction of anti-kinetic beta synchrony and an elevation of prokinetic mu rhythm/alpha synchrony on the level of the cortex. In line with several previous ECoG studies[33,76,77], cortical synchrony during DBS revealed no reduction in high beta power, suggesting site-specific mesoscale effects of dopamine and stimulation. We note that one previous study investigating the effects of STN-DBS using ECoG recordings described an ameliorating effect of stimulation on motor cortex beta power in two patients[78], however this discrepancy may be explained by the use of higher stimulation energies (185 Hz frequency and 90 μs pulse width). Importantly, while stimulation did not alter cortical oscillatory power in most previous reports, it is worth mentioning other work highlighting the suppressive effects of STN-DBS on cortical non-sinusoidal waveform shape and beta-gamma phase-amplitude coupling, which were reported to correlate with the severity of motor impairment in PD patients[76,77]. Altogether, our findings reinforce the importance of considering mesoscale activity in the wider cortex – basal ganglia network for PD pathology.

While previous studies have sought to characterise the effects of dopaminergic medication and DBS on local cortical activity and cortico-subthalamic communication, the present study has identified several previously undocumented phenomena. For example, although STN-DBS was previously shown to suppress cortico-subthalamic communication – attributed to reduced activity of the hyperdirect pathway[34] – no suppressive effects of medication on cortico-subthalamic coupling have been described[19,20,79], nor the suppression of cortical high beta power with dopamine. A trivial explanation for the latter effect could be that ON levodopa recordings in our cohort were contaminated by overt or covert motor output – such as through a more dyskinetic state – associated with reduced beta power. To address this potential problem, we have carefully excluded periods with movements from the data and attended all recordings to ensure that patients were truly at rest. Nevertheless, we acknowledge that this does not rule out potential effects from changes in muscle tone, which could either be a cause or consequence of changes in beta activity. However, the previously undocumented effects may also be explained by the predominant use in prior studies of non-invasive magnetoencephalography (MEG) recordings of the cortex, univariate measures of power, and bivariate measures of connectivity. First, non-invasive MEG recordings of cortical activity have different signal characteristics compared to invasive ECoG recordings, potentially mixing divergent modulation patterns from distinct and more distant cortical sources. Furthermore, whole-brain statistics on beamforming-based imaging of MEG recordings may have been more conservative and strongly driven by peaks in the supplementary motor area, a region not covered by ECoG electrodes in our study. Second, these earlier findings may have been compounded by the reliance on univariate measures of power and bivariate measures of connectivity, which could in the future, be augmented by advanced multivariate metrics as shown here. The typical procedure for aggregating uni/bivariate information is to average over channels. Crucially, if the oscillatory sources are present in only a specific set of channels, averaging naturally diminishes the signal-to-noise ratio of the final results. In contrast, multivariate methods go beyond the channel level to capture the strongest sources of activity in the component space, alongside which the spatial maps of these components can be identified. Not only do multivariate approaches enhance the signal-to-noise ratio and interpretability of results, but their utility for the identification of optimal stimulation targets is highlighted here, with

multivariate power analysis identifying the strongest source of low beta activity to be within 0.5 mm of the STN-DBS sweet spot[30]. Altogether, it is reasonable to suggest that invasive cortical recordings, in combination with advanced signal processing methods, provided an additional layer of detail for the characterisation of therapeutic meso- and macro-scale effects of dopamine and DBS.

Before we continue with an outlook, it is important to consider the limitations of this study. First, all recordings were performed a few days following the implantation of the DBS electrodes in the STN. It is well-documented that electrode implantation can introduce a 'stun effect' – a transient suppression of Parkinsonian symptoms in the absence of medication or stimulation – potentially reflecting a microlesion of the STN[80,81]. Accordingly, recorded activity in the STN and other components of the cortex – basal ganglia network may not reflect the typical Parkinsonian state. However, STN-LFP recordings months and years after implantation have repeatedly revealed similar profiles of activity to those seen in the immediate postoperative state[49,82–85]. Second, in contrast to other studies, an objective measure of symptom alleviation during DBS was not performed, obviating a more direct comparison between DBS parameters and clinical effects[53]. Instead, a preliminary clinical review of stimulation parameters was conducted to determine those used in the ON STN-DBS recordings. The observed stimulation effects may, therefore, not fully represent those following a more extensive parameter review in the chronic DBS state, although similar settings were used in a 3-month follow-up clinical review (see Supplementary Tables 1 and 2). Third, it is possible that the distinct mesoscale therapeutic effects reflect differences between subjects in the larger medication and smaller stimulation groups. However, a control analysis using only subjects that had undergone both medication and stimulation conditions reproduced the distinct effects of dopamine and DBS on low vs. high beta activity. It is further noteworthy that our study reports a frequency-specific high beta power modulation with STN-DBS. Whilst most previous studies have performed statistics solely on a broad 13–30 Hz range which did not allow for a more precise spectral distinction, some studies show significant modulations that extend into the low beta band[9,53]. Nevertheless, even these previous studies show the strongest modulation to occur at or above 20 Hz[53] and feature stronger residual low beta ON STN-DBS when compared to ON levodopa[9]. In addition, the unilateral ECoG strip implantation prohibited an exploration of interhemispheric circuit communication, a recent topic of interest for chronic biomarker studies[49]. Similarly, our exploration of network synchrony remains non-exhaustive, as recent work has highlighted burst dynamics, phase-amplitude coupling, and waveform shape as metrics that provide additional insights into the therapeutic effects of dopamine and DBS. Regarding the time delay analysis, we note that we were unable to adequately estimate time delays in 1 subject due to excess noise in the recordings, as determined by a bootstrap-based confidence criterion. Although the bispectrum shows a greater – if not complete – resilience to Gaussian noise compared to traditional cross-correlation analysis[27], it can nonetheless be contaminated by non-Gaussian noise, with the estimation of time delays requiring a greater signal-to-noise ratio compared to methods such as spectral power and coupling. Overall, we were still able to determine time delays for 20 subjects. In addition, we limited our analyses to the broadband 3–100 Hz range, which prevented identifying time delays associated with the low and high beta bands specifically. This decision was informed by simulations which showed time delay estimates from individual frequency bands to have limited accuracy and ability to meet the confidence criterion. More generally, we have restricted our spectral analyses to lower frequency bands up to 40 Hz. Recent studies have shown that DBS and dopamine can have important effects on gamma-band oscillations that can be entrained to half the stimulation frequency, often occurring in temporal relation to the clinical ON state or dyskinesia[51,86]. The relationship between pathway-specific

therapeutic effects and these high-frequency patterns warrants further investigation. As a final noteworthy limitation, we would like to highlight that the functional and structural connectivity analyses relied on atlas data and group connectomes, as patient-specific fMRI and dMRI scans were not taken. It is thus unclear to what degree the connectomic findings relate to individual anatomy. On the other hand, this has the advantage that robust connectomes derived from large cohorts have been validated multiple times[19,30,31], with superior image quality when compared to individual clinical MRI scans.

Our findings have important implications for adaptive DBS and brain circuit prosthetics. Adaptive DBS aims to use recordings of neural activity to tailor stimulation to the current brain state, with the goal of increasing treatment efficacy and reducing side effects. Existing approaches for PD have largely focused on subthalamic beta-band activity-based control policies[87–89], with clinical trials ongoing (clinicaltrials.gov: NCT04681534; NCT04547712). Whilst promising, such single biomarker paradigms are inherently limited in the amount of information that can be captured about the complex Parkinsonian state[90–93], and sensing parameters will have strong effects on the consistency of stimulation delivery[94]. Crucially, tailoring stimulation to individual symptoms and behaviours may prove critical to advancing treatment[95,96]. Recent work has highlighted the utility of invasive cortical recordings for characterising patient behaviour, with the ability to accurately decode ongoing and predict upcoming movements from ECoG recordings in PD patients, including through connectivity-based biomarkers[49,69,95]. Our findings build on this picture further, highlighting pathological cortical activity and communication between the cortex and STN as additional biomarkers of the Parkinsonian state, available through the incorporation of invasive cortical recordings. Altogether, multisite readouts of neural activity offer significant innovations for next-generation adaptive stimulation paradigms[21,22] and could pave the way to delivering stimulation precisely when and how it is needed. In the future, we envision a neuroprosthetics approach that can deliver DBS to transiently shut down hyperdirect pathway activation and mimic temporally precise dopaminergic transients for the support of neural reinforcement[97]. If successful, DBS could be used to restore physiological circuit communication instead of merely suppressing pathological activity, affording the hope to help millions of patients suffering from dopaminergic disorders.

In summary, our study reveals shared pathway-specific effects of dopamine and STN-DBS on cortex – basal communication. Both medication and stimulation diminished connectivity in the high beta band, activity attributed to the hyperdirect pathway. In addition, whilst the cortex drove communication with STN, a distinct effect of stimulation on this directionality was identified. These findings provide support for the hypothesis that communication between the cortex and STN is pathologically increased in the Parkinsonian state, proposed as a key factor in the origin of pathological subthalamic beta activity, a hallmark of Parkinson's disease. Ultimately, our study provides mechanistic insights into the role of therapeutic interventions on the wider cortex – basal ganglia circuitry, offering further support for a clinical paradigm shift towards targeting dynamic brain networks with neurotechnology in Parkinson's disease.

## Methods
### Participants
21 patients (17 male) diagnosed with idiopathic PD of primary akinetic-rigid motor phenotype with clinical indication for DBS were enroled at the Department of Neurosurgery at the Charité – Universitätsmedizin Berlin (Supplementary Table 1). DBS leads were bilaterally implanted into the STN. A subdural ECoG electrode strip was implanted unilaterally, targeting the hand knob area of the primary motor cortex for research purposes. Sex and gender were not considered in the study design due to the small sample size.

### Ethics declaration
The research and brain signal recordings presented in this manuscript were performed according to the standards set by the declaration of Helsinki and after approval by the ethics committee at Charité Universitätsmedizin Berlin (EA2/129/17). All patients provided informed consent to participate in the research. The data was collected, stored, and processed in compliance with the General Data Protection Regulation of the European Union.

### DBS and ECoG placement
DBS implantation followed a two-step approach. In the first surgery, DBS leads were placed stereotactically after co-registering preoperative MRI and CT images. A single ECoG electrode strip was placed subdurally onto one hemisphere after minimal enlargement (~2 mm) of the frontal burr hole. The ECoG strip was aimed posteriorly toward the hand knob region of the motor cortex. Defining the hand-knob region to be a single MNI coordinate (x: 36, y: −19, z: 73), the closest ECoG contact per subject had an average distance of $12.3 \pm 1.6$ mm (mean ± SEM) to this coordinate. The ECoG strip was placed ipsilaterally to the implantable pulse generator (right = 16, left = 5). All electrodes were then externalised through the burr holes via dedicated externalisation cables. Patients remained on the ward for a duration of 4–7 days until the second surgery. Between surgeries, electrophysiological recordings were performed. In the second intervention, externalisation cables of the DBS leads were replaced with permanent cables that were tunnelled subcutaneously and connected to a subclavicular implantable pulse generator. ECoG electrodes were removed via the burr hole during the second surgery.

### Anatomical localisation of electrodes
DBS and ECoG electrodes were localised using standard settings in Lead-DBS[98]. Preoperative MRI and postoperative CT images were co-registered, corrected for brain shift, and normalised to MNI space (Montreal Neurological Institute; MNI 2009b NLIN ASYM atlas). ECoG contact artefacts were marked manually, and MNI coordinates were extracted. DBS electrodes were reconstructed using the PaCER[99] algorithm and manually refined, with DBS electrode rotation being manually corrected using artefacts of electrode orientation markers with the DiODe v2[100] algorithm. In brief, the postoperative CT slice showing the most visible marker artefact in the electrode was first selected, and the marker of the reconstructed electrode model was then rotated to align with this. MNI coordinates were then extracted for the DBS lead contacts. See Fig. 2a for reconstructed electrode localisations. Individual ECoG contacts were assigned to one of either parietal, sensory, motor, or prefrontal cortex based on proximity to these regions using the AAL3 parcellation[101]. Bipolar ECoG channels were similarly assigned according to their constituent individual electrodes.

### Experimental paradigm
Study participants were asked to rest comfortably in an armchair and asked not to speak or move for the duration of the recording. These rest sessions were done either under the patient's current clinical intake of dopaminergic medication (ON levodopa; see Supplementary Table 1 for details of levodopa-equivalent daily doses at time of recording) or after at least 12 h of withdrawal of all dopaminergic medication (OFF therapy). 18 subjects took part in the rest sessions OFF therapy and ON levodopa. In addition, 12 subjects took part in rest sessions after the withdrawal of medication but during the application of high-frequency DBS to the STN (ON STN-DBS). Contacts and stimulation parameters used during recording were determined in a monopolar clinical review. Clinically effective contacts were chosen while avoiding stimulation-induced side effects. DBS was applied at 130 Hz with 60 μs pulse width, with a mean amplitude per subject of $2.4 \pm 0.1$ mA (see Supplementary Table 2 for details). 10 participants

received bilateral monopolar stimulation. Due to a low side effect threshold, 2 participants received unilateral monopolar stimulation in the hemisphere ipsilateral to the ECoG strip. UPDRS-III scores at 12 months post-implantation for subjects recorded both OFF therapy and ON STN-DBS were 47.0/24.0 ± 3.8/4.4, respectively (n = 5, unavailable in n = 7; see Supplementary Table 1 for details).

### Electrophysiology recordings

ECoG and STN-LFP recordings were conducted in the days between the first and second surgical intervention. Subdural ECoG strips had 6 contacts facing the cortex. STN-LFP were recorded from 3 DBS lead models implanted with either 4 (n = 1), 8 (n = 19), or 16 (n = 1) contacts (Supplementary Table 2). We considered only STN-LFP recordings from the hemisphere ipsilateral to the ECoG strip. All signals were amplified and digitised with a Saga64 + (Twente Medical Systems International; 4000 Hz sampling rate) device. ECoG and STN-LFP signals were hardware-referenced to the lowermost contact of the DBS electrode (contralateral to ECoG hemisphere n = 17; ipsilateral n = 4). In a small number of cases where excessive noise was visible before recording, a different STN contact was used as the hardware reference. Data was saved to disk for offline processing and converted to iEEG-BIDS[102] format. Offline processing was performed with custom scripts in Python v3.11 using MNE-Python v1.6[103], MNE-Connectivity v0.5 (mne.tools/mne-connectivity/), MNE-BIDS v0.13[104], PyPARRM v1.1[105], PyBispectra v1.1[106], PTE Stats v0.2 (github.com/richardkoehler/pte-stats), statsmodels v0.14[107], NumPy v1.24[108], Pandas v2.0[109], and SciPy v1.1[110], as well as custom scripts in MATLAB r2022b using FieldTrip r20221223[111], SPM12 r7771 (www.fil.ion.ucl.ac.uk/spm/software/spm12/), and Lead-DBS v2.6[98].

### Electrophysiological data preprocessing

ECoG and STN-LFP data was notch filtered to remove line noise (at 50 Hz and all higher harmonics), bandpass filtered at 3–150 Hz, divided into epochs with a 2 s duration, and resampled at 500 Hz (1000 Hz for the time delay analysis). Epochs were visually inspected, with periods containing high amplitude artefacts (e.g., due to cable movements) and muscle movement artefacts being marked and excluded from the analyses.

To avoid biases in oscillatory connectivity analyses arising from differences in data lengths, total recording durations were standardised across patients by partitioning the epoched data into 60 s segments, giving 30 epochs per segment[112]. 30 epochs from the entire recording duration were sampled with replacement using a uniform distribution 200 times, producing 200 60 s long segments which together covered the entire recording length.

As an additional sanity check for the quality of time delay analysis results, the same epoch sampling procedure was employed, with the exception that epochs were sampled 400 times. Time delay estimates were computed for each of these 400 segments, and a confidence interval-based criterium used to exclude excessively noisy data from the time delay results (see Time delay analysis).

In the ON STN-DBS recordings, DBS artefacts were removed from the ECoG and STN-LFP data using the period-based artefact reconstruction and removal method[113]. To ensure comparability of information content between OFF therapy and ON STN-DBS conditions, the data of STN-LFP contacts designated as stimulation contacts in the ON STN-DBS recording were also excluded from the corresponding OFF therapy recording. The data of these contacts was retained for the comparisons of OFF therapy and ON levodopa conditions.

### Power spectral analysis

Data was referenced in a bipolar montage for all adjacent ECoG and STN-LFP contacts, removing contaminating information from the original subthalamic reference. Power was calculated using multitapers (5 Hz bandwidth), normalised to percentage total power[114] (from

5–60 Hz with a line noise exclusion window at 45–55 Hz), and averaged across epochs to give a single spectrum per channel, per recording. An upper bound of 60 Hz was chosen for the normalisation to prevent possible contamination from residual stimulation artefacts following the artefact removal procedure. Given that there is a risk of changes in any one frequency band affecting power in other frequencies for small normalisation windows, a broader 5–95 Hz normalisation range was additionally performed, however this did not meaningfully alter the interpretation of the results.

An additional multivariate analysis of power spectral information was performed using spatio-spectral decomposition[28]. Here, frequency band-resolved spatial filters were applied to the data such that the signal-to-noise ratio of this frequency band was maximised, returning a single power spectral component for each frequency band. This decomposition was performed on the ECoG and STN-LFP data separately in the following frequency bands, with flanking noise frequencies of ±1 Hz: low beta (12–20 Hz); and high beta (20–30 Hz). Prior to performing spatio-spectral decomposition, the number of components in the ECoG and STN-LFP data was normalised across recordings regardless of the number of contacts by extracting the principal components of the data using singular value decomposition and taking the first n components, where n was the minimum number of components across the cohort[112] (bipolar ECoG = 4; bipolar STN-LFP = 3). After applying the spatial filters to the data and bandpass filtering in the corresponding frequency bands to isolate the optimised activity, the power spectral densities of these strongest power components were computed using multitapers as above. Finally, spatial maps of the contribution of cortex and STN to the strongest power components were extracted from the spatial filters[28,35], the absolute values taken, and Z-scored.

### Oscillatory connectivity analysis

Cross-spectral densities of unipolar ECoG and STN-LFP data were calculated from each segment using multitapers (5 Hz bandwidth). For the multivariate connectivity analyses, we normalised the data by extracting the principal components of the ECoG and STN-LFP data and taking the first n components (unipolar ECoG = 5; unipolar STN-LFP = 3).

Undirected connectivity between cortex and STN was quantified using the imaginary part of coherency – a bivariate measure of correlation in the frequency domain immune to spurious connectivity estimates from zero time-lag interactions[24]. Since we were interested in the overall degree of connectivity regardless of the phase angle differences encoded in coherency's sign, we took the absolute values of connectivity. In addition, we computed the maximised imaginary part of coherency – a multivariate extension of the method[23]. Here, frequency-resolved spatial filters were applied to the data such that connectivity between the seed (ECoG) and target (STN-LFP) data was maximised, returning a single spectrum for cortico-subthalamic connectivity. As with spatio-spectral decomposition, spatial maps of the contribution of the cortex and STN to the maximised connectivity component were extracted from the spatial filters[23,35], the absolute values taken, and Z-scored.

Directed connectivity was quantified using a multivariate form of time-reversed Granger causality[25,26] with a vector autoregressive model order of 60 to avoid excessive smoothing of Granger scores across frequencies. Following the definition of time-reversed Granger causality[26], Granger scores with ECoG data as seeds and STN-LFP data as targets were taken, and Granger scores with STN-LFP data as seeds and ECoG data as targets subtracted from this to obtain net Granger scores. Subsequently, net Granger scores from time-reversed data (through the transposition of the autocovariance sequence[115]) were obtained and subtracted from the original net Granger scores, producing the time-reversed Granger scores. These final scores thus represent the net drive of information flow between cortex and STN,

corrected for spurious connectivity estimates arising from weak data asymmetries.

For all connectivity methods, results were averaged across the 200 bootstrapped segments to give a single set of connectivity values per recording. Importantly, it was unnecessary to bipolar reference the data to remove contamination from the original subthalamic reference as: 1) the imaginary part of coherency does not capture zero time-lag interactions[24], and 2) net Granger scores eliminate bidirectionally uniform interactions[26].

### Time delay analysis

Fourier coefficients of the bipolar ECoG and STN-LFP channels were computed using a hamming window and 4001 points. This ensured time delay estimates were returned at intervals of 1 ms for the full length of each 2 s long, 1000 Hz sampling rate epoch, in the positive and negative directions. Signal interactions were characterised from the Fourier coefficients in the broadband 3–100 Hz range using the bispectrum – the Fourier transform of the third-order moment. Time delay estimates were subsequently computed through the extraction of phase spectrum periodicity and monotony followed by an inverse Fourier transform, as in method I of Nikias and Pan[27]. ECoG channels were defined as seeds, and STN-LFP channels as targets. The time at which the delay estimate is strongest is defined as tau, the time delay of this connection. Using the 400 bootstrapped segments, 80% confidence intervals were computed for each connection from the segment taus. Connections for which $t = 0$ ms falls within the confidence interval reflect an uncertain time delay estimate that is highly sensitive to the sampled epochs, an indication of excessive noise in the data. These uncertain connections were excluded from the time delay results, acting as an additional statistical barrier to ensure the quality of the estimates. 1 subject for the OFF therapy – ON levodopa comparison had no connections meeting the confidence criteria and was thus excluded from the analysis. The remaining time delays were averaged across the 400 bootstrapped segments. In addition to analysing tau (the global maximum across the estimates for all times), peaks (local maxima) in discrete time windows were also analysed. Treating the estimated strength at $t = 0$ ms as a baseline, local maxima above this threshold and at least 5 ms apart from one another were identified and grouped into 10 ms bins (apart from the first bin of 1–9 ms where $t = 0$ ms was excluded). Peak counts were normalised to percentages to account for variation in the number of cortex – STN connections across subjects and cortical regions. When reporting the tau results, given our focus on the directed cortex to STN communication, we enforced the additional criterion that only connections with tau > 0 ms were considered.

### dMRI-based structural connectivity analysis

Cortico-subthalamic hyperdirect axonal pathway fibres of the human holographic subthalamic atlas were used to determine structural connectivity profiles from individual electrode locations in MNI space[29]. Hyperdirect fibres connecting the cortical and subthalamic recording regions were identified as those fibres in a given radius of each ECoG (5 mm) and STN-LFP (3 mm) contact. The 5 mm radius for ECoG contacts was chosen to sample from a wide area whilst preventing overlap between contacts. A smaller radius of 3 mm did not meaningfully alter the interpretation of the results. For the OFF therapy and ON levodopa recordings, the number of fibres connecting a given ECoG and STN-LFP contact were then compared to the average low (12–20 Hz) and high (20–30 Hz) beta band spatial contribution maps extracted from maximised imaginary coherency (see Oscillatory connectivity analysis) averaged over the respective ECoG and STN-LFP contacts. This enabled us to determine the spectral associations of the hyperdirect pathway. Given the similarity of spatial contribution maps in the levodopa and STN-DBS groups (Supplementary Fig. 7), only the spatial contribution maps from the larger levodopa group were

considered. Including the STN-DBS maps did not meaningfully alter the interpretation of the results. Atlas-based connectomes were used, as patient-specific dMRI scans were not performed.

### fMRI-based functional connectivity analysis

An openly available Parkinson's disease fMRI group connectome (www.lead-dbs.org; previously used in Horn et al.[30,31]) derived from the Parkinson's progression markers initiative (PPMI) database[41] was used to investigate the relationship of invasive oscillatory connectivity from ECoG-LFP recordings with fMRI resting-state connectivity on the whole-brain level. All scanning parameters are published on the website (www.ppmi-info.org). Whole-brain functional connectivity maps were generated, using the MNI coordinates for each ECoG contact as a seed with a radius of 5 mm. As for the structural connectivity analysis, a smaller radius of 3 mm did not meaningfully alter the interpretation of the results. For each map, the average connectivity values with indirect pathway nuclei (caudate, putamen, external segment of the globus pallidus, and STN) as defined in the Atlas of the Basal Ganglia and Thalamus[42] were extracted. For the OFF therapy and ON levodopa recordings, these bivariate connectivity values were compared to the average low (12–20 Hz) and high (20–30 Hz) beta band spatial contribution maps extracted from maximised imaginary coherence (see Oscillatory connectivity analysis) for the respective ECoG contacts. This enabled us to determine the spectral associations of the indirect pathway. Given the similarity of spatial contribution maps in the levodopa and STN-DBS groups (Supplementary Fig. 7), only the spatial contribution maps from the larger levodopa group were considered. Including the STN-DBS maps did not meaningfully alter the interpretation of the results. Normative connectomes derived from a large population were used, as patient-specific fMRI scans were not performed.

### Statistical analysis

Statistical analysis of power, oscillatory connectivity, and time delay results was performed with PTE Stats (github.com/richardkoehler/pte-stats) using non-parametric permutation tests with 100,000 permutations at an alpha level of 0.05, with cluster correction for significant $p$-values to control for multiple comparisons where appropriate[32]. Permutations involved randomly assigning conditions for each of the subject pairs, using the difference in means as the test statistic for comparison between the original and permuted data.

Linear mixed effects modelling was performed with statsmodels[107] for analysis of the relationships between dMRI-/fMRI-based connectivity and oscillatory connectivity measures with the following model construction: "dMRI/fMRI connectivity ~ Spatial patterns + Medication + (1 | Subject)". Values from OFF therapy and ON levodopa recordings were considered in the same model, with medication state used as a fixed condition. Subjects were treated as a random effect.

### Reporting summary

Further information on research design is available in the Nature Portfolio Reporting Summary linked to this article.

## Data availability

Data can be made available conditionally to data sharing agreements in accordance with data privacy statements signed by the patients within the legal framework of the General Data Protection Regulation of the European Union. Requests should be directed to the lead contact, Wolf-Julian Neumann (julian.neumann@charite.de), or the Open Data officer (opendata-neuromodulation@charite.de). Responses to such requests will be made within 1 week. The data may only be used for non-commercial purposes and only to those persons with a valid ethics proposal for the use of the data and an approved Data Privacy Impact Assessment at the Charité – Universitätsmedizin Berlin (2–7 weeks

processing time). The data will be made available for 48 months, after which re-application for access is possible. Source data for figure graphs are provided in this paper. Source data are provided in this paper.

## Code availability

All code is made publicly available on GitHub (github.com/neuromodulation/manuscript-binns_cortex_stn_comm) and archived on Zenodo (https://doi.org/10.5281/zenodo.10974655).

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

## Acknowledgements

Funded by Deutsche Forschungsgemeinschaft - Project-ID 424778381 - TRR 295 (S.H., A.A.K. and W.-J.N.) and NeuroCure Clinical Research Centre - Germany's Excellence Initiative-EXC-2049-390688087 (A.A.K.), the Bundesministerium für Bildung und Forschung (BMBF, project FKZ01GQ1802, W.-J.N.) and the European Union (ERC, ReinforceBG, 101077060, W.-J.N.; ERC, TrueBrainConnect, 758985, S.H.). Views and opinions expressed are, however, those of the author(s) only and do not necessarily reflect those of the European Union or the European Research Council. Neither the European Union nor the granting authority can be held responsible for them. We thank all patients who participated in this study. Without their dedication to contribute to the understanding and treatment of Parkinson's disease, our research would not be possible. Computation has been performed on the HPC for the Research cluster of the Berlin Institute of Health.

## Author contributions

Conceptualisation, W.-J.N.; Methodology, T.S.B., S.H., and W.-J.N.; Software, T.S.B., R.M.K., M.G., T.M., F.P., N.L., A.H., S.H. and W.-J.N.; Formal Analysis, T.S.B. and W.-J.N.; Investigation, R.M.K., J.L.B., J.G.V.H., A.C., J.-C.B., and W.-J.N.; Resources, P.K., K.F., G.-H.S. and A.A.K.; Data Curation, T.S.B., R.M.K., J.V., M.C., J.-C.B and B.A.-F.; Writing – Original Draft, T.S.B. and W.-J.N.; Writing – Review & Editing, T.S.B., R.M.K., J.V., M.G., T.M., B.A.-F., A.H., S.H., A.A.K. and W.-J.N.; Visualisation, T.S.B. and W.-J.N.; Supervision, S.H., A.A.K. and W.-J.N.; Project Administration, A.A.K. and W.-J.N.; Funding Acquisition, S.H., A.A.K. and W.-J.N.

## Funding

## Competing interests

A.A.K. reports personal fees from Medtronic and Boston Scientific. G.-H.S. reports personal fees from Medtronic, Boston Scientific, and

Abbott. W.-J.N. serves as a consultant to InBrain and reports personal fees from Medtronic. The remaining authors declare no competing interests.

## Additional information

[1]Movement Disorder and Neuromodulation Unit, Department of Neurology, Charité – Universitätsmedizin Berlin, corporate member of Freie Universität Berlin and Humboldt-Universität zu Berlin, Berlin, Germany. [2]Einstein Center for Neurosciences Berlin, Charité – Universitätsmedizin Berlin, corporate member of Freie Universität Berlin and Humboldt-Universität zu Berlin, Berlin, Germany. [3]Bernstein Center for Computational Neuroscience Berlin, Berlin, Germany. [4]Research Group Neural Interactions and Dynamics, Department of Neurology, Max Planck Institute for Human Cognitive and Brain Sciences, Leipzig, Germany. [5]Neurophysics Group, Department of Neurology, Charité – Universitätsmedizin Berlin, corporate member of Freie Universität Berlin and Humboldt-Universität zu Berlin, Berlin, Germany. [6]Berlin Center for Advanced Neuroimaging, Bernstein Center for Computational Neuroscience, Berlin, Germany. [7]Center for Brain Circuit Therapeutics, Department of Neurology, Brigham and Women's Hospital, Harvard Medical School, Boston, MA, USA. [8]Department of Neurosurgery, Massachusetts General Hospital, Harvard Medical School, Boston, MA, USA. [9]Department of Neurosurgery, Charité – Universitätsmedizin Berlin, corporate member of Freie Universität Berlin and Humboldt-Universität zu Berlin, Berlin, Germany. [10]Technische Universität Berlin, Berlin, Germany. [11]Physikalisch-Technische Bundesanstalt Braunschweig und Berlin, Berlin, Germany. [12]NeuroCure Clinical Research Centre, Charité – Universitätsmedizin Berlin, corporate member of Freie Universität Berlin and Humboldt-Universität zu Berlin, Berlin, Germany. [13]Berlin School of Mind and Brain, Humboldt-Universität zu Berlin, Berlin, Germany. ✉e-mail: julian.neumann@charite.de

