## [Transparent Peer Review file · Nature Communications]

Shared pathway-specific network mechanisms of dopamine and deep brain stimulation for the treatment of Parkinson's disease

Corresponding Author: Professor Wolf-Julian Neumann

Version 0:

Reviewer comments:

Reviewer #1

(Remarks to the Author)

This paper examines the effects of levodopa versus therapeutic DBS on field potentials and interregional coupling metrics, in a group of individuals with Parkinson's disease. Common mechanisms for the two therapies are identified, as well as some distinct effects on LFPs. The summary of the findings in figure 5 is a nice touch given the complexity of the results. Overall the paper is a tour de force both experimentally and analytically. Physiology data were collected from invasive recordings of STN and motor cortex, with these leads externalized for several days postop. This is a novel paradigm for multisite recordings, which have previously been done intraoperatively (high sampling rate but severe time constraints on experimental activities) or using chronically implanted sensing enabled neural interfaces (no time constraints, but limited sampling rates). The paper leverages normative connectomics to assess the structural connectivity of "hot spots" of oscillatory activity. Complex results are presented in a fairly accessible manner.

My only disappointment is that the paper focuses exclusively on low frequencies (beta and below); while gamma band oscillations are a very rich source of knowledge for both levodopa and DBS effects. Nevertheless the paper is pretty dense with just beta and the authors acknowledge the need to explore gamma. I hope the authors will produce subsequent papers in this same patient group analyzing gamma band phenomena.

One result of this paper is support for a major role for the hyperdirect pathway in mechanisms of both levodopa and DBS. There are a few elements of this that could be considered further. First, while there is an obvious mechanism for DBS to affect hyperdirect inputs, the manner in which levodopa does so is not as obvious. Do the authors think that levodopa acts on the subthalamic (or cortical) dopaminergic innervation, even though those pathways are minor compared to striatal innervation? Or is there some way in which the very prominent striatal changes induced by levodopa "propagate" in the circuit to involve the hyperdirect pathway in some manner? Second, a longstanding problem in the field of DBS mechanisms is the need to explain why pallidal and subthalamic DBS have similar motor effects. Modulation of the hyperdirect pathway is a mechanism only available to STN DBS as there is no known hyperdirect innervation of the pallidum. Doesn't this suggest that modulation of other pathways (such as the indirect), orthodromically, predominate over retrograde hyperdirect effects (since similar orthodromic effects would be available to both targets). This is worth a mention in the discussion.

A few minor specific comments are below.

Results

The finding that high beta cortical spectral power is reduced by levodopa – couldn't this easily be confounded by movement? For example movement related beta desynchronization due to dyskinesia or greater limb mobility in the dopamine on state with more spontaneous movement. Related to this, the patient demographic information should mention the proportion of subjects who had dyskinesia preoperatively, perhaps in table s1. The occurrence or non-occurrence of dyskinesia during the testing sessions should be mentioned in results.

Figure 2c – legend refers to gray shading indicating statistical significance – I see the color coding of contacts but not gray shading

Table 1 – MNI coordinates of the STN region with strongest contribution to beta spectral power – the z-coordinates seem quite inferior at -6.4 and -6.6. I understand that the origin of the MNI coordinate system is at the anterior commissure, but I believe that its horizontal axis is defined as parallel to the AC-PC line? If so those z coordinates would, in most persons, correspond to ventral STN rather than dorsal if one examines each subject in their own (non-normalized) AC-PC space. Any explanation for this discrepancy?

Discussion

Interpretations: their results indicate that low beta coupling reflects indirect pathway connectivity while high beta reflects hyperdirect. Presumably their prediction for pallidal-cortical connectivity would be that the parkinsonian state is primarily “represented” in low beta coupling (since there is no known hyperdirect connectivity). Indeed this is the case when pallido-cortical coupling (measured by phase coherence) is compared between the parkinsonian state and dystonia (Wang DD et al J Neurosci 2018). But: Doesn’t the similar efficacy of pallidal and STN stim argue that the most likely mechanism of DBS is the modulation of low beta /indirect pathway connectivity, as that mechanism is available to both targets?.

“Moreover, our findings align with the long-held position of a spectral distinction in cortico-subthalamic coupling, with indirect and hyperdirect pathway interactions being mediated predominantly by low and high beta activity, respectively 55,56”. – check these references – 55 and 56 do not appear to relate to oscillatory phenomena.

“The relationship between pathway-specific therapeutic effects and these high-frequency [gamma] patterns warrants further investigation. Instead, we have extended our findings in the spatial domain by relating invasive multisite cortex – basal ganglia neurophysiology to whole-brain MRI connectomics.” The second sentence here does not seem to follow logically from the first.

congrats on this excellent paper! philip starr UCSF

(Remarks on code availability)

Reviewer #2

(Remarks to the Author)

In this paper, the authors investigated the potential underlying network mechanisms of Levodopa and deep brain stimulation treatments of Parkinson’s disease (PD). Eighteen people with PD undergoing bilateral DBS targeting subthalamic nucleus (STN) and unilateral electrocorticography (ECoG) targeting sensorimotor cortex participated in the experiments involving recordings local field potentials (LFPs) and ECoG simultaneously when the participants were at rest, in different therapeutic conditions including OFF therapy (N = 18), ON Levodopa therapy (N = 18), and ON DBS therapy (N = 9). Based on the recorded signals, the authors performed several analyses including spatio-spectral decomposition based power spectral analysis, undirected (imaginary coherence) and directed (time-reversed Granger Causality) connectivity analysis, bispectrum based time delayed analysis, and dMRI/fMRI-based structural/functional connectivity analysis. The derived results suggested that Levodopa and DBS treatments have shared pathway-specific network mechanisms by diminishing connectivity in the high beta band, which has been suggested to be activity attributed to the hyperdirect pathway. Overall, the manuscript is well written. The effects of medication and DBS on the cortical activities and cortical-STN connectivity measured using ECoG is novel, even though the effect of dopamine medication and DBS on STN LFPs have been reported in multiple previous studies.

However, my main impression of the drawback is that, the presented results that have been used to support the Inferences/conclusions are relatively weak. In other words, the conclusions are not well supported by the results. The effect sizes for several analyses were relatively small even though statistically significant. Some of the results are different from what have been repeated in multiple previous studies. There are also some lack of methodological details which make the interpretation of some results difficult. Please see below my comments and hopefully they will be useful.

1. About the conclusion that DBS and dopamine have different mesoscale effects, especially within the STN (reported in Figure 2): I’m wondering how much this is due to the difference in the sample sizes for the Levodopa and DBS conditions. Only half of the all eighteen patients received stimulation (i.e., N = 18 and N = 9 for Levodopa and DBS, respectively). Because of this, the baseline in terms of the power modulation in Fig 2 and the modulation of cortico-subthalamic coupling in Fig. 3 during OFF therapy were different for different experimental conditions. This different baseline levels could potentially affect the statistic especially giving the relatively small sample sizes. For instance, there was a more prominent peak at high beta band in the STN LFP in STN-DBS condition compared with Levodopa condition (Fig. 2A). Similarly, there was a more prominent peak at alpha band in the cortex-STN coupling in Levodopa condition compared with STN-DBS condition (Fig. 3B). To alleviate this concern, an extra analysis only considering those participated with both Levodopa and DBS recordings (N = 9) would be helpful. One-way ANOVA with three conditions (No treatment, ON Medication, ON DBS) can be used for these 9 participants, applying to the low beta and high beta frequency band separately, to see if there is really a difference between different treatment states.

2. I also feel uncomfortable about the conclusion that DBS only reduces high beta but not low beta in the STN. This is contradictory to multiple previous studies showing that DBS can reduce average power or peaks in a broad frequency band including the low beta frequency band, such as: <https://pubmed.ncbi.nlm.nih.gov/38580641/>; <https://pubmed.ncbi.nlm.nih.gov/22787444/>; <https://pubmed.ncbi.nlm.nih.gov/16623853/>
The results reported in this study can be compromised due to a few reasons: 1) the small sample size of 9 for DBS condition,

2) maybe most of the sampled participants only had peaks in the high frequency band in the PSD of the STN LFPs. Despite this, in Subject EL008, there seem to be a peak in the low beta band (<20 Hz) in the STN PSD. In this patient, is this low beta peak suppressed by DBS? Would it be possible to separate the hemispheres with peaks in the low beta band vs. those vs peaks in the high beta band, and see if it is true that only high beta activities are modulated by DBS? 3) the dose of the DBS. The effect of the DBS changes with stimulation intensity. Higher stimulation intensity induced more changes in beta across a wider frequency band (<https://pubmed.ncbi.nlm.nih.gov/22675296/>;) – Did the stimulation intensity used during the recording induce changes in the UPDRS scores? How does the stimulation intensity used in the test compared with intensity used for clinical stimulation? 4) How the power is normalised. If the power is normalised against the total power within the 1-45 Hz band, a reduction in the higher beta band may lead to a relative increase in the alpha or low beta band. So it is actually difficult to interpret the results. In addition, this normalisation may also mask the potential effect of DBS in reducing the total power of this broad frequency band.

3. I also feel uncomfortable about the conclusion that DBS does not change local activities in the cortex. Whitemer et al. 2012 (<https://pubmed.ncbi.nlm.nih.gov/22675296/>) who recorded cortical activity with ECoG during DBS showed that cortical signals over the estimated origin of the hyperdirect pathway (estimated using DTI) did demonstrated attenuation of beta hypersynchrony during DBS delivered dorsal to or within STN, whereas signals from non-specific regions of motor cortex were not attenuated. For the position of the ECoG in this study, it was mentioned that the ECoG strip was aimed posteriorly towards the hand knob region of the motor cortex. But the actual location of the ECoG contacts shown in Figure 2A was very variable across participant. How many participants actually have ECoG contacts over the 'hand knob region'? How many contact showed a high beta peak in the PSD? Is there any contact on the origin of the hyper direct pathway? Would the difference in the effect of DBS vs medication reported here actually be the 'spatial specificity' of the effect? Would it be possible that the effect of DBS on cortex is more focused on specific area, whereas the effect of dopamine is more wide spread? Would the algorithm used in the study (quantifying the grand average) capture this potential difference? Maybe the author should re-run the analysis focusing on the ECoG contacts on the hyperdirect pathway vs. those on other areas?

4. There is lack of details in the methods for results presented in Figure 2: 1)What is 'grand average power spectra' as reported in Fig 2A? Are they the average across all bipolar contacts for different regions? Or are they the most significant components? 2) All patients received bilateral STN implantation, but the sample size is 9, which the patient number for the effect of DBS. How did the authors deal with the two hemispheres for each participants? 3) How power is normalised? What is the frequency band considered as total? 4) In Fig. 2C, it seems the contributions to power for different contacts in the same level were quantified and mapped in MNI space. I was just wondering how the directionality of the DBS lead was identified, and how reliable this information is.

5. For the results on the time delays suggesting mono- and poly-synaptic pathway communication (results reported in Fig 3), I'm wondering how the results may be impacted by the selection of patients for analysis. The authors mentioned, "4 subjects had no positive time delays which met the confidence criteria, and were thus excluded from the time delay analysis." (Last sentence of Page 23). This further reduced the sample sizes in both experimental conditions. I was wondering if the authors could explain a bit more regarding why there were 4 out of 18 subjects (22.22%) did not meet the criteria. In addition, here only those connections with positive tau (motor cortex to STN) were analysed, would this selection bias the results toward the hypothesis and lead to type 1 error? How about the connections with negative tau that representing information flow from STN to motor cortex? Why they should be excluded from the statistics?

6. Following the previous comment, on Page 9, the authors mentioned that, 'in addition to suggesting a combination of mono- and poly-synaptic cortico-subthalamic communication, these results further highlight the suppression of hyperdirect pathway activity as a shared therapeutic mechanism of dopamine and DBS'. 1) I guess this is based on the statistics shown in Fig. 3D. However, it was not clear to me how the statistics were conducted. Is there any difference in the 'Time delay estimate strength' caused the treatment (medication or DBS)? The stars indicating statistical significance were indicated on the 'peak count (% maximal)' plots. What does this 'peak count (% maximal)' mean? How was it calculated? What is the sample size of the statistics? Were the test controlled for multiple comparison correction? 2) Although significant, the difference in the time delay estimate strength from motor cortex to STN at 1-9 ms window (i.e., first bin) between Levodopa therapeutic conditions was pretty small. Considering this is one of the main results supporting the mono- and poly-synaptic argument, I was wondering apart from theoretical interpretation with the support of existing studies about hyperdirect pathway, if there is any other evidence to show that this measurement is actually clinically meaningful. It was just because the clinical meaning of beta power and cortex-STN coupling was relatively clearer, but not the strength of time delay estimation used here.

7. The results on the link between the connectivity of the low beta and high beta with the indirect and hyperdirect pathway, respectively, are really nice. Can these be further confirmed by the time delay analyses? Would the high beta band coupling have more peaks in the 1-9 ms window, and the low beta coupling have more spread time delays or longer time delays? This will make the time delay analyses more convincing.

8. Fig. 4, it seems not very clear to me how the estimated (y-axis) and empirical (x-axis, upper/left panel) were quantified. The stats showed that the estimated number of fibres were correlated with empirical number of fibres for both high and low beta coupling. I was just wondering why the statistic was applied to empirical number of fibres, rather than directly applied to the contribution to connectivity (lower/right panel), which was derived directly from the electrophysiological recordings.

9. There is no direct support in the statement the authors made in the 'Discussion' (Page 12: 'Altogether, we argue that excessive hyperdirect pathway communication between the cortex and STN is a central factor in PD pathology'. Is there any

correlation between the connectivity in the hyperdirect pathway and UPDRS scores when OFF any treatment? Does the changes in the connectivity induced by treatment (medication or DBS) correlate with the effect of the treatment? Those analyses could support the statement the authors made, but was not presented.

Methodological details and presentation:

10. What is the total duration of recording for each conditions for each participant?

11. For the structural and functional connectivity analysis based on group connectomes, I was wondering if there is any guidance or rationale for the selection of radius (5 mm and 3 mm for ECoG and STN-LFP here, respectively) and how it would affect the results. I saw difference values were used in existing studies, for example, a radius of 2 mm was used in Oswal et al. 2021.

12. The last two plots in Figure 3D (the bottom right): how are they different from those presented in 3B?

13. Fig. 5, it seems the results/plots about power, coherency, granger causality, and time delays shown here were not consistent with the results shown in previous figures. Were they just schematics rather than actual results?

(Remarks on code availability)

Reviewer #3

(Remarks to the Author)

(Remarks on code availability)

Reviewer #4

(Remarks to the Author)

This is an interesting study on patients with Parkinson's Disease that underwent deep brain stimulation (DBS) surgery. The authors use a multimodal dataset (intraoperative LFP recordings and fMRI connectomes) to investigate the differences in cortico-subthalamic network changes induced by dopaminergic drug treatment vs DBS, even if their therapeutic effect are similar. Their findings suggest a difference at the mesoscale/LFP level, while they have similar effects at the macroscale (suppression of the hyperdirect pathway). It is also a great inclusion to specifically address these (lack of) findings in previous studies. I do believe the study is of high interest for the field, and by addressing the comments below would be a great addition to the literature. I do believe the authors will be able to address my points below without difficulty.

Few major comments:

1) Overall, even if the authors specify it later, they seem to suggest that they have computed connectivity measures during both medication ON and medication OFF, and compared it with spectral features (e.g. from the abstract). However, that is not the case, as connectivity measures were based on an fMRI connectome, and compared with spectral (and other) features from the LFPECoG recordings. I would advise the authors to be clear with this analysis choice in the abstract, to facilitate the reader to understand what were the methodological choices taken in this study. This can be placed in the methods.

a. Due to this point, I would invite the authors to revise this statement for clarity "Our study is the first to systematically compare the effects of dopamine and STN-DBS on local mesoscale and interregional macroscale circuit communication with fully-invasive neurophysiology and MRI connectomics in Parkinson's disease."

2) While it is great to see multiple causality-related methods being deployed to validate their findings, it would be great to have the authors explain why they choose to have multiple causality-related methods (with coherency, Granger causality, time-delay), and what is the rationale behind their usage in different contexts (e.g. coherency is taken to just estimate undirected connectivity, while granger causality for directed connectivity).

3) In "Oscillatory connectivity analysis", it is unclear how the order 60 is chosen for Granger Causality. For example, in their cited work (22), the authors indicate that they have used a bootstrap based testing for optimizing the order p (with BIC). Was a similar preliminary analysis conducted on the data to choose the model order? This is important as it can influence the causality relationship found by the method.

4) In Time delay analysis (Methods), the author mention to use "Using the 200 bootstrapped segments, 80% confidence intervals were computed for each connection from the segment τ ". I would suggest the authors to estimate the underlying distribution to potentially increase statistical insights, or have larger number of bootstraps. At the same time, they indicate "4 subjects had no positive time delays which met the confidence criteria". I would suggest to 1) for clarity to the reader, rephrase as "4 subjects out of ...". And 2) discuss the fact that 4 subjects showed no time delays, compared to other patients.

5) In the main text, in the introductory paragraph of the Result section, it is mentioned that ECoG strips were used for recordings perioperatively. However in the limitations it is mentioned that "all recordings were performed a few days following implantation of the DBS electrodes in the STN". Were the recordings made during surgery (as I imagine ECoG strips were temporarily placed), or ECoGs were part of a chronic implantation and therefore recordings were made few days later (and if so, how many (mean and std)? Considering the type of recordings (medication ON/OFF, DBS ON/OFF), I imagine it is the latter, due to time required.

Few minor comments:

1) For clarity to the reader, and improve readability, I would invite the authors to include references to main papers proposing these key theories, in the following statement : "to this theory with the demonstration that: a) cortex drives subthalamic

activity; b) cortex – STN oscillatory connectivity is spatio-spectrally linked to hyperdirect and indirect pathway communication, as revealed through whole-brain MRI connectomics; and c) both dopamine and DBS suppress 1-9 ms monosynaptic input to the STN. “

(Remarks on code availability)

The code does provide a README file for usage.

For the code in python, it provides a configuration file for the environment, which is greatly appreciated, and works great during testing.

However, as the data is not provided, it is difficult to validate group analyses and other methods. There is a DEMO section to partially address this requirement, but that is not helpful in replicating the paper itself.

From the point of view of code organization, it is well structured and documented, and the analysis routines are easily useable. For the python codebase, the jupyter notebook are especially welcomed.

Regarding code review, I wanted to add that the currently posted zenodo link is non-functional. However the authors have posted the following github: https://github.com/neuromodulation/manuscript-binns_cortex_stn_comm

Version 1:

Reviewer comments:

Reviewer #1

(Remarks to the Author)

the authors have responded appropriately to reviewer comments.

(Remarks on code availability)

Reviewer #2

(Remarks to the Author)

Thank you for the extensive revision of the previous comments. The extra analyses and results from added participants in the revised version are convincing. I think the manuscript will add new knowledge to the field and will be of great interest to the community. However, I still have one remaining major concern and some minor comments, as detailed below:

Major:

I appreciate the idea of linking oscillatory connectivity measures with normative MRI-based whole-brain connectomics.

However, I am unclear about the rationale for using ‘Medication’ as an independent predictor variable, alongside oscillatory spatial patterns, to predict the response variable of dMRI/fMRI-based connectivity using the following model architecture: “dMRI/fMRI connectivity ~ Spatial patterns + Medication + (1 | Subject)”.

As far as I’m aware of, the dMRI/fMRI-based connectivity is derived purely from the coordinates of the DBS/ECOG electrodes, the connectivity should be identical for the same ECOG-LFP contact pair across OFF therapy, ON levodopa, and ON STN-DBS conditions. My concerns with this analysis include:

1. You have already shown that Medication significantly reduces the maximized imaginary coherence in the high beta band (Fig. 3d). In the aforementioned model, this will almost certainly result in a positive and significant coefficient for high beta imaginary coherence. For example:

• $Y = A * \text{Beta_High_1} + B * \text{Med_OFF}$;

• $Y = A * \text{Beta_High_2} + B * \text{Med_ON}$;

Here, $Y > 0$, $\text{Beta_High_1} > \text{Beta_High_2} > 0$, $\text{Med_ON} = 1$, and $\text{Med_OFF} = 0$ (or -1, I assume). Thus, coefficient A will be positive with a significant p-value.

2. The goal here is to determine if the effect of levodopa (and similarly for STN-DBS) on high/low beta oscillatory connectivity measures can be explained by the underlying dMRI/fMRI-based connectivity. I believe a more intuitive model would be:

• “Spatial patterns ~ dMRI/fMRI connectivity + Medication + (1 | Subject)”.

This way, you can investigate whether both medication and dMRI/fMRI connectivity independently contribute to the variance in ‘spatial patterns’ (high/low beta).

3. You could also include STN-DBS in the same model as another independent predictor variable, as it also significantly affects ‘spatial patterns’ as previously discussed.

4. It would be helpful to provide details of each linear mixed effect model in a supplementary table to make it easier to follow the presented results.

Minor comments:

1. I suggest specifying imaginary coherence at the beginning, instead of coherence-based measures, as it is too broad.

2. In supplementary Figure 5, please update the results to reflect the new cohorts, i.e., N=18 and 12 for dopamine and STN-DBS, respectively.

3. In Figure 4, please provide R² values for both models.

(Remarks on code availability)

Reviewer #3

(Remarks to the Author)

(Remarks on code availability)

Reviewer #4

(Remarks to the Author)

I wanted to thank the authors as I have appreciated the responsiveness and completeness in addressing all the points and concerns that were raised by me and other reviewers.

I do believe that the manuscript has been significantly strengthened and is now clearer, including the revision of the figures (due to the slight change in statistical procedures, such as number of folds). I do appreciate the revision of the limitations sections as well.

I accept the manuscript in its current form.

(Remarks on code availability)

The code does provide a README file for usage, and I maintain the comments expressed in my previous revision. I have appreciated the revision of the zenodo link and inclusion of the github as well.

Regarding data availability, the authors have expressed an understandable limitation in direct open sharing. As a general point, my comment was in light of the limitations of my review of the results, as I can only give direct insight on code usage for the demo data portion.

For the code in python, it provides a configuration file for the environment, which is greatly appreciated, and works great during testing. From the point of view of code organization, it is well structured and documented, and the analysis routines are easily useable. For the python codebase, the jupyter notebook are especially welcomed.

Version 2:

Reviewer comments:

Reviewer #2

(Remarks to the Author)

The authors have addressed all the comments.

(Remarks on code availability)

Reviewer #3

(Remarks to the Author)

(Remarks on code availability)

Response to Reviewers

Shared pathway-specific network mechanisms of dopamine and deep brain stimulation for the treatment of Parkinson's disease

Authors

Thomas S. Binns^{1,2,3}, Richard M. Köhler¹, Jojo Vanhoecke¹, Meera Chikermane¹, Moritz Gerster^{3,4,5}, Timon Merk¹, Franziska Pellegrini^{3,6}, Johannes L. Busch¹, Jeroen G.V. Habets¹, Alessia Cavallo^{1,2,3}, Jean-Christin Beyer¹, Bassam Al-Fatly¹, Ningfei Li¹, Andreas Horn^{1,2,7,8}, Patricia Krause¹, Katharina Faust⁹, Gerd-Helge Schneider⁹, Stefan Haufe^{2,3,6,10,11}, Andrea A. Kühn^{1,2,3,12,13}, Wolf-Julian Neumann^{1,2,3,*}

Affiliations

¹Movement Disorder and Neuromodulation Unit, Department of Neurology, Charité – Universitätsmedizin Berlin, corporate member of Freie Universität Berlin and Humboldt-Universität zu Berlin, Berlin, Germany

²Einstein Center for Neurosciences Berlin, Charité – Universitätsmedizin Berlin, corporate member of Freie Universität Berlin and Humboldt-Universität zu Berlin, Berlin, Germany

³Bernstein Center for Computational Neuroscience Berlin, Berlin, Germany

⁴Research Group Neural Interactions and Dynamics, Department of Neurology, Max Planck Institute for Human Cognitive and Brain Sciences, Leipzig, Germany

⁵Neurophysics Group, Department of Neurology, Charité – Universitätsmedizin Berlin, corporate member of Freie Universität Berlin and Humboldt-Universität zu Berlin, Berlin, Germany

⁶Berlin Center for Advanced Neuroimaging, Bernstein Center for Computational Neuroscience, Berlin, Germany

⁷Center for Brain Circuit Therapeutics, Department of Neurology, Brigham and Women's Hospital, Harvard Medical School, Boston, MA, USA

⁸Department of Neurosurgery, Massachusetts General Hospital, Harvard Medical School, Boston, MA, USA

⁹Department of Neurosurgery, Charité – Universitätsmedizin Berlin, corporate member of Freie Universität Berlin and Humboldt-Universität zu Berlin, Berlin, Germany

¹⁰Technische Universität Berlin, Berlin, Germany

¹¹Physikalisch-Technische Bundesanstalt Braunschweig und Berlin, Berlin, Germany

¹²NeuroCure Clinical Research Centre, Charité – Universitätsmedizin Berlin, corporate member of Freie Universität Berlin and Humboldt-Universität zu Berlin, Berlin, Germany

¹³Berlin School of Mind and Brain, Humboldt-Universität zu Berlin, Berlin, Germany

*Correspondence: julian.neumann@charite.de

Reviewer 1

This paper examines the effects of levodopa versus therapeutic DBS on field potentials and interregional coupling metrics, in a group of individuals with Parkinson's disease. Common mechanisms for the two therapies are identified, as well as some distinct effects on LFPs. The summary of the findings in figure 5 is a nice touch given the complexity of the results. Overall the paper is a tour de force both experimentally and analytically. Physiology data were collected from invasive recordings of STN and motor cortex, with these leads externalized for several days postop. This is a novel paradigm for multisite recordings, which have previously been done intraoperatively (high sampling rate but severe time constraints on experimental activities) or using chronically implanted sensing enabled neural interfaces (no time constraints, but limited sampling rates). The paper leverages normative connectomics to assess the structural connectivity of "hot spots" of oscillatory activity. Complex results are presented in a fairly accessible manner.

My only disappointment is that the paper focuses exclusively on low frequencies (beta and below); while gamma band oscillations are a very rich source of knowledge for both levodopa and DBS effects. Nevertheless the paper is pretty dense with just beta and the authors acknowledge the need to explore gamma. I hope the authors will produce subsequent papers in this same patient group analyzing gamma band phenomena.

1. One of the results of this paper is support for a major role for the hyperdirect pathway in mechanisms of both levodopa and DBS. There are a few elements of this that could be considered further. First, while there is an obvious mechanism for DBS to affect hyperdirect inputs, the manner in which levodopa does so is not as obvious. Do the authors think that levodopa acts on the subthalamic (or cortical) dopaminergic innervation, even though those pathways are minor compared to striatal innervation? Or is there some way in which the very prominent striatal changes induced by levodopa "propagate" in the circuit to involve the hyperdirect pathway in some manner?

Reply: The reviewer raises an important point with regard to the potential mechanisms through which dopaminergic agents could act on the hyperdirect pathway. While we cannot directly address this question analytically further, we can speculate on candidate mechanisms and comment on their neurobiological plausibility. For that, we reiterate the presented evidence of hyperdirect pathway modulation by dopamine presented in our study: we found that levodopa intake could suppress cortex – STN beta coherence that is correlated with hyperdirect pathway fibre tract connectivity, and it further reduced short latency input to the STN without affecting directionality of information flow.

We believe that the most plausible effect that can explain these findings takes root in the susceptibility of the circuit to beta resonance from hyperdirect cortical input to the STN. This resonance can be altered through a dopaminergic modulation of striatal medium spiny neuron activity, and consecutive pallido-subthalamic communication.

In the absence of dopamine, we expect excessive activation of D2+ medium spiny neurons with consequential reciprocal rhythmic GPe – STN subcortical network activity to generate low beta frequency oscillations, as suggested by a combination of empirical and computational modelling studies (Pavlidis *et al.*, 2015, DOI: [10.1371/journal.pcbi.1004609](https://doi.org/10.1371/journal.pcbi.1004609); Holgado *et al.*, 2010, DOI: [10.1523/jneurosci.0817-10.2010](https://doi.org/10.1523/jneurosci.0817-10.2010)).

Building on these studies, more recent works have shown that hyperdirect beta input to the STN can exaggerate this reciprocal rhythmic activity in PD and drive low beta oscillations (Oswal *et al.*, 2021, DOI: [10.1038/s41467-021-25366-0](https://doi.org/10.1038/s41467-021-25366-0)). This is expected to have direct effects on basal ganglia output via the internal pallidum and consequently thalamo-cortical excitability, with potential for further entrainment of the cortex – basal ganglia circuits in excessive beta synchrony and thus hyperdirect output.

Dopamine increases GABAergic output of D1+ direct pathway neurons, and decreases GABAergic output from D2+ indirect pathway neurons, leading to net inhibition of the STN. While dopamine, unlike DBS, does not block afferent input to the STN, it does reduce STN – GPe resonance, leading to lower susceptibility of the circuit to high beta input from the hyperdirect pathway.

This theory can be tested experimentally, and we are currently performing further studies on this matter by providing cortical direct electrical stimulation pulses during subthalamic LFP recordings. If correct, we would expect that: a) in the absence of dopamine, lower stimulation currents are required to trigger a beta wave in the STN; and b) relative phase locking of stimulation to STN – GPe activity predicts the effect strength. The results so far are encouraging, with stronger beta resonance in STN after cortical stimulation pulses in the dopaminergic OFF state, however these results require further validation in a larger sample size, for which patient recruitment is ongoing in our centre.

Beyond that, as mentioned by the reviewer, an additional or alternative more direct mechanism could be at play. Given that both the STN and cortex exhibit expression of dopamine receptors, dopamine may directly influence hyperdirect pathway activity. At the level of the STN, a study investigating the effects of dopaminergic agents via intra-STN administration on subthalamic neurons in non-human primates reported that D1 agonists can imminently reduce STN firing (Galvan *et al.*, 2014, DOI: [10.1152/jn.00849.2013](https://doi.org/10.1152/jn.00849.2013)). This could lead to a circuit wide modulation of cortex – basal ganglia activity, starting from reduced susceptibility to glutamatergic input from the hyperdirect afferents and leading to lower rhythmicity in motor cortical hyperdirect motor output via the abovementioned downstream modulation of thalamo-cortical excitability.

A similar mechanism could be attributed to direct modulation of cortical dopamine receptors, but given the relatively higher complexity of the potential effects, we refrain from further speculation on this.

Ultimately, given the relatively low dopamine receptor density of cortical and subthalamic neurons when compared to striatal medium spiny neurons, we believe that circuit modulation via the striatum is the more likely mechanism, which may additionally be affected by direct effects of dopaminergic innervation of the cortex and STN, and can account for potential mechanisms involving the GPi (see next comment). We have now added a short discussion of these considerations to the manuscript.

Changes in manuscript: The *Discussion* section now reads: “The mechanisms by which dopaminergic agents act on the hyperdirect pathway remain speculative. One plausible explanation is that dopamine reduces cortex – STN beta coupling by modulating striatal medium spiny neuron activity, leading a modulation of pallido-subthalamic communication. In the absence of dopamine, reciprocal GPe – STN network activity drives pathological beta activity^{16,55}. While dopamine, unlike STN-DBS, does not block glutamatergic cortical input to the STN, it may lower the susceptibility to beta resonance from the hyperdirect pathway. Alternatively, or in addition, local dopamine receptor activation in the STN or cortex may also influence hyperdirect pathway communication directly.”

2. ... Second, a longstanding problem in the field of DBS mechanisms is the need to explain why pallidal and subthalamic DBS have similar motor effects. Modulation of the hyperdirect pathway is a mechanism only available to STN DBS as there is no known hyperdirect innervation of the pallidum. Doesn't this suggest that modulation of other pathways (such as the indirect), orthodromically, predominate over retrograde hyperdirect effects (since similar orthodromic effects would be available to both targets). This is worth a mention in the discussion.

Reply: We agree with the reviewer and would like to point out, also in light of the previous comment, that we believe the shared therapeutic effects of dopaminergic medication, subthalamic DBS, and pallidal DBS, is their ability to modulate basal ganglia output with circuit wide consequences in the cortex – basal ganglia loop. In brief, dopaminergic innervation reduces STN and GPi excitability, and thus susceptibility to hyperdirect input with orthodromic changes in thalamo-cortical activation and consequent reduction of hyperdirect synchrony. STN-DBS directly suppresses hyperdirect afferent input to the STN and reduces STN firing, while GPi-DBS overrides subthalamic, and thus hyperdirect modulation of the basal ganglia, again leading to orthodromic modulation of cortical excitability.

However, it should be noted that pallidal DBS does not allow for a reduction in dopaminergic medication to the same extent as STN-DBS, which we can speculate to be attributed to the additional modulation of the hyperdirect pathway that dopamine provides, if pallidal DBS does not directly affect it.

We thank the reviewer for this hint, and have specifically added the paradox of functional neurosurgery with similar effects of STN- and GPi-DBS to the *Discussion*.

Changes in manuscript: The *Discussion* section now reads: “Notably, while pallidal DBS is similarly effective to STN-DBS, pallidal stimulation does not directly affect hyperdirect afferents to the STN. We speculate that the more immediate modulation of the hyperdirect pathway with STN-DBS may explain the clinical observation that STN-DBS allows for a stronger reduction in dopaminergic medication when compared to pallidal DBS in PD patients. Nevertheless, even pallidal DBS likely induces modulation of hyperdirect synchrony through orthodromic changes in thalamo-cortical activation.”

3. The finding that high beta cortical spectral power is reduced by levodopa – couldn't this easily be confounded by movement? For example movement related beta desynchronization due to dyskinesia or greater limb mobility in the dopamine on state with more spontaneous movement.

Reply: While we are carefully discarding periods of explicit movements in the recordings as artefacts, we accept the reviewer's suggestion that covert movement or a more pro- or dyskinetic state could contribute to the suppression of high beta cortical activity. Generally, our opinion is that it is impossible to rule out the potential for changes in muscle tone and covert

motor excitability to be a primary effect associated with beta modulation in PD, which may ultimately pose a hen or egg problem.

If beta activity is related to motor excitability, then dopamine is expected to lead to beta reduction and increased motor output. Humans are never truly at rest, which to us is most vividly observable in the form of alleviation of hypomimia with dopamine, which leads the face to “come to life”, even in the absence of voluntary facial expression.

In fact, we believe that the problem mentioned by the reviewer even extends to STN beta modulation, which could be the consequence of covert movement or change in muscle tone.

On the other hand, modulation of beta could cause the change of motor excitability, and higher motor output could be the consequence of this beta modulation.

Ultimately, we are willing to accept this uncertainty and would be more concerned if these activity patterns were not responsive to motor activation, as the change in movement and muscle tone are part of the primary symptom complex in PD that we are targeting with DBS and dopaminergic medication. We consider the cortico-spinal tract and motor unit as a neural extension of the motor cortex, so it is only natural that changes in motor cortex activity are associated with modulations of muscle tone and vice versa.

We now acknowledge the possibility that covert movement and dyskinetic states may have influenced the modulation of beta activity in the *Discussion*.

Changes in manuscript: The *Discussion* section now reads: “A trivial explanation for the latter effect could be that ON levodopa recordings in our cohort were contaminated by overt or covert motor output – such as through a more dyskinetic state – associated with reduced beta power. To address this potential problem, we have carefully excluded periods with movements from the data, and attended all recordings to ensure that patients were truly at rest. Nevertheless, we acknowledge that this does not rule out potential effects from changes in muscle tone, which could either be a cause or consequence of changes in beta activity.”

4. Related to this, the patient demographic information should mention the proportion of subjects who had dyskinesia preoperatively, perhaps in table s1.

Reply: We have updated Supplementary Table 1 to include information about the occurrence of dyskinesia preoperatively. Indeed, the majority of patients (17 of 21) exhibited some degree of dyskinesia preoperatively.

5. Figure 2c – legend refers to gray shading indicating statistical significance – I see the color coding of contacts but not gray shading.

Reply: Thank you for pointing us to this lack of clarity. The information on shading was meant to relate to all panels after the specific information on panels a-c, but upon rereview we

noticed that it is indeed only relevant for panel a. Therefore, we have now moved this information to avoid confusion.

6. Table 1 – MNI coordinates of the STN region with strongest contribution to beta spectral power – the z-coordinates seem quite inferior at -6.4 and -6.6. I understand that the origin of the MNI coordinate system is at the anterior commissure, but I believe that its horizontal axis is defined as parallel to the AC-PC line? If so those z coordinates would, in most persons, correspond to ventral STN rather than dorsal if one examines each subject in their own (non-normalized) AC-PC space. Any explanation for this discrepancy?

Reply: This is a common issue that results from the inaccuracy of the MNI coordinate system which “roughly” aligns to the anterior commissure and AC-PC line. For reference, the figure below depicts the MNI origin ($x=0$, $y=0$, $z=0$), which resides roughly 4 mm above the real AC-PC line. Consequently, this results in a shift of coordinates. For reference, the STN sweet spot described by Caire *et al.* (2013; DOI: [10.1007/s00701-013-1782-1](https://doi.org/10.1007/s00701-013-1782-1)) converts from $x=12.02$, $y=-1.53$, and $z=1.91$ in AC-PC coordinates to $x=12.58$, $y=-13.41$, and $z=-5.87$ in MNI coordinates. Similarly, the pallidal DBS target described in Starr *et al.* (2006; DOI: [10.3171/jns.2006.104.4.488](https://doi.org/10.3171/jns.2006.104.4.488)), converts from $x=20$, $y=5.8$, and $z=0.5$ in AC-PC to $x=22.37$, $y=-5.57$, and $z=-4.97$ in MNI coordinates.

The average z-coordinate of previously published DBS therapeutic response sweet-spots as listed in Table 2 of Dembek *et al.* (2019; DOI: [10.1002/ana.25567](https://doi.org/10.1002/ana.25567)) is $z=-6.01$ mm. Previously published coordinates for peak beta power in a study across 106 PD patients were $z=-6.4$ (Darcy *et al.*, 2022, DOI: [10.1016/j.expneurol.2022.114150](https://doi.org/10.1016/j.expneurol.2022.114150)). In terms of “height” across

inferior-superior axes, these points are situated in the upper half of the nucleus (nicely visualized by Dembek *et al.*, 2019 in Fig. 6). Thus, beta power localises to the motor division of the STN, which is more immediately segregated via the anterior-posterior axis than the dorso-ventral axis.

7. Interpretations: their results indicate that low beta coupling reflects indirect pathway connectivity while high beta reflects hyperdirect. Presumably their prediction for pallidal-cortical connectivity would be that the parkinsonian state is primarily “represented” in low beta coupling (since there is no known hyperdirect connectivity). Indeed this is the case when pallido-cortical coupling (measured by phase coherence) is compared between the parkinsonian state and dystonia (Wang DD et al J Neurosci 2018). But: Doesn’t the similar efficacy of pallidal and STN stim argue that the most likely mechanism of DBS is the modulation of low beta /indirect pathway connectivity, as that mechanism is available to both targets?

Reply: We thank the reviewer for suggesting this work; it is indeed a perfect fit to further support our interpretation of low vs. high beta coupling. We now cite the paper in the *Discussion*, before the abovementioned section on pallidal vs. STN-DBS added in response to comment 2.

Changes in manuscript: The *Discussion* section now reads: “This also aligns with a previous ECoG-LFP study reporting predominantly low beta coupling related to PD pathophysiology between cortex and internal pallidum, where there is no hyperdirect pathway connectivity⁵⁶.”

8. “Moreover, our findings align with the long-held position of a spectral distinction in cortico-subthalamic coupling, with indirect and hyperdirect pathway interactions being mediated predominantly by low and high beta activity, respectively 55,56”. – check these references – 55 and 56 do not appear to relate to oscillatory phenomena.

Reply: Our apologies, these references were indeed incorrect, and we thank the reviewer for highlighting this error. The intended references were 18 (Oswal *et al.*, 2021, DOI: [10.1038/s41467-021-25366-0](https://doi.org/10.1038/s41467-021-25366-0)) and 30 (Oswal *et al.*, 2016, DOI: [10.1093/brain/aww048](https://doi.org/10.1093/brain/aww048)). We have updated the text accordingly. This error has now been corrected.

Changes in manuscript: The *Discussion* section now reads: “Our findings further corroborate previous studies suggesting that cortico-subthalamic interactions via the indirect pathway are mediated predominantly by low beta coupling, while high beta coupling reflects hyperdirect communication^{18,30}.”

9. “The relationship between pathway-specific therapeutic effects and these high-frequency [gamma] patterns warrants further investigation. Instead, we have extended our findings in the spatial domain by relating invasive multisite cortex – basal ganglia neurophysiology to whole-brain MRI connectomics.” The second sentence here does not seem to follow logically from the first.

Reply: We thank the reviewer for highlighting this, and hope the following changes improve the clarity of the text:

Changes in manuscript: The *Discussion* section now reads: “As a final noteworthy limitation, we would like to highlight that the functional and structural connectivity analyses relied on atlas data and group connectomes, as patient-specific fMRI and dMRI scans were not taken. It is thus unclear to what degree the connectomic findings relate to individual anatomy. On the other hand, this has the advantage that robust connectomes derived from large cohorts have been validated multiple times^{18,26,27}, with superior image quality when compared to individual clinical MRI scans.”

Reviewer 2

In this paper, the authors investigated the potential underlying network mechanisms of Levodopa and deep brain stimulation treatments of Parkinson's disease (PD). Eighteen people with PD undergoing bilateral DBS targeting subthalamic nucleus (STN) and unilateral electrocorticography (ECoG) targeting sensorimotor cortex participated in the experiments involving recordings local field potentials (LFPs) and ECoG simultaneously when the participants were at rest, in different therapeutic conditions including OFF therapy (N = 18), ON Levodopa therapy (N = 18), and ON DBS therapy (N = 9). Based on the recorded signals, the authors performed several analyses including spatio-spectral decomposition-based power spectral analysis, undirected (imaginary coherence) and directed (time-reversed Granger Causality) connectivity analysis, bispectrum based time delayed analysis, and dMRI/fMRI-based structural/functional connectivity analysis. The derived results suggested that Levodopa and DBS treatments have shared pathway-specific network mechanisms by diminishing connectivity in the high beta band, which has been suggested to be activity attributed to the hyperdirect pathway. Overall, the manuscript is well written. The effects of medication and DBS on the cortical activities and cortical-STN connectivity measured using ECoG is novel, even though the effect of dopamine medication and DBS on STN LFPs have been reported in multiple previous studies.

However, my main impression of the drawback is that, the presented results that have been used to support the Inferences/conclusions are relatively weak. In other words, the conclusions are not well supported by the results. The effect sizes for several analyses were relatively small even though statistically significant. Some of the results are different from what have been repeated in multiple previous studies. There are also some lack of methodological details which make the interpretation of some results difficult. Please see below my comments and hopefully they will be useful.

Reply: We thank the expert reviewer for their sincere evaluation of our manuscript. In response to the concerns with respect to effect and cohort sizes we have now recruited three additional subjects for the ON STN-DBS condition, leading to a new cohort size of n=12, which exceeds many previous impactful studies such as the ones referenced with respect to the low/high beta comment below. The additional subjects corroborated our findings, and no qualitative changes had to be made with respect to effects or conclusions.

1. About the conclusion that DBS and dopamine have different mesoscale effects, especially within the STN (reported in Figure 2): I'm wondering how much this is due to the difference in the sample sizes for the Levodopa and DBS conditions. Only half of the all eighteen patients received stimulation (i.e., N = 18 and N = 9 for Levodopa and DBS, respectively). Because of this, the baseline in terms of the power modulation in Fig 2 and the modulation of cortico-subthalamic coupling in Fig. 3 during OFF therapy were different for different experimental conditions. This different baseline levels could potentially affect the statistic especially giving the relatively small sample sizes. For instance, there was a more prominent peak at high beta band in the STN LFP in STN-DBS condition compared with Levodopa condition (Fig. 2A). Similarly, there was a more prominent peak at alpha band in the cortex-STN coupling in Levodopa condition compared with STN-DBS condition (Fig. 3B). To alleviate this concern, an extra analysis only considering those participated with both Levodopa and DBS recordings (N = 9) would be helpful. One-way ANOVA with three conditions (No treatment, ON

Medication, ON DBS) can be used for these 9 participants, applying to the low beta and high beta frequency band separately, to see if there is really a difference between different treatment states.

Reply: We have now followed the reviewer's suggestion and repeated the statistical analysis OFF therapy vs. ON levodopa and OFF therapy vs. ON STN-DBS with only those 9 subjects in both medication and stimulation cohorts. This did not lead to a qualitative change in the results, as we could replicate the significant reduction in cortex high beta power with dopamine; a significant reduction in STN low beta with dopamine; and a significant reduction in STN high beta with stimulation. We have included a figure of these results in the *Supplementary Information*.

Supplementary Figure 2: Cortical and subthalamic spectral power in those patients with both OFF therapy – ON levodopa and – ON STN-DBS recordings. There are distinct modulations in the grand average power spectra across all bipolar contacts in the cortex and STN OFF therapy, ON levodopa, and ON STN-DBS. These modulations strongly resemble those in the wider cohorts (Fig. 2). Shaded coloured areas show standard error of the mean. Shaded light grey areas indicate a significant difference in the average values of canonical frequency bands between conditions. Shaded dark grey areas indicate clusters of significant differences between conditions for the respective frequency bins. * $p < 0.05$. Abbreviations: DBS – deep brain stimulation; STN – subthalamic nucleus.

Nevertheless, we further acknowledge the potential effect of differences in subjects in the *Results and Discussion* sections of the manuscript.

Changes in manuscript: The *Results* section now reads: “Finally, for the possibility that the distinct therapeutic mesoscale effects were influenced by differences between the subjects in the medication and stimulation cohorts, we re-analysed the data from only those 9 subjects with both ON levodopa and ON STN-DBS recordings. Permutation tests again showed the significant reductions in cortex high beta power and STN low beta power with dopamine, and

the significant reduction in STN high beta power with DBS, as described for the full cohort above (Supplementary Fig. 2).”

The *Discussion* section now reads: “Third, it is possible that the distinct mesoscale therapeutic effects reflect differences between subjects in the larger medication and smaller stimulation cohorts. However, a control analysis using only subjects that have undergone both medication and stimulation conditions reproduced the distinct effects of dopamine and DBS on low vs. high beta activity.”

2. I also feel uncomfortable about the conclusion that DBS only reduces high beta but not low beta in the STN. This is contradictory to multiple previous studies showing that DBS can reduce average power or peaks in a broad frequency band including the low beta frequency band, such as: <https://pubmed.ncbi.nlm.nih.gov/38580641/>; <https://pubmed.ncbi.nlm.nih.gov/22787444/>; <https://pubmed.ncbi.nlm.nih.gov/16623853/>. The results reported in this study can be compromised due to a few reasons: 1) the small sample size of 9 for DBS condition. ...

Reply: We recognise the reviewer’s expertise on the matter and their apprehension in our conclusion that DBS selectively reduces high beta power in STN. We also acknowledge that in the context of previous literature, it is not correct to state that DBS only affects high beta power. We therefore discuss this matter more cautiously in the present version of the manuscript.

Nevertheless, we would like to ask the reviewer to keep an open mind about the potential that indeed there is a stronger modulation of high beta vs. low beta, even in the cited studies. For example, in the cohort of 7 PD patients reported in Mathiopoulou *et al.* (2024, DOI: [10.1038/s41531-024-00693-3](https://doi.org/10.1038/s41531-024-00693-3)), there is clearly more residual low beta in the M0-S1 condition when compared to the M1-S0 condition (Fig. 1c). A similarly dominant high beta effect is visible, despite not being explicitly distinguished in Fig. 1d of Kehnemouyi *et al.* (2021, DOI: [10.1093/brain/awaa394](https://doi.org/10.1093/brain/awaa394)), perhaps the most complete and impactful STN-DBS neurophysiology study featuring 10 subjects. Even in in Whitmer *et al.* (2012, DOI: [10.3389/fnhum.2012.00155](https://doi.org/10.3389/fnhum.2012.00155)) Fig. 5b, the effect of DBS is stronger 20-30 Hz than 13-20 Hz.

Additionally, in the referenced review article of Eusebio *et al.* (2012, DOI: [10.3389/fnint.2012.00047](https://doi.org/10.3389/fnint.2012.00047)), no categorical distinction is made between low and high beta, and while an example of broader beta suppression is shown in Fig. 1b, the research article from which this was adapted (Eusebio *et al.*, 2011; DOI: [10.1136/jnnp.2010.217489](https://doi.org/10.1136/jnnp.2010.217489)) presents the selective suppression of high beta power as an example of DBS effects (Fig. 1):

Before highlighting our adjustments to the manuscript to acknowledge the discrepancy with the broader effects reported previously, we would like to point out that: a) most of the above-mentioned studies had similar sample sizes; and b) none of the studies explicitly addressed the distinction of low vs. high beta statistically. Notably, after including the additional 3 subjects (now n=12) with dedicated OFF therapy and ON STN-DBS recordings for the revision, we could replicate the same effects, with a significant reduction in subthalamic high beta power with DBS ($p < 0.05$), but no significant reduction in low beta power ($p > 0.05$). Nevertheless, we have now adapted the discussion to acknowledge the uncertainty as to whether previous studies would have observed a similar distinction if the statistical analysis would have been performed for low vs. high beta bands separately.

Changes in manuscript: The *Discussion* section now reads: “It is further noteworthy that our study is the first to report a frequency-specific high beta power modulation with STN-DBS. Whilst most previous studies have performed statistics solely on a broad 13-30 Hz range which did not allow for a more precise spectral distinction, some studies show significant modulations that extend into the low beta band^{9,50}. Nevertheless, even these previous studies show the strongest modulation to occur at or above 20 Hz⁵⁰, and feature stronger residual low beta ON STN-DBS when compared to ON levodopa⁹.”

3. ... 2) maybe most of the sampled participants only had peaks in the high frequency band in the PSD of the STN LFPs. Despite this, in Subject EL008, there seem to be a peak in the low beta band (<20 Hz) in the STN PSD. In this patient, is this low beta peak suppressed by DBS? Would it be possible to separate the hemispheres with peaks in the low beta band vs. those vs peaks in the high beta band, and see if it is true that only high beta activities are modulated by DBS?

Reply: The reviewer raises a valid point that the observed lack of modulation in low beta STN power could be influenced by the dominant beta frequency observed in the subjects at hand. Regarding the reviewer’s request for Subject EL008 who demonstrated a strong low beta peak, this subject was unfortunately not recorded in the ON STN-DBS condition. Nonetheless, to address the reviewer’s latter point, we isolated those channels with peaks in the low beta band (5.0 ± 0.8 channels per subject; mean \pm SEM) and those with peaks in the high beta band (5.3 ± 0.9 channels per subject) based on visual inspection of the OFF therapy power spectra. Both groups had data from all subjects. We also note that many channels had peaks in both the low and high beta bands and were thus assigned to both groups, leading to some degree of low and high beta activity being present in the opposing group. The distinction of low and high beta peaks was corroborated by a statistical analysis which showed that the low beta peak group had a cluster of significantly greater power in the low beta band (12.5-16 Hz; $p < 0.05$, cluster-corrected), while channels in the high beta group had significantly greater power in the canonical high beta band ($p < 0.05$). When examining the effects of DBS, we found no significant reduction in low beta power, including in those channels with low beta peaks ($p > 0.05$). In contrast, a significant reduction in high beta power was found in both groups, regardless of low or high beta dominance ($p < 0.05$). Altogether, this demonstrates that the lack of observed suppression in STN low beta power with DBS was not due to an absence of low beta activity. We have included a figure of these results in the *Supplementary Information*.

Supplementary Figure 1: Selection of STN-LFP channels for peaks in low and high beta band power and the effects of DBS. STN-LFP channels were grouped according to the presence of peaks in power in the low and high beta bands, based on visual inspection of the OFF therapy state. Many channels shared peaks in both the low and high beta bands, and as such were assigned to both groups, leading to some degree of low and high beta activity being present in the opposing groups. **a** The presence of peaks in power was corroborated by a statistical analysis, which showed that the low beta peak group had significantly greater power in the low beta band (12.5-16 Hz; $p < 0.05$, cluster-corrected), while the high beta peak group had significantly greater power in the canonical high beta band ($p < 0.05$). **b** With DBS, no significant reduction in low beta power was found, even in those channels with low beta peaks ($p > 0.05$), whereas significant reductions in high beta power were found in those channels with predominantly low and high beta peaks ($p < 0.05$). * $p < 0.05$. Abbreviations: DBS – deep brain stimulation; n.s. – not significant; STN – subthalamic nucleus.

Changes in manuscript: The following has been added to the *Results* section: “A significant suppression of STN low beta power with DBS was not observed ($p > 0.05$), potentially reflecting an absence of peaks in the low beta band. However, when isolating those STN-LFP channels with peaks in the low beta band, there remained no significant suppression in low beta power with DBS ($p > 0.05$; Supplementary Fig. 1). In contrast, channels with predominantly low beta and predominantly high beta peaks showed a significant suppression of high beta power with DBS ($p < 0.05$).”

4. ... 3) the dose of the DBS. The effect of the DBS changes with stimulation intensity. Higher stimulation intensity induced more changes in beta across a wider frequency band (<https://pubmed.ncbi.nlm.nih.gov/22675296/>) – Did the stimulation intensity used during the recording induce changes in the UPDRS scores? How does the stimulation intensity used in the test compared with intensity used for clinical stimulation?

Reply: We recognise that the spectral effects of DBS can be influenced by stimulation intensity. We note that the stimulation amplitudes used during the recordings closely resembled those used for chronic DBS determined during a 3-month follow-up clinical review, in fact

tending to be slightly higher during the recording (average across subjects during recordings – 2.4 ± 0.1 mA; average at a 3-month follow-up – 1.9 ± 0.2 mA; mean \pm SEM) . This information is now included in Supplementary Tables 1 and 2. The other stimulation settings during the recordings (130 Hz frequency and 60 μ s pulse width) matched those used in the chronic DBS setting. Notably, this contrasts our study to the cited paper of Whitmer *et al.* (2012, DOI: [10.3389/fnhum.2012.00155](https://doi.org/10.3389/fnhum.2012.00155)), which used a stimulation frequency of 185 Hz and a pulse width of 90 μ s. This difference in stimulation frequency and pulse width leads to a dramatic increase in the total energy delivered. Stimulation settings can vary across clinical centres, however our reported stimulation settings reflect chronic clinical settings in this specific cohort, as well as the general approach in our clinic, more closely than the settings reported in Whitmer *et al.* (2012). Unfortunately, the paper does not mention how the used stimulation settings reflected the chronic settings in their cohort. Despite this, the reviewer hints to a relevant point: the fact that an objective measure of acute clinical effects with stimulation (e.g., UPDRS scores) was not obtained in our study. Instead, the stimulation effect was determined by a monopolar review, focusing on anti-bradykinetic effects through finger tapping. This contrasts with other impactful studies (e.g., Kehnemouyi *et al.*, 2021, DOI: [10.1093/brain/awaa394](https://doi.org/10.1093/brain/awaa394)), obviating a more direct comparison of DBS mechanisms and clinical effects.

Changes in manuscript: We now discuss these points in the limitations portion of the Discussion section:

“Second, in contrast to other studies, an objective measure of symptom alleviation during DBS was not performed, obviating a more direct comparison between DBS parameters and clinical effects⁵⁰.”

“The observed stimulation effects may therefore not fully represent those following a more extensive parameter review in the chronic DBS state, although similar settings were used in a 3-month follow up clinical review (see Supplementary Tables 1 and 2).”

5. ... 4) How the power is normalised. If the power is normalised against the total power within the 1-45 Hz band, a reduction in the higher beta band may lead to a relative increase in the alpha or low beta band. So it is actually difficult to interpret the results. In addition, this normalisation may also mask the potential effect of DBS in reducing the total power of this broad frequency band.

Reply: We believe there was a misunderstanding, and we apologise for the lack of clarity in our presentation of the power normalisation procedure. Normalisation against total power was performed using the 5-60 Hz range. The upper bound of 60 Hz was chosen to prevent possible contamination from residual DBS artefacts remaining following the artefact removal procedure. We also accept the reviewers point that suppression of power in the higher frequencies of this range could cause a relative increase in lower frequency power. To this end, we performed the analyses again using a broader normalisation window (5-95 Hz) for which any changes in high beta power would have less possible influence on low beta power. Our initial effects were reproduced, with a suppression of STN power in the high ($p < 0.05$), but not low ($p > 0.05$) beta band.

Changes in manuscript: The following additions have been made to the *Methods* section: “Power was calculated using multitapers (5 Hz bandwidth), normalised to percentage total power¹¹⁴ (from 5-60 Hz with a line noise exclusion window at 45-55 Hz), and averaged across epochs to give a single spectrum per channel, per recording. An upper bound of 60 Hz was chosen for the normalisation to prevent possible contamination from residual stimulation artefacts following the artefact removal procedure. Given that there is a risk of changes in any one frequency band affecting power in other frequencies for small normalisation windows, a broader 5-95 Hz normalisation range was additionally performed, however this did not meaningfully alter the interpretation of the results.”

6. ... 3.) I also feel uncomfortable about the conclusion that DBS does not change local activities in the cortex. Whitmer et al. 2012 (<https://pubmed.ncbi.nlm.nih.gov/22675296/>) who recorded cortical activity with ECoG during DBS showed that cortical signals over the estimated origin of the hyperdirect pathway (estimated using DTI) did demonstrated attenuation of beta hypersynchrony during DBS delivered dorsal to or within STN, whereas signals from non-specific regions of motor cortex were not attenuated.

Reply: We recognise the reviewer’s apprehension over our conclusion that DBS does not modulate cortical power. Whitmer *et al.* (2012, DOI: [10.3389/fnhum.2012.00155](https://doi.org/10.3389/fnhum.2012.00155)) report ECoG data from 3 patients, of which data from 1 patient was discounted because of aberrant targeting. The small sample size obviated any group statistic, but in some of the contacts of those 2 patients, beta modulation was visible. Notably, in alignment with our results, another study by de Hemptinne *et al.* (2015, DOI: [10.1038/nn.3997](https://doi.org/10.1038/nn.3997)) reported no consistent cortical beta power modulation across a cohort of 23 patients. Perhaps this discrepancy may again be explainable by the difference in stimulation parameters, as de Hemptinne *et al.* used 60 μ s pulse widths and stimulation frequencies below 185 Hz, according to optimal clinical settings. We have made changes to address this point accordingly.

Changes in manuscript: The *Discussion* section now reads: “We note that one previous study investigating the effects of STN-DBS using ECoG recordings described an ameliorating effect of stimulation on motor cortex beta power in two patients⁷⁶, however this discrepancy may be explained by the use of higher stimulation energies (185 Hz frequency and 90 μ s pulse width).”

7. ... For the position of the ECoG in this study, it was mentioned that the ECoG strip was aimed posteriorly towards the hand knob region of the motor cortex. But the actual location of the ECoG contacts shown in Figure 2A was very variable across participant. How many participants actually have ECoG contacts over the ‘hand knob region’?

Reply: There is indeed variation in the ECoG electrode positions, owing to the patient-specific planning of burr hole locations. Using a highly conservative estimate of the hand-knob region as a single MNI coordinate (x=36, y=-19, z=73), we note that the closest contact per subject was a distance of 12.3 \pm 1.6 mm (mean \pm SEM) from this coordinate. This information

has now been added to the *Methods* section. However, we would like to clarify that the hand knob itself simply served as a rough orientation, and we did not pose any specific hypotheses based on this target. The hyperdirect pathway connects the entire prefrontal and motor cortex with the STN, and the anatomical variation allowed us to investigate the spatial specificity of our findings.

Changes in manuscript: The *Methods* section now reads: “The ECoG strip was aimed posteriorly toward the hand knob region of the motor cortex. Defining hand-knob region to be a single MNI coordinate (x: 36, y: -19, z: 73), the closest ECoG contact per subject had an average distance of 12.3 ± 1.6 mm (mean \pm SEM) to this coordinate.”

8. ... How many contact showed a high beta peak in the PSD? Is there any contact on the origin of the hyperdirect pathway? Would the difference in the effect of DBS vs medication reported here actually be the ‘spatial specificity’ of the effect? Would it be possible that the effect of DBS on cortex is more focused on specific area, whereas the effect of dopamine is more wide spread? Would the algorithm used in the study (quantifying the grand average) capture this potential difference? Maybe the author should re-run the analysis focusing on the ECoG contacts on the hyperdirect pathway vs. those on other areas?

Reply: Across the cohort, there was an average of $60 \pm 6\%$ (mean \pm SEM) of ECoG channels with a peak in high beta power, as determined by visual inspection, with all subjects having at least one channel with a peak. We appreciate the possibility that DBS effects may be more spatially specific than the effect of dopaminergic medication. Notably, the hyperdirect pathway originates from layer 5 neurons of motor cortex and more frontal areas, as delineated in the anatomical Petersen fibre atlas (Petersen *et al.*, 2019, DOI: [10.1016/j.neuron.2019.09.030](https://doi.org/10.1016/j.neuron.2019.09.030)) used in our study. Therefore, all contacts over motor cortex spatially coincide with regions of hyperdirect pathway origin, and all patients had at least one contact over motor cortex.

For these motor cortex channels, the presence of high beta peaks rose to $83 \pm 8\%$. The therapeutic modulation in these motor cortex channels reproduced our previous findings, with a suppression of high beta power with medication ($p < 0.05$), but not DBS ($p > 0.05$). As for the possibility of the results being explained by a more spatially specific DBS effect, we recognise this as a potential explanation. However, the above analysis of motor cortex power corresponds to only 1.8 ± 0.2 bipolar channels per subject, limiting the extent to which averaging results over channels could mask a difference in the spatial specificity of modulatory effects between medication and stimulation.

Further evidence against a difference in the spatial specificity of therapeutic effects comes from the spatio-spectral decomposition analysis, a multivariate technique where a set of spatial patterns is produced describing the degree to which a given contact contributes to band power. The average of these patterns over medication states is already presented in Fig. 2b. However, we can also examine the patterns for each condition separately. This figure has now been added to the *Supplementary Information*, and it shows that the spatial distributions of high beta power are strikingly similar OFF therapy, ON levodopa, and ON STN-DBS, being localised most strongly to motor cortex:

Supplementary Figure 3: Spatial maps of cortical high beta power. Localisation of the strongest power component extracted with spatio-spectral decomposition for cortical high beta. Localisations are highly similar across therapeutic states for the **(a)** OFF therapy – ON levodopa cohort and **(b)** OFF therapy – ON STN-DBS cohort. Localisations are shown for individual electrodes and interpolated to surfaces. Abbreviations: A – anterior; ECoG – electrocorticography; L – left; P – posterior; R – right.

Changes in manuscript: We describe this finding in the Results as follows:

“DBS did not modulate cortical power spectra, even when selecting only contacts over motor cortex, ...”

“Furthermore, motor cortex high beta localisations showed a striking similarity across medication and stimulation states, suggesting that the lack of modulation of grand average cortical high beta with DBS was not due to a more spatially specific suppression of activity compared to with dopamine (Supplementary Fig. 3).”

9. There is lack of details in the methods for results presented in Figure 2: 1) What is ‘grand average power spectra’ as reported in Fig 2A? Are they the average across all bipolar contacts for different regions? Or are they the most significant components?

Reply: We apologise for the lack of clarity in the results of Fig. 2. The grand average power spectra of Fig. 2a represent the average across bipolar contacts.

Changes in manuscript: We have made the following addition to the figure legend to clarify this: “There are distinct modulations in the grand average power spectra across all bipolar contacts in the cortex and STN...”

10. ... 2) All patients received bilateral STN implantation, but the sample size is 9, which the patient number for the effect of DBS. How did the authors deal with the two hemispheres for each participants?

Reply: We apologise for the lack of clarity in the description of our approach. We focused our analyses on the hemisphere ipsilateral to the ECoG strip to ensure comparability between the power and oscillatory connectivity analyses. Notably, the sample size for STN-DBS is now 12 due to the additional subjects added for the revision.

Changes in manuscript: We have made the following addition to the *Methods* section to clarify this fact: “We considered only STN-LFP recordings from the hemisphere ipsilateral to the ECoG strip.”

11. ... 3) How power is normalised? What is the frequency band considered as total?

Reply: We thank the reviewer for highlighting this important lack of clarity, which we have addressed in response to comment 5 above.

Changes in manuscript: The following additions have been made to the *Methods* section: “Power was calculated using multitapers (5 Hz bandwidth), normalised to percentage total power¹¹⁴ (from 5-60 Hz with a line noise exclusion window at 45-55 Hz), and averaged across epochs to give a single spectrum per channel, per recording. An upper bound of 60 Hz was chosen for the normalisation to prevent possible contamination from residual stimulation artefacts following the artefact removal procedure. Given that there is a risk of changes in any one frequency band affecting power in other frequencies for small normalisation windows, a broader 5-95 Hz normalisation range was additionally performed, however this did not meaningfully alter the interpretation of the results.”

12. ... 4) In Fig. 2C, it seems the contributions to power for different contacts in the same level were quantified and mapped in MNI space. I was just wondering how the directionality of the DBS lead was identified, and how reliable this information is.

Reply: Following reconstruction of the DBS electrodes in Lead-DBS using the PaCER algorithm (Husch *et al.*, 2018, DOI: [10.1016/j.nicl.2017.10.004](https://doi.org/10.1016/j.nicl.2017.10.004)), rotation of the DBS electrode was manually corrected for using artefacts of electrode orientation markers with the DiODE v2 algorithm (Dembek *et al.*, 2021, DOI: [10.3390/brainsci11111450](https://doi.org/10.3390/brainsci11111450)). This allowed us to accurately determine the particular orientation of individual segmented contacts on the DBS leads. While we are not aware of a post-mortem validation of this algorithm it is deemed precise with respect to the identification of neuroradiological information with 100% accuracy. We now describe this procedure in more detail.

Changes in manuscript: We have made the following addition to the *Methods* section to clarify this procedure: “ECoG contact artefacts were marked manually and MNI coordinates were extracted. DBS electrodes were reconstructed using the PaCER⁹⁹ algorithm and manually refined, with DBS electrode rotation being manually corrected using artefacts of electrode orientation markers with the DiODE v2¹⁰⁰ algorithm. In brief, the postoperative CT slice showing the most visible marker artefact in the electrode was first selected, and the marker of the reconstructed electrode model was then rotated to align with this. MNI coordinates were then extracted for the DBS lead contacts.”

13. ... 5) For the results on the time delays suggesting mono- and poly-synaptic pathway communication (results reported in Fig 3), I’m wondering how the results may be impacted by the selection of patients for analysis. The authors mentioned, “4 subjects had no positive time delays which met the confidence criteria, and were thus excluded from the time delay analysis.” (Last sentence of Page 23). This further reduced the sample sizes in both experimental conditions. I was wondering if the authors could explain a bit more regarding why there were 4 out of 18 subjects (22.22%) did not meet the criteria.

Reply: We apologise that this was not discussed in more detail. In short, these criteria act as an additional statistical check for the quality of results, allowing us to identify contacts with excessive noise that threaten the validity of the time delay analysis. In response to the excellent suggestions by reviewer 3, we have increased the number of bootstraps used to generate this confidence interval (was 200, now 400). With this change, there is only 1 subject (of the now 21 subjects) that does not meet the criteria. Please see also the replies to reviewer 3 below for a more detailed discussion of the methodology.

Changes in manuscript: We have made the following additions in the *Discussion* section to clarify and discuss this point in more detail: “Regarding the time delay analysis, we note that we were unable to adequately estimate time delays in 1 subject due to excess noise in the recordings, as determined by a bootstrap-based confidence criterion. Although the bispectrum shows a greater – if not complete – resilience to Gaussian noise compared to traditional cross-correlation analysis²³, it can nonetheless be contaminated by non-Gaussian noise, with the estimation of time delays requiring a greater signal-to-noise ratio compared to methods such as spectral power and coupling. Overall, we were still able to determine time delays for 20 subjects.”

The following additions were also made to the *Methods* section:

“As an additional sanity check for the quality of time delay analysis results, the same epoch sampling procedure was employed, with the exception that epochs were sampled 400 times. Time delay estimates were computed for each of these 400 segments, and a confidence interval-based criterium used to exclude excessively noisy data from the time delay results (see *Time delay analysis*).”

“Connections for which $t = 0$ ms falls within the confidence interval reflects an uncertain time delay estimate that is highly sensitive to the sampled epochs, an indication of excessive noise in the data. These uncertain connections were excluded from the time delay results, acting as an additional statistical barrier to ensure the quality of the estimates.”

14. ... In addition, here only those connections with positive tau (motor cortex to STN) were analysed, would this selection bias the results toward the hypothesis and lead to type 1 error? How about the connections with negative tau that representing information flow from STN to motor cortex? Why they should be excluded from the statistics?

Reply: The reviewer raises a valid point. Given that the Granger causality analysis demonstrated that cortex acts as the driver of communication with STN, we have deemed it appropriate to place a special focus on those individual connections exhibiting this property in the time delay analysis. In the context of the tau values (the time point corresponding to the strongest time delay estimate), given that this is determined from the global maximum, it is critical that we assess directed cortex to STN communication from these appropriate connections. However, we do accept that for the time delay results presented in Fig. 3d in which the delays across a broader period are considered (not only at the global maximum as for tau), the criterion for connections exhibiting a positive tau is less essential. Accordingly, we have re-analysed the time delay results in Fig. 3d without this requirement, which does not meaningfully alter the interpretation of results. That is, motor cortex – STN communication remains significant below 10 ms OFF therapy, but is not significant with dopamine or DBS. Additionally, there remains a significant number of local peaks in the 1-9 ms window OFF therapy, but not with dopamine or DBS.

15. ... 6. Following the previous comment, on Page 9, the authors mentioned that, ‘in addition to suggesting a combination of mono- and poly-synaptic cortico-subthalamic communication, these results further highlight the suppression of hyperdirect pathway activity as a shared therapeutic mechanism of dopamine and DBS’. 1) I guess this is based on the statistics shown in Fig. 3D. However, it was not clear to me how the statistics were conducted. Is there any difference in the ‘Time delay estimate strength’ caused the treatment (medication or DBS)?

Reply: We apologise for the lack of clarity in our description of the time delay results. We have now re-written the time delay portion of the *Results* section to better describe our findings, including changes to the concluding paragraph as well as the legend of Fig. 3. For the question of whether there is a difference in the degree of modulation, we compared the

effect for those 9 subjects with both medication and stimulation recordings. Although there was a slightly larger suppression of communication strength with DBS than with dopamine, this was not objectifiable statistically, and no significant clusters of suppression emerged ($p > 0.05$). We have added a description of this new analysis to the text.

Changes in manuscript: The *Results* section now reads:

“In those subjects with data from both medication and stimulation states ($n = 9$), no significant differences in the degree of modulation of time delay estimate strength between dopamine and DBS were found ($p > 0.05$, cluster-corrected).”

“Therefore, in addition to showing cortico-subthalamic communication in time windows congruent with indirect and hyperdirect pathway activity, these results further highlight the suppression of hyperdirect pathway communication as a shared therapeutic mechanism of dopamine and DBS.”

The legend for Fig. 3d now reads: “Bispectral time delay analysis highlights the contributions of mono- and poly-synaptic pathways to cortico-subthalamic communication, and the shared therapeutic suppression of monosynaptic pathway communication. Upper plots show the strength of grand average time delay estimates for each time bin, with opaque lines representing estimates which are significantly greater than the physiological control of parietal cortex – STN communication. Bottom plots show the number of peaks in each connection of the time delay estimates aggregated over 10 ms windows, with significance again determined against the physiological control of parietal cortex – STN communication.”

16. ... The stars indicating statistical significance were indicated on the ‘peak count (% maximal)’ plots. What does this ‘peak count (% maximal)’ mean? How was it calculated? What is the sample size of the statistics? Were the test controlled for multiple comparison correction?

Reply: We apologise for the lack of clarity in our presentation of the time delay results. Here, peak count refers to the number of peaks (local maxima) of time delay estimate strength occurring in each 10 ms window. This is now clarified in the *Methods* section. Furthermore, we have updated the legend of Fig. 3 to make these results clearer. The sample size of the statistics are $n=17$ (OFF therapy – ON levodopa comparison), and $n=12$ (OFF therapy – ON STN-DBS comparison), which we now mark on Fig. 3d. Finally, the tests were controlled for multiple comparisons using Bonferroni correction, which we now mention alongside the results.

Changes in manuscript: We have made the following addition to the *Methods* section describing how this measure was calculated: “In addition to analysing tau (the global maximum across the estimates for all times), peaks (local maxima) in discrete time windows were also analysed. Treating the estimate strength at $t = 0$ ms as a baseline, local maxima above this threshold and at least 5 ms apart from one another were identified and grouped into 10 ms bins (apart from the first bin of 1-9 ms where $t = 0$ ms was excluded). Peak counts were normalised to percentages to account for variation in the number of cortex – STN connections across subjects and cortical regions.”

The *Results* section now reads:

“... a significant number of local peaks occurred in the 1-9 ms window OFF therapy ($p < 0.05$, Bonferroni-corrected; time of peaks 5.2 ± 0.3 ms)...”

“... nor were there a significant number of local peaks in the 1-9 ms window (both $p > 0.05$, Bonferroni-corrected)...”

The legend for Fig. 3d now reads: “Bispectral time delay analysis highlights the contributions of mono- and poly-synaptic pathways to cortico-subthalamic communication, and the shared therapeutic suppression of monosynaptic pathway communication. Upper plots show the strength of grand average time delay estimates for each time bin, with opaque lines representing estimates which are significantly greater than the physiological control of parietal cortex – STN communication. Bottom plots show the number of peaks in each connection of the time delay estimates aggregated over 10 ms windows, with significance again determined against the physiological control of parietal cortex – STN communication.”

17. ... 2) Although significant, the difference in the time delay estimate strength from motor cortex to STN at 1-9 ms window (i.e., first bin) between Levodopa therapeutic conditions was pretty small. Considering this is one of the main results supporting the mono- and poly-synaptic argument, I was wondering apart from theoretical interpretation with the support of existing studies about hyperdirect pathway, if there is any other evidence to show that this measurement is actually clinically meaningful. It was just because the clinical meaning of beta power and cortex-STN coupling was relatively clearer, but not the strength of time delay estimation used here.

Reply: We recognise the reviewer’s point that difference in time delay estimate strength with dopamine was somewhat small, however we respectfully disagree that this threatens the argument of mono- (hyperdirect pathway) versus poly-synaptic (indirect pathway) communication. For one, this result is directly supplemented by the observation that a significant number of peaks (local maxima) are observed in the 1-9 ms window congruent with monosynaptic, hyperdirect pathway activity only for OFF therapy and not ON levodopa. Furthermore, we have purposefully chosen a multi-faceted approach to corroborate the findings of any one modality or analysis technique. To this end, we show also that cortico-subthalamic high beta coupling is suppressed with dopamine, a spectral signature which has been previously linked to hyperdirect pathway activity, something we confirm in our own data through comparisons with dMRI-based fibre tracking.

18. The results on the link between the connectivity of the low beta and high beta with the indirect and hyperdirect pathway, respectively, are really nice. Can these be further confirmed by the time delay analyses? Would the high beta band coupling have more peaks in the 1-9 ms window, and the low beta coupling have more spread time delays or longer time delays? This will make the time delay analyses more convincing.

Reply: This is an excellent idea, which is also an analysis we have attempted. Unfortunately, it is a technical limitation of the method that when analysing delays from individual frequency bands, the small number of coefficients from which to estimate delays produces results which are inaccurate and unreliable. For instance, we performed preliminary simulation work which showed a broadband range (e.g., the 3-100 Hz range used in the current analysis) was required for accurate estimates. Individual frequency bands also had a much higher proportion of estimates that failed to meet the confidence criterion. Additionally, limiting the number of coefficients introduces a strong degree of smoothing in the estimates over time, such that any local maxima are lost. Therefore, we are not confident that estimating time delays for the low and high beta bands would produce valid estimates. Ultimately, we have weighed this limitation in the context of the wider benefits of a bispectrum-based time delay method against other approaches, such as the superior resilience to Gaussian noise, offering more accurate estimates of time delays given an adequately large frequency range.

Changes in manuscript: We now discuss this limitation in the *Discussion* section: “Additionally, we limited our analyses to the broadband 3-100 Hz range, which prevented identifying time delays associated with the low and high beta bands specifically. This decision was informed by simulations which showed time delay estimates from individual frequency bands to have a limited accuracy and ability to meet the confidence criterion.”

19. ...8. Fig. 4, it seems not very clear to me how the estimated (y-axis) and empirical (x-axis, upper/left panel) were quantified. The stats showed that the estimated number of fibres were correlated with empirical number of fibres for both high and low beta coupling. I was just wondering why the statistic was applied to empirical number of fibres, rather than directly applied to the contribution to connectivity (lower/right panel), which was derived directly from the electrophysiological recordings.

Reply: Our apologies for the lack of clarity in Fig. 4. The R^2 and p-values in the plots comparing empirical vs. estimated fibre counts (upper plot of Fig. 4a inset) and empirical vs. estimated fMRI connectivity (left plot of Fig. 4a inset) represent a statistical measure of the quality of the linear mixed effects models. In contrast, the p-values reported in the main body of the text refer to the relationship between the contribution to connectivity derived from the electrophysiological recordings and the number of fibres (lower plot of Fig. 4a inset) and fMRI connectivity (right plot of Fig. 4b inset). We recognise this was a source of confusion and have now also included the p-values we report in text in the relevant plots, alongside an additional explanation in the figure legend of what the R^2 and p-values in each plot represent.

Changes in manuscript: (Fig. 4a legend) “The inset shows the relationship between the estimated number of hyperdirect pathway fibres from the linear mixed effects model versus the empirical number of fibres (top; conditional R^2 and p-value represents the quality of the linear mixed effects model) and the contributions to high beta oscillatory connectivity (bottom; p-value reflects that reported in text).”

(Fig. 4b legend) “Lower inset shows the relationship between the estimated functional connectivity from cortex to putamen, GPe, and STN versus the empirical functional connectivity to these regions (left; conditional R^2 and p-value represents the quality of the

linear mixed effects model), as well as the contributions of ECoG contacts to low beta connectivity (right; p-value reflects that reported in text).”

20. There is no direct support in the statement the authors made in the ‘Discussion’ (Page 12: ‘Altogether, we argue that excessive hyperdirect pathway communication between the cortex and STN is a central factor in PD pathology’. Is there any correlation between the connectivity in the hyperdirect pathway and UPDRS scores when OFF any treatment? Does the changes in the connectivity induced by treatment (medication or DBS) correlate with the effect of the treatment? Those analyses could support the statement the authors made, but was not presented.

Reply: We agree with the reviewer and have removed this sentence from the Discussion section.

21. What is the total duration of recording for each condition for each participant?

Reply: The recording durations are: OFF therapy 314±16 s; ON levodopa 351±37 s; and ON STN-DBS 273±24 s (mean±SEM). We have added information about the duration of recordings to the new Supplementary Table 2.

22. For the structural and functional connectivity analysis based on group connectomes, I was wondering if there is any guidance or rationale for the selection of radius (5 mm and 3 mm for ECoG and STN-LFP here, respectively) and how it would affect the results. I saw different values were used in existing studies, for example, a radius of 2 mm was used in Oswal et al. 2021.

Reply: The reviewer raises a valid point regarding the choice of radii for the d/fMRI connectivity analyses. We were unable to conclusively identify a range for the sensitivity of contacts to neural sources in the current literature, and instead resorted to what we believe are appropriate values based on the sizes of our electrode contacts, taking particular care to avoid overlapping spheres between ECoG contacts (as strong source-mixing in ECoG data is not assumed).

For Oswal *et al.* (2021; DOI: [10.1038/s41467-021-25366-0](https://doi.org/10.1038/s41467-021-25366-0)), we note that the fibre tracking values used were seed spheres of ~2 mm radius for STN-LFP contacts (the exact value was dependent on the DBS lead model used, but nonetheless very similar to our 3 mm radius) and 2 mm³ target voxels for the cortex. The resulting fibre density maps were subsequently smoothed with an 8 mm³ Gaussian kernel. It is therefore difficult to directly compare our chosen ECoG radius of 5 mm, where spheres rather than cubes were used, and no spatial smoothing was applied. Again, however, we accept that some may find our use of 5 mm radius spheres for ECoG contacts to be too large. On this point, we note that when using 3 mm radius spheres

for both ECoG and STN-LFP contacts, the selective association of high beta coupling – hyperdirect pathway fibre density remains (low beta, $\beta = 0.927$, $p > 0.05$; high beta, $\beta = 3.302$, $p < 0.05$), as well as the selective association of low beta coupling – indirect pathway fMRI connectivity (low beta, $\beta = 0.005$, $p < 0.05$; high beta, $\beta = 0.004$, $p > 0.05$).

Changes in manuscript: We now describe this choice in the *Methods* section: “The 5 mm radius for ECoG contacts was chosen to sample from a wide area whilst preventing an overlap between contacts. A smaller radius of 3 mm did not meaningfully alter the interpretation of the results.”

23. The last two plots in Figure 3D (the bottom right): how are they different from those presented in 3B?

Reply: These plots represent the maximised imaginary coherency spectra generated alongside the spatial topographies of Fig. 3a. We have updated the x-axis label of these plots to make this association clearer.

24. Fig. 5, it seems the results/plots about power, coherency, granger causality, and time delays shown here were not consistent with the results shown in previous figures. Were they just schematics rather than actual results?

Reply: These plots are indeed schematics rather than actual results, and we have updated the title of Fig. 5 to reflect this: “Schematic summary of therapeutic effects.”.

Reviewers 3 & 2

Reviewer 3

Reply: We are very thankful for your valuable input!

Reviewer 4

This is an interesting study on patients with Parkinson's Disease that underwent deep brain stimulation (DBS) surgery. The authors use a multimodal dataset (intraoperative LFP recordings and fMRI connectomes) to investigate the differences in cortico-subthalamic network changes induced by dopaminergic drug treatment vs DBS, even if their therapeutic effect are similar. Their findings suggest a difference at the mesoscale/LFP level, while they have similar effects at the macroscale (suppression of the hyperdirect pathway). It is also a great inclusion to specifically address these (lack of) findings in previous studies. I do believe the study is of high interest for the field, and by addressing the comments below would be a great addition to the literature. I do believe the authors will be able to address my points below without difficulty.

1. Overall, even if the authors specify it later, they seem to suggest that they have computed connectivity measures during both medication ON and medication OFF, and compared it with spectral features (e.g. from the abstract). However, that is not the case, as connectivity measures were based on an fMRI connectome, and compared with spectral (and other) features from the LFP-ECOG recordings. I would advise the authors to be clear with this analysis choice in the abstract, to facilitate the reader to understand what were the methodological choices taken in this study. This can be placed in the methods.

Reply: We apologise for the lack of clarity in our description of the combination of electrophysiological and connectomic results, and thank the reviewer for their suggestions to improve this. We have made several changes throughout the manuscript accordingly.

Changes in manuscript: We have made the following changes to the *Abstract*, *Introduction*, and *Results* sections:

“To address this critical knowledge gap, we combined fully invasive neural multisite recordings in patients undergoing DBS surgery with normative MRI-based whole-brain connectomics.”

“To overcome this hurdle, we developed a multimodal approach to compare the neural circuit effects of levodopa and DBS with fully invasive cortex – STN multisite intracranial EEG recordings in patients undergoing DBS electrode implantation for PD, combined with normative MRI-based whole-brain connectivity mapping.”

“Whilst atlas-based structural connectivity provides a straightforward estimate of connection probability for monosynaptic pathways, ...”

Additionally, we have added the following statements to the *Methods* section to facilitate the reader's understanding of the methodological choices:

"Atlas-based connectomes were used, as patient-specific dMRI scans were not performed."

"Normative connectomes derived from a large population were used, as patient-specific fMRI scans were not performed."

This is discussed further in the *Discussion* section:

"As a final noteworthy limitation, we would like to highlight that the functional and structural connectivity analyses relied on atlas data and group connectomes, as patient-specific fMRI and dMRI scans were not taken. It is thus unclear to what degree the connectomic findings relate to individual anatomy. On the other hand, this has the advantage that robust connectomes derived from large cohorts have been validated multiple times^{18,26,27}, with superior image quality when compared to individual clinical MRI scans."

2. Due to this point, I would invite the authors to revise this statement for clarity "Our study is the first to systematically compare the effects of dopamine and STN-DBS on local mesoscale and interregional macroscale circuit communication with fully-invasive neurophysiology and MRI connectomics in Parkinson's disease."

Reply: We thank the reviewer for their suggestion to improve the clarity of this statement, and have modified the introduction to the *Discussion* as follows:

Changes in manuscript: The *Discussion* section now reads: "Our study systematically compares the effects of dopamine and STN-DBS on local mesoscale and interregional macroscale circuit communication with fully invasive neurophysiology and normative MRI connectomics in Parkinson's disease."

3. While it is great to see multiple causality-related methods being deployed to validate their findings, it would be great to have the authors explain why they choose to have multiple causality-related methods (with coherency, Granger causality, time-delay), and what is the rationale behind their usage in different contexts (e.g. coherency is taken to just estimate undirected connectivity, while granger causality for directed connectivity).

Reply: We have followed the reviewer's suggestion and now include references to relevant publications where appropriate.

Changes in manuscript: The *Introduction* section now reads: "We characterised neurophysiological cortex – STN interactions using: coherency-based measures for undirected spectral coupling^{19,20}; Granger causality for directed spectral coupling, capturing information

about the direction of information flow^{21,22}; and bispectral time delay analysis for estimating latencies of information transfer²³.”

4. In “Oscillatory connectivity analysis”, it is unclear how the order 60 is chosen for Granger Causality. For example, in their cited work (22), the authors indicate that they have used a bootstrap-based testing for optimizing the order p (with BIC). Was a similar preliminary analysis conducted on the data to choose the model order? This is important as it can influence the causality relationship found by the method.

Reply: We apologise for the lack of clarity in the chosen order of the vector autoregressive models used in the Granger causality analysis. The reviewer rightly points out that strategies do exist for informing this hyperparameter choice, however it is for us primarily a matter of avoiding excessive smoothing of results in the spectral domain whilst maintaining an affordable computational demand (a higher model order results in less smoothing, but greater compute cost). For our chosen multitaper-based spectral representations, we found that a model order of 60 avoided excessive spectral smoothing, with diminishing returns beyond this point. We highlight the following example using our implementation of the state space-based Granger causality method for a demonstration of this effect: https://mne.tools/mne-connectivity/stable/auto_examples/granger_causality.html#controlling-spectral-smoothing-with-the-number-of-lags.

Changes in manuscript: We have updated the *Methods* section as follows to clarify our methodological choice: “Directed connectivity was quantified using a multivariate form of time-reversed Granger causality^{21,22} with a vector autoregressive model order of 60 to avoid excessive smoothing of Granger scores across frequencies.”

5. In Time delay analysis (Methods), the authors mention to use “Using the 200 bootstrapped segments, 80% confidence intervals were computed for each connection from the segment taus”. I would suggest the authors to estimate the underlying distribution to potentially increase statistical insights, or have larger number of bootstraps.

Reply: We thank the reviewer for this suggestion to improve the robustness of the time delay results, and have doubled the number of bootstraps from 200 to 400. In addition to treating parietal cortex – STN interactions as a conservative physiological control (where less communication is expected) to secure the neuroanatomical specificity of our results, we are confident that this will improve the robustness of the time delay analysis.

6. ... At the same time, they indicate “4 subjects had no positive time delays which met the confidence criteria”. I would suggest to 1) for clarity to the reader, rephrase as “4

subjects out of ...". And 2) discuss the fact that 4 subjects showed no time delays, compared to other patients.

Reply: Thankfully, using the increased number of bootstraps (400 instead of the previous 200), there is now only 1 subject (of the now 21 subjects) that does not meet the criteria. Furthermore, we now better describe this procedure and discuss its relevance in more detail.

Changes in manuscript: We have made the following additions in the *Discussion* section to clarify and discuss this point in more detail: "Regarding the time delay analysis, we note that we were unable to adequately estimate time delays in 1 subject due to excess noise in the recordings, as determined by a bootstrap-based confidence criterion. Although the bispectrum shows a greater – if not complete – resilience to Gaussian noise compared to traditional cross-correlation analysis²³, it can nonetheless be contaminated by non-Gaussian noise, with the estimation of time delays requiring a greater signal-to-noise ratio compared to methods such as spectral power and coupling. Overall, we were still able to determine time delays for 20 subjects."

The following additions were also made to the *Methods* section:

"As an additional sanity check for the quality of time delay analysis results, the same epoch sampling procedure was employed, with the exception that epochs were sampled 400 times. Time delay estimates were computed for each of these 400 segments, and a confidence interval-based criterium used to exclude excessively noisy data from the time delay results (see *Time delay analysis*)."

"Connections for which $t = 0$ ms falls within the confidence interval reflects an uncertain time delay estimate that is highly sensitive to the sampled epochs, an indication of excessive noise in the data. These uncertain connections were excluded from the time delay results, acting as an additional statistical barrier to ensure the quality of the estimates."

7. In the main text, in the introductory paragraph of the Result section, it is mentioned that ECoG strips were used for recordings perioperatively. However in the limitations it is mentioned that "all recordings were performed a few days following implantation of the DBS electrodes in the STN". Were the recordings made during surgery (as I imagine ECoG strips were temporarily placed), or ECoGs were part of a chronic implantation and therefore recordings were made few days later (and if so, how many (mean and std)? Considering the type of recordings (medication ON/OFF, DBS ON/OFF), I imagine it is the latter, due to time required.

Reply: We apologise to the reviewer for the lack of clarity. It is indeed the case that the ECoG strip was implanted in the same surgery as the DBS electrodes, and retracted 5-7 days later in during the IPG implantation surgery. The recordings were taken in the days between these two surgeries: OFF therapy 3.9 ± 0.3 days; ON levodopa 3.7 ± 0.4 days; ON STN-DBS 4.4 ± 0.4 days (mean \pm SEM). The time of recordings post-implantation are included in the new Supplementary Table 2, and we have updated the introduction to the *Results* section to clarify this as follows:

Changes in manuscript: The *Results* section now reads: “Resting-state recordings of unilateral electrocorticography (ECoG) targeted at sensorimotor cortex and subthalamic LFP (STN-LFP) were performed through externalised leads in the days following surgical implantation of the electrodes (see Supplementary Table 2). The ECoG electrodes were later retracted at the time of neurostimulator implantation.”

8. For clarity to the reader, and improve readability, I would invite the authors to include references to main papers proposing these key theories, in the following statement : “to this theory with the demonstration that: a) cortex drives subthalamic activity; b) cortex – STN oscillatory connectivity is spatio-spectrally linked to hyperdirect and indirect pathway communication, as revealed through whole-brain MRI connectomics; and c) both dopamine and DBS suppress 1-9 ms monosynaptic input to the STN.”

Reply: We thank the reviewer for this suggestion and have added references to the appropriate publications.

Changes in manuscript: The *Discussion* section now reads: “Indeed, our study confirms multiple necessary considerations for this theory with the demonstration that: a) cortex drives subthalamic activity^{18,30}; b) cortex – STN oscillatory connectivity is spatio-spectrally linked to hyperdirect and indirect pathway communication^{18,30}, as revealed through whole-brain MRI connectomics; and c) both dopamine and DBS suppress 1-9 ms monosynaptic input to the STN^{33,35–37}.”

Remarks on code availability:

9. The code does provide a README file for usage. For the code in python, it provides a configuration file for the environment, which is greatly appreciated, and works great during testing. However, as the data is not provided, it is difficult to validate group analyses and other methods. There is a DEMO section to partially address this requirement, but that is not helpful in replicating the paper itself.

Reply: We are grateful to the reviewer for their feedback on the codebase. We are unfortunately unable to upload the patient data due to the General Data Protection Regulation of the European Union. However, data can be made available within this legal framework conditionally to data sharing agreements in accordance with data privacy statements signed by the patients. This is stated in the *Methods – Data Availability* section, alongside contact information for the relevant persons. We are happy to say that such data sharing agreements are already in place with several groups, and will gladly expand this as requested.

10. From the point of view of code organization, it is well structured and documented, and the analysis routines are easily useable. For the python codebase, the jupyter notebook are especially welcomed. Regarding code review, I wanted to add that the currently posted Zenodo link is non-functional. However the authors have posted the following GitHub: https://github.com/neuromodulation/manuscript-binns_cortex_stn_comm.

Reply: We apologise for any difficulties with the link to the Zenodo archive. In line with the *Nature* reporting standards, we now also include a working link pointing to the latest version of the Zenodo archive in the *Methods* section.

Changes in manuscript: The *Methods* section now includes the following: “All code is made publicly available on GitHub (github.com/neuromodulation/manuscript-binns_cortex_stn_comm) and archived on Zenodo (DOI: 10.5281/zenodo.10974655).”

Response to Reviewers

Shared pathway-specific network mechanisms of dopamine and deep brain stimulation for the treatment of Parkinson's disease

Authors

Thomas S. Binns^{1,2,3}, Richard M. Köhler¹, Jojo Vanhoecke¹, Meera Chikermane¹, Moritz Gerster^{3,4,5}, Timon Merk¹, Franziska Pellegrini^{3,6}, Johannes L. Busch¹, Jeroen G.V. Habetts¹, Alessia Cavallo^{1,2,3}, Jean-Christin Beyer¹, Bassam Al-Fatly¹, Ningfei Li¹, Andreas Horn^{1,2,7,8}, Patricia Krause¹, Katharina Faust⁹, Gerd-Helge Schneider⁹, Stefan Haufe^{2,3,6,10,11}, Andrea A. Kühn^{1,2,3,12,13}, Wolf-Julian Neumann^{1,2,3,*}

Affiliations

¹Movement Disorder and Neuromodulation Unit, Department of Neurology, Charité – Universitätsmedizin Berlin, corporate member of Freie Universität Berlin and Humboldt-Universität zu Berlin, Berlin, Germany

²Einstein Center for Neurosciences Berlin, Charité – Universitätsmedizin Berlin, corporate member of Freie Universität Berlin and Humboldt-Universität zu Berlin, Berlin, Germany

³Bernstein Center for Computational Neuroscience Berlin, Berlin, Germany

⁴Research Group Neural Interactions and Dynamics, Department of Neurology, Max Planck Institute for Human Cognitive and Brain Sciences, Leipzig, Germany

⁵Neurophysics Group, Department of Neurology, Charité – Universitätsmedizin Berlin, corporate member of Freie Universität Berlin and Humboldt-Universität zu Berlin, Berlin, Germany

⁶Berlin Center for Advanced Neuroimaging, Bernstein Center for Computational Neuroscience, Berlin, Germany

⁷Center for Brain Circuit Therapeutics, Department of Neurology, Brigham and Women's Hospital, Harvard Medical School, Boston, MA, USA

⁸Department of Neurosurgery, Massachusetts General Hospital, Harvard Medical School, Boston, MA, USA

⁹Department of Neurosurgery, Charité – Universitätsmedizin Berlin, corporate member of Freie Universität Berlin and Humboldt-Universität zu Berlin, Berlin, Germany

¹⁰Technische Universität Berlin, Berlin, Germany

¹¹Physikalisch-Technische Bundesanstalt Braunschweig und Berlin, Berlin, Germany

¹²NeuroCure Clinical Research Centre, Charité – Universitätsmedizin Berlin, corporate member of Freie Universität Berlin and Humboldt-Universität zu Berlin, Berlin, Germany

¹³Berlin School of Mind and Brain, Humboldt-Universität zu Berlin, Berlin, Germany

*Correspondence: julian.neumann@charite.de

Reviewer 1

The authors have responded appropriately to reviewer comments.

Reply: We deeply appreciate your valuable feedback, which has greatly improved the quality of our revised manuscript.

Reviewer 2

Thank you for the extensive revision of the previous comments. The extra analyses and results from added participants in the revised version are convincing. I think the manuscript will add new knowledge to the field and will be of great interest to the community. However, I still have one remaining major concern and some minor comments, as detailed below:

Major:

I appreciate the idea of linking oscillatory connectivity measures with normative MRI-based whole-brain connectomics. However, I am unclear about the rationale for using 'Medication' as an independent predictor variable, alongside oscillatory spatial patterns, to predict the response variable of dMRI-/fMRI-based connectivity using the following model architecture: "dMRI/fMRI connectivity ~ Spatial patterns + Medication + (1 | Subject)".

As far as I'm aware of, the dMRI-/fMRI-based connectivity is derived purely from the coordinates of the DBS/ECOG electrodes, the connectivity should be identical for the same ECOG-LFP contact pair across OFF therapy, ON levodopa, and ON STN-DBS conditions. My concerns with this analysis include:

1. You have already shown that Medication significantly reduces the maximized imaginary coherence in the high beta band (Fig. 3d). In the aforementioned model, this will almost certainly result in a positive and significant coefficient for high beta imaginary coherence. For example:

- $Y = A * \text{Beta_High_1} + B * \text{Med_OFF};$

- $Y = A * \text{Beta_High_2} + B * \text{Med_ON};$

Here, $Y > 0$, $\text{Beta_High_1} > \text{Beta_High_2} > 0$, $\text{Med_ON} = 1$, and $\text{Med_OFF} = 0$ (or -1, I assume). Thus, coefficient A will be positive with a significant p-value.

Reply: The reviewer is indeed correct that the suppression of connectivity values with dopamine would bias the model towards a significant effect for high beta coupling. However, we wish to note that the *spatial patterns* describing the contributions to oscillatory connectivity in the maximised imaginary coherency analysis will not necessarily reflect the changes in the *degree* of connectivity across medication or stimulation states.

For instance, in the OFF therapy state, there is a strong level of high beta cortico-subthalamic imaginary coherency, which the spatial patterns reveal to be focal over motor cortex and dorsolateral STN (Supplementary Fig. 7a-b). With dopamine, while the level of high beta connectivity is suppressed, similar spatial patterns can be expected if the motor cortex and dorsolateral STN still contribute to what high beta coupling remains. This is what we indeed observe, such that although the degree of coupling is suppressed with medication, the areas contributing to this connectivity are consistent (Supplementary Fig. 7a-b). Therefore, it is not guaranteed that changes in the degree of oscillatory coupling with therapy will be accompanied by significant effects in the linear mixed effects models comparing *spatial patterns* of oscillatory coupling with dMRI/fMRI connectivity.

Supplementary Figure 7: Spatial maps of cortico-subthalamic coupling. Localisation of the strongest high beta connectivity component extracted from the maximised imaginary coherency analysis. Localisations are highly similar across therapeutic states for the OFF therapy – ON levodopa group in the **a** cortex and **b** STN, as well as for the OFF therapy – ON STN-DBS group in the **c** cortex and **d** STN. Localisations are shown for individual electrodes and interpolated to surfaces. Abbreviations: A – anterior; ECoG – electrocorticography; I – inferior; L – left; P – posterior; R – right; S – superior; STN – subthalamic nucleus.

We acknowledge that the multivariate methods and their corresponding spatial patterns are relatively novel methods in the context of Parkinson’s disease neurophysiology research, which can introduce difficulties in the interpretation of these results. We apologise for any lack of clarity in their presentation, and we now provide additional context for interpreting the spatial patterns in the main text.

Changes in manuscript: The *Results* section now reads: “This confirmed a significant, positive association of hyperdirect fibre counts and the contribution of contacts to high beta oscillatory connectivity ($\beta = 4.064$, $p < 0.05$), but not low beta oscillatory connectivity ($\beta = -1.142$, $p > 0.05$). In both cases, medication state did not have a significant effect on this relationship (both $p > 0.05$). This is not unexpected, as although the degree of high beta coupling changes with dopamine, the multivariate spatial patterns capture the amount a given channel contributes to the connectivity that is present. Therefore, medication state will not have a significant effect on the relationship between hyperdirect pathway connectivity and high beta coupling in our model if the spatial patterns of coupling are stable across medication states, which is indeed the case (Supplementary Fig. 7).”

2. The goal here is to determine if the effect of levodopa (and similarly for STN-DBS) on high/low beta oscillatory connectivity measures can be explained by the underlying dMRI-/fMRI-based connectivity. I believe a more intuitive model would be:

- “Spatial patterns \sim dMRI/fMRI connectivity + Medication + (1 | Subject)”.

This way, you can investigate whether both medication and dMRI/fMRI connectivity independently contribute to the variance in ‘spatial patterns’ (high/low beta).

Reply: We agree with the reviewer that to determine whether the therapeutic effect on oscillatory coupling is explained by dMRI/fMRI connectivity, using models with the suggested architecture would be more intuitive. However, we respectfully note that this is not the goal of the dMRI/fMRI analyses. Rather, we are more simply trying to associate oscillatory coupling with activity of the indirect and hyperdirect pathways – specifically high beta – hyperdirect pathway interactions and low beta – indirect pathway interactions.

The relationship between high beta coupling and the hyperdirect pathway was previously demonstrated in Oswal *et al.* (2021) [18]. As such, whilst us showing this is not a novel effect, it is a critical point in the assertion of the hyperdirect pathway as a key component of therapeutic action in Parkinson’s disease. It is therefore essential that we can also demonstrate an association of hyperdirect pathway connectivity – high beta coupling using our own data, replicating this highly relevant finding in an independent data set.

Ultimately, altering the design of the linear mixed effects models as suggested does not meaningfully alter the outcome of the results. For the structural connectivity of the hyperdirect pathway, it remains the case that there is a selective association of high beta coupling with the number of adjacent fibres ($p < 0.05$), with no significant effect for low beta coupling ($p > 0.05$). Additionally, there remains no significant effect of medication on these interactions ($p > 0.05$), again reflecting the similarity of spatial contributions to oscillatory coupling across medication states. It is a similar case for the functional connectivity between the cortex and basal ganglia nuclei, with significant interactions for low beta coupling to the putamen, GPe, and STN ($p < 0.05$; Bayesian information criterion = 435), and a similar but less robust effect for high beta coupling ($p < 0.05$; Bayesian information criterion = 493). Furthermore, there is no significant effect of medication state on the interactions (both $p > 0.05$). As such, these findings match those using the existing model constructions.

We fully recognise that exploring the associations between therapeutic effects on oscillatory connectivity and the underlying structural/functional connectivity is a valid line of enquiry. However, our ability to investigate such an association using the spatial contributions to oscillatory coupling is inherently limited, given the previously discussed similarities in the contributions to coupling across medication states. Instead, we have opted for the current

analyses which allow for the use of these high signal-to-noise ratio maps of coupling contributions. This provides the necessary insights for our goals of associating oscillatory coupling with particular cortex – STN pathways, findings which are not altered by shifting the response and predictor variables.

We sincerely apologise for the lack of clarity in our presentation of the motivation behind the structural and functional connectivity analyses. We have now added additional context to the *Results* and *Methods* sections to clarify our intended goals. If these changes to better describe the intentions of the analyses are not deemed adequate, we are willing to adapt the model designs as suggested.

Changes in manuscript: The introduction to the *Results* section now reads: “Finally, coupling metrics were correlated with whole-brain MRI connectivity derived from precisely curated fibre tracts for DBS research²⁵ (Fig. 1d) and large-scale normative fMRI connectomes from PD patients^{26,27} (Fig. 1e) to determine the association between cortex – STN signalling pathways and spectral coupling bands.”

This is reiterated later in the *Results* section for the structural connectivity analysis: “To analyse the relationship between hyperdirect pathway structural connectivity and oscillatory connectivity in specific frequency bands, we used a linear mixed effects model with medication state as a fixed effect and subjects as a random variable.”

The response to comment #1 above also adds context to the lack of observed significant effect for medication state: “In both cases, medication state did not have a significant effect on this relationship (both $p > 0.05$). This is not unexpected, as although the degree of high beta coupling changes with dopamine, the multivariate spatial patterns capture the amount a given channel contributes to the connectivity that is present. Therefore, medication state will not have a significant effect on the relationship between hyperdirect pathway connectivity and high beta coupling in our model if the spatial patterns of coupling are stable across medication states, which is indeed the case (Supplementary Fig. 7).”

A similar addition is made for the functional connectivity analysis: “To investigate the relationship of oscillatory connectivity in specific frequency bands with indirect pathway connectivity, we repeated the multimodal analysis using functional MRI connectivity derived from an openly available Parkinson’s disease fMRI group connectome...”

And the lack of medication effect is now described further: “There were no instances in which medication state had a significant effect on these relationships (all $p > 0.05$), again reflecting the similarity of spatial contributions to coupling across medication states (Supplementary Fig. 7).”

Finally, the goals of the structural and functional connectivity analyses are further clarified in the *Methods* section:

“For the OFF therapy and ON levodopa recordings, the number of fibres connecting a given ECoG and STN-LFP contact were then compared to the average low (12-20 Hz) and high (20-30 Hz) beta band spatial contribution maps extracted from maximised imaginary coherency (see *Oscillatory connectivity analysis*) averaged over the respective ECoG and STN-LFP contacts. This enabled us to determine the spectral associations of the hyperdirect pathway.”

“For the OFF therapy and ON levodopa recordings, these bivariate connectivity values were compared to the average low (12-20 Hz) and high (20-30 Hz) beta band spatial contribution maps extracted from maximised imaginary coherency (see *Oscillatory connectivity analysis*) for the respective ECoG contacts. This enabled us to determine the spectral associations of the indirect pathway.”

3. You could also include STN-DBS in the same model as another independent predictor variable, as it also significantly affects 'spatial patterns' as previously discussed.

Reply: The reviewer raises a valid point that stimulation state could also be included as an additional fixed effect in the linear mixed effects models. However, we note that, like for dopamine, DBS does not notably alter the spatial contributions to oscillatory coupling. Given the minimal effect of DBS on these patterns and their similarity to the OFF therapy – ON levodopa group (i.e., high beta coupling being focal between motor cortex and dorsolateral STN), the spatial contributions in the OFF therapy – ON STN-DBS group were not shown in the previous manuscript iterations.

However, we recognise that this risks potential confusion over the nature of stimulation's effect on the spatial contributions to oscillatory coupling. As such, we now include the spatial contribution maps for the OFF therapy – ON STN-DBS group in Supplementary Fig. 7c-d. Indeed, they are highly similar across stimulation states, like for medication states.

Supplementary Figure 7: Spatial maps of cortico-subthalamic coupling. Localisation of the strongest high beta connectivity component extracted from the maximised imaginary coherency analysis. Localisations are highly similar across therapeutic states for the OFF therapy – ON levodopa group in the **a** cortex and **b** STN, as well as for the OFF therapy – ON STN-DBS group in the **c** cortex and **d** STN. Localisations are shown for individual electrodes and interpolated to surfaces. Abbreviations: A – anterior; ECoG – electrocorticography; I – inferior; L – left; P – posterior; R – right; S – superior; STN – subthalamic nucleus.

Furthermore, similar to the suggestion in comment #2 of flipping the response and predictor variables, the addition of stimulation data and the inclusion of stimulation state as a fixed effect does not meaningfully alter the conclusions. For structural connectivity, it again remains the case that there is a selective association of high beta coupling with the number of adjacent hyperdirect pathway fibres ($p < 0.05$), with no significant effect seen for low beta coupling ($p > 0.05$). Additionally, there remains no significant effects of medication state (both $p > 0.05$), and

no significant effects are introduced for stimulation state (both $p > 0.05$). Again, it is a similar case for the functional connectivity analysis, where for connectivity between the cortex and the Putamen, GPe, and STN, no significant effects are observed for medication or stimulation states for low or high beta coupling (all $p > 0.05$).

Notably, we wish to highlight potential concerns over introducing OFF therapy – ON STN-DBS group data into the same model as the OFF therapy – ON levodopa group data. For those subjects in both groups, there are two non-identical OFF therapy observations. This is due to the fact that for comparability of the OFF therapy – ON STN-DBS recordings, those STN-LFP channels missing due to stimulation are also removed from the OFF therapy data. In turn, the multivariate spatial contribution maps for the OFF therapy data in the medication group correspond to a discrete set of channels than in the stimulation group, and are therefore not comparable. This risks detrimentally affecting the degree to which the linear mixed effects models can characterise the nature of OFF therapy oscillatory coupling. While this does not affect the core results as discussed above, it is nonetheless a limitation worth considering.

The alternative approach to overcome this limitation is to generate separate models for the OFF therapy – ON levodopa and OFF therapy – ON STN-DBS groups. However, this introduces an additional layer of complexity in the analyses in what is already an extensive set of results. Ultimately, given the striking similarity of spatial contributions to oscillatory coupling across stimulation states, the existing observations that dMRI/fMRI connectivity – oscillatory coupling relationships are not significantly altered by medication state, and the fact that this does not alter the conclusions over specific spectral associations of cortex – STN communication pathways, this alteration does not add meaningfully to the analyses. We have updated the *Methods* section to mention this choice. If the explicit inclusion of the spatial contributions for the OFF therapy – ON STN-DBS group are deemed to not provide sufficient context for their omission from the linear mixed effects models, we are willing to adapt the model designs as suggested.

Changes in manuscript: The *Methods* section now reads for the dMRI and fMRI connectivity analysis: “Given the similarity of spatial contribution maps in the levodopa and STN-DBS groups (Supplementary Fig. 7), only the spatial contribution maps from the larger levodopa group were considered. Including the STN-DBS maps did not meaningfully alter the interpretation of the results.”

4. It would be helpful to provide details of each linear mixed effect model in a supplementary table to make it easier to follow the presented results.

Reply: We fully agree with the reviewer that providing details for the linear mixed effects models would help to follow the respective results. We now include this information in Supplementary Tables 3-10 and thank the reviewer for this useful suggestion to improve the interpretation of the presented results.

Supplementary Table 3: Hyperdirect pathway fibres ~ high beta oscillatory connectivity.

	Beta coefficient (A.U.)	Standard error (A.U.)	z	p > z	[0.025	0.975]
Intercept	7.917	1.345	5.888	< 0.001	5.281	10.552
Spatial patterns	4.064	0.850	-0.088	< 0.001	2.398	5.731
C(Medication)[ON]	-0.061	0.692	4.780	0.930	-1.416	1.295
Subject Var	28.113	0.758				

$n_{\text{observations}} = 1676$; $n_{\text{groups}} = 18$; mean group size = 93.1 (range 48-168); scale = 200.306; Log-likelihood = -6839.031; Bayesian information criterion = 13707.759; $R^2_{\text{marginal/conditional}} = 0.037/0.155$. Medication state was used as a fixed condition. Subjects were treated as a random effect.

Supplementary Table 4: Hyperdirect pathway fibres ~ low beta oscillatory connectivity.

	Beta coefficient (A.U.)	Standard error (A.U.)	z	p > z	[0.025	0.975]
Intercept	7.997	1.197	6.682	< 0.001	5.651	10.343
Spatial patterns	-1.142	0.790	-1.446	0.148	-2.689	0.406
C(Medication)[ON]	0.018	0.697	0.026	0.979	-1.348	1.384
Subject Var	21.298	0.574				

$n_{\text{observations}} = 1676$; $n_{\text{groups}} = 18$; mean group size = 93.1 (range 48-168); scale = 203.449; Log-likelihood = -6849.839; Bayesian information criterion = 13729.374; $R^2_{\text{marginal/conditional}} = 0.003/0.097$. Medication state was used as a fixed condition. Subjects were treated as a random effect.

Supplementary Table 5: cortex – STN fMRI connectivity ~ low beta oscillatory connectivity.

	Beta coefficient (A.U.)	Standard error (A.U.)	z	p > z	[0.025	0.975]
Intercept	-0.048	0.003	-18.297	< 0.001	-0.053	-0.043
Spatial patterns	0.006	0.002	2.967	0.007	0.002	0.011
C(Medication)[ON]	< -0.001	0.003	< -0.001	1.000	-0.006	0.006
Subject Var	< 0.001	0.002				

$n_{\text{observations}} = 214$; $n_{\text{groups}} = 18$; mean group size = 11.9 (range 10-12); scale = $5e^{-4}$; Log-likelihood = 489.321; Bayesian information criterion = -957.179; $R^2_{\text{marginal/conditional}} = 0.063/0.131$. Medication state was used as a fixed condition. Subjects were treated as a random effect.

Supplementary Table 6: cortex – STN fMRI connectivity ~ high beta oscillatory connectivity.

	Beta coefficient (A.U.)	Standard error (A.U.)	z	p > z	[0.025	0.975]
Intercept	-0.048	0.003	-18.840	< 0.001	-0.053	-0.043
Spatial patterns	0.004	0.002	2.006	0.045	< 0.001	0.008
C(Medication)[ON]	< -0.001	0.003	< -0.001	1.000	-0.006	0.006
Subject Var	< 0.001	0.001				

$n_{\text{observations}} = 214$; $n_{\text{groups}} = 18$; mean group size = 11.9 (range 10-12); scale = $5e^{-4}$; Log-likelihood = 486.911; Bayesian information criterion = -952.359; $R^2_{\text{marginal/conditional}} = 0.028/0.080$. Medication state was used as a fixed condition. Subjects were treated as a random effect.

Supplementary Table 7: cortex – Putamen, GPe, and STN fMRI connectivity ~ low beta oscillatory connectivity.

	Beta coefficient (A.U.)	Standard error (A.U.)	z	p > z	[0.025	0.975]
Intercept	-0.050	0.003	-15.851	< 0.001	-0.056	-0.043
Spatial patterns	0.005	0.002	2.271	0.023	0.001	0.010
C(Medication)[ON]	< 0.001	0.003	< 0.001	1.000	-0.006	0.006
Subject Var	< 0.001	0.003				

$n_{\text{observations}} = 214$; $n_{\text{groups}} = 18$; mean group size = 11.9 (range 10-12); scale = $5e^{-4}$; Log-likelihood = 494.516; Bayesian information criterion = -967.567; $R^2_{\text{marginal/conditional}} = 0.049/0.219$. Medication state was used as a fixed condition. Subjects were treated as a random effect.

Supplementary Table 8: cortex – Putamen, GPe, and STN fMRI connectivity ~ high beta oscillatory connectivity.

	Beta coefficient (A.U.)	Standard error (A.U.)	z	p > z	[0.025	0.975]
Intercept	-0.050	0.003	-16.486	< 0.001	-0.056	-0.044
Spatial patterns	0.004	0.002	1.968	0.049	< 0.001	0.008
C(Medication)[ON]	< -0.001	0.003	< -0.001	1.000	-0.006	0.006
Subject Var	< 0.001	0.002				

$n_{\text{observations}} = 214$; $n_{\text{groups}} = 18$; mean group size = 11.9 (range 10-12); scale = $5e^{-4}$; Log-likelihood = 493.632; Bayesian information criterion = -965.799; $R^2_{\text{marginal/conditional}} = 0.029/0.180$. Medication state was used as a fixed condition. Subjects were treated as a random effect.

Supplementary Table 9: cortex – Putamen and GPe fMRI connectivity ~ low beta oscillatory connectivity.

	Beta coefficient (A.U.)	Standard error (A.U.)	z	p > z	[0.025	0.975]
Intercept	-0.050	0.003	-14.986	< 0.001	-0.056	-0.043
Spatial patterns	0.005	0.002	2.055	0.040	< 0.001	0.010
C(Medication)[ON]	< -0.001	0.003	< -0.001	1.000	-0.006	0.006
Subject Var	< 0.001	0.003				

$n_{\text{observations}} = 214$; $n_{\text{groups}} = 18$; mean group size = 11.9 (range 10-12); scale = $5e^{-4}$; Log-likelihood = 484.166; Bayesian information criterion = -946.869; $R^2_{\text{marginal/conditional}} = 0.039/0.218$. Medication state was used as a fixed condition. Subjects were treated as a random effect.

Supplementary Table 10: cortex – Putamen and GPe fMRI connectivity ~ high beta oscillatory connectivity.

	Beta coefficient (A.U.)	Standard error (A.U.)	z	p > z	[0.025	0.975]
Intercept	-0.050	0.003	-15.484	< 0.001	-0.056	-0.044
Spatial patterns	0.004	0.002	1.842	0.065	< -0.001	0.008
C(Medication)[ON]	< -0.001	0.003	< -0.001	1.000	-0.006	0.006
Subject Var	< 0.001	0.002				

$n_{\text{observations}} = 214$; $n_{\text{groups}} = 18$; mean group size = 11.9 (range 10-12); scale = $5e^{-4}$; Log-likelihood = 483.579; Bayesian information criterion = -945.693; $R^2_{\text{marginal/conditional}} = 0.025/0.188$. Medication state was used as a fixed condition. Subjects were treated as a random effect.

Minor comments:

1. I suggest specifying imaginary coherence at the beginning, instead of coherence-based measures, as it is too broad.

Reply: We recognise the reviewer’s point that “coherency-based” measures is an overly broad description. We have improved the description of the methods when they are first presented in the *Introduction* and *Results* sections according to the reviewer’s suggestion.

Changes in manuscript: The *Introduction* section now reads: “We characterised neurophysiological cortex – STN interactions using: imaginary coherency-based measures for undirected spectral coupling^{19,20}...”

The *Results* section now reads: “For this, we utilised three distinct analytic approaches: 1) spatio-spectral patterns of undirected communication with imaginary coherency-based metrics^{19,20}...”

2. In supplementary Figure 5, please update the results to reflect the new cohorts, i.e., N=18 and 12 for dopamine and STN-DBS, respectively.

Reply: We note that the group sizes in Supplementary Fig. 5 (n = 17 for medication and n = 11 for stimulation) are up to date. Although the full groups are $n_{\text{medication}} = 18$ and $n_{\text{stimulation}} = 12$, one subject did not have electrode coverage of the sensory cortex, which the results in this figure are shown for. We have now updated the figure legend to clarify this point, and thank the reviewer for highlighting this source of potential confusion.

Changes in manuscript: The figure legend now includes: “Note that of the n = 18 (medication) and n = 12 (stimulation) subjects, one subject had no electrode coverage of the sensory cortex, producing the n = 17 and n = 11 group sizes seen here, respectively.”

3. In Figure 4, please provide R^2 values for both models.

Reply: In Fig. 4, the two scatter plots shown for each panel contain information from the same linear mixed effects model: 1) the relationship between the empirical structural/functional connectivity and that estimated by the model; and 2) the relationship between the oscillatory connectivity and the structural/functional connectivity estimated by the model. Since the two scatter plots correspond to the same model, there is only one R^2 value.

Currently, the single R^2 value was shown on the first scatter plot, given that this plot provides more insight into the quality of the model fit. However, the two plots do show separate p-values, where the p-value in the first scatter plot shows that there is a significant relationship between the empirical and estimated data, and the p-value in the second scatter plot matches what is reported in the *Results* section main text where the relationship between oscillatory and structural/functional connectivity is presented.

We recognise that this may lead to confusion in the interpretation of the scatter plots. We have therefore moved the R^2 value to a more central position that visually represents the association of the R^2 value to both plots. The figure legend has also been updated to distinguish the R^2 value from any one scatter plot. We thank the reviewer for highlighting this source of potential confusion.

Figure 4: Cortico-subthalamic structural and functional connectivity. **a** Contribution of ECoG and STN-LFP contacts to high beta cortico-subthalamic connectivity correlates significantly with the number of hyperdirect pathway fibres connecting these regions. Hyperdirect pathway fibres are coloured according to the grand average contributions of ECoG and STN-LFP contacts to high beta cortico-subthalamic connectivity (obtained from maximised imaginary coherency). The inset shows the relationship between the estimated number of hyperdirect pathway fibres from the linear mixed effects model versus the empirical number of fibres (top; correlation coefficient, r , and p -value represent the quality of the linear mixed effects model as determined using the Pearson correlation) and the contributions to high beta oscillatory connectivity (bottom; beta coefficient and p -value reflect that reported in text and Supplementary Table 3). The conditional R^2 value representing the quality of the model is also shown. **b** Contribution of ECoG contacts to low beta cortico-subthalamic connectivity (obtained from maximised imaginary coherency) correlates significantly with the functional MRI connectivity of ECoG contact locations to indirect pathway nuclei. Upper inset shows the beta coefficients of the models for the caudate, putamen, GPe, and STN. Lower inset shows the relationship between the estimated functional connectivity from

cortex to putamen, GPe, and STN versus the empirical functional connectivity to these regions (left; correlation coefficient, r , and p -value represent the quality of the linear mixed effects model as determined using the Pearson correlation), as well as the contributions of ECoG contacts to low beta connectivity (right; beta coefficient and p -value reflect that reported in text and Supplementary Table 7). The conditional R^2 value representing the quality of the model is also shown. * $p < 0.05$. Abbreviations: GPe – external segment of the globus pallidus; n.s. – not significant; STN – subthalamic nucleus.

Changes in manuscript: The legend for Fig. 4a now reads: “The inset shows the relationship between the estimated number of hyperdirect pathway fibres from the linear mixed effects model versus the empirical number of fibres (top; correlation coefficient, r , and p -value represent the quality of the linear mixed effects model as determined using the Pearson correlation) and the contributions to high beta oscillatory connectivity (bottom; beta coefficient and p -value reflect that reported in text and Supplementary Table 3). The conditional R^2 value representing the quality of the model is also shown.”

The legend for Fig. 4b now reads: “Upper inset shows the beta coefficients of the models for the caudate, putamen, GPe, and STN. Lower inset shows the relationship between the estimated functional connectivity from cortex to putamen, GPe, and STN versus the empirical functional connectivity to these regions (left; correlation coefficient, r , and p -value represent the quality of the linear mixed effects model as determined using the Pearson correlation), as well as the contributions of ECoG contacts to low beta connectivity (right; beta coefficient and p -value reflect that reported in text and Supplementary Table 7). The conditional R^2 value representing the quality of the model is also shown.”

Reviewers 3 & 4

Reviewer 3

Reply: Again, we are very thankful for your valuable input!

Reviewer 4

Remarks to the author:

I wanted to thank the authors as I have appreciated the responsiveness and completeness in addressing all the points and concerns that were raised by me and other reviewers. I do believe that the manuscript has been significantly strengthened and is now clearer, including the revision of the figures (due to the slight change in statistical procedures, such as number of folds). I do appreciate the revision of the limitations sections as well. I accept the manuscript in its current form.

Remarks on code availability:

The code does provide a README file for usage, and I maintain the comments expressed in my previous revision. I have appreciated the revision of the zenodo link and inclusion of the github as well.

Regarding data availability, the authors have expressed an understandable limitation in direct open sharing. As a general point, my comment was in light of the limitations of my review of the results, as I can only give direct insight on code usage for the demo data portion.

For the code in python, it provides a configuration file for the environment, which is greatly appreciated, and works great during testing. From the point of view of code organization, it is well structured and documented, and the analysis routines are easily useable. For the python codebase, the jupyter notebook are especially welcomed.

Reply: We are very grateful for your understanding in these matters.